# The Eocene-Oligocene transition: a review of marine and terrestrial proxy data, models and model-data comparisons

David K. Hutchinson[1], Helen K. Coxall[1], Daniel J. Lunt[2], Margret Steinthorsdottir[1,3], Agatha M. De Boer[1], Michiel Baatsen[4], Anna von der Heydt[4,5], Matthew Huber[6], Alan T. Kennedy-Asser[2], Lutz Kunzmann[7], Jean-Baptiste Ladant[8], Caroline H. Lear[9], Karolin Moraweck[7], Paul N. Pearson[9], Emanuela Piga[9], Matthew J. Pound[10], Ulrich Salzmann[10], Howie D. Scher[11], Willem P. Sijp[12], Kasia K. Śliwińska[13], Paul A. Wilson[14], Zhongshi Zhang[15,16]

1. Department of Geological Sciences and Bolin Centre for Climate Research, Stockholm University, Stockholm, Sweden
2. School of Geographical Sciences, University of Bristol, UK
3. Department of Palaeobiology, Swedish Museum of Natural History, Stockholm, Sweden
4. Institute for Marine and Atmospheric Research, Department of Physics, Utrecht University, Utrecht, the Netherlands
5. Centre for Complex Systems Studies, Utrecht University, Utrecht, the Netherlands
6. Department of Earth, Atmospheric, and Planetary Sciences, Purdue University, West Lafayette, USA
7. Senckenberg Natural History Collections Dresden, Germany
8. Department of Earth and Environmental Sciences, University of Michigan, USA
9. School of Earth and Ocean Sciences, Cardiff University, Cardiff, UK
10. Department of Geography and Environmental Sciences, Northumbria University, UK
11. School of the Earth, Ocean and Environment, University of South Carolina, USA
12. Climate Change Research Centre, University of New South Wales, Sydney, Australia
13. Department of Stratigraphy, Geological Survey of Denmark and Greenland (GEUS), Copenhagen, Denmark
14. University of Southampton, National Oceanography Centre Southampton, UK
15. Department of Atmospheric Science, China University of Geoscience, Wuhan, China
16. NORCE Research and Bjerknes Centre for Climate Research, Bergen, Norway

*Correspondence to*: David K. Hutchinson (david.hutchinson@geo.su.se)

**Abstract.** The Eocene-Oligocene transition (EOT) was a climate shift from a largely ice-free greenhouse world to an icehouse climate, involving the first major glaciation of Antarctica and global cooling occurring ~34 million years ago (Ma) and lasting ~790 kyr. The change is marked by a global shift in deep sea $\delta^{18}O$ representing a combination of deep-ocean cooling and growth in land ice volume. At the same time, multiple independent proxies for ocean temperature indicate sea surface cooling, and major changes in global fauna and flora record a shift toward more cold-climate adapted species. The two principal suggested explanations of this transition are: a decline in atmospheric $CO_2$, and changes to ocean gateways, while orbital forcing likely influenced the precise timing of the glaciation. Here we review and synthesize proxy evidence of paleogeography, temperature, ice sheets, ocean circulation, and $CO_2$ change from the marine and terrestrial realms. Furthermore, we quantitatively compare proxy records of change to an ensemble of climate model simulations of temperature change across the EOT. The simulations compare three forcing mechanisms across the EOT: $CO_2$ decrease, paleogeographic

changes, and ice sheet growth. Our model ensemble results demonstrate the need for a global cooling mechanism beyond the imposition of an ice sheet or paleogeographic changes. We find that $CO_2$ forcing involving a large decrease in $CO_2$ of ca. 40% (~325 ppm drop) provides the best fit to the available proxy evidence, with ice sheet and paleogeographic changes playing a secondary role. While this large decrease is consistent with some $CO_2$ proxy records (the extreme end member of decrease), the positive feedback mechanisms on ice growth are so strong that a modest $CO_2$ decrease beyond a critical threshold for ice sheet initiation is well capable of triggering rapid ice-sheet growth. Thus, the amplitude of $CO_2$ decrease signalled by our data-model comparison should be considered an upper- estimate and perhaps artificially large, not least because the current generation of climate models do not include dynamic ice sheets and in some cases may be under-sensitive to $CO_2$ forcing. The model ensemble also cannot exclude the possibility that paleogeographic changes could have triggered a reduction in $CO_2$.

# 1 Introduction

## 1.1 Scope of Review

Since the last major review of the EOT (Coxall and Pearson, 2007) the fields of palaeoceanography and palaeoclimatology have advanced considerably. New proxy techniques, drilling and field archives of Cenozoic (66 Ma to present) climates, have expanded global coverage and added increasingly detailed views of past climate patterns, forcings and feedbacks. From a broad perspective, statistical interrogation of an astronomically dated, continuous composite of benthic foraminifera isotope records confirms that the EOT is the most prominent climate transition of the whole Cenozoic and suggests that the polar ice sheets that ensued seem to play a critical role in determining the predictability of Earth's climatological response to astronomical forcing (Westerhold et al., 2020). New proxy records capture near and far-field signals of the onset of Antarctic glaciation. Meanwhile, efforts to simulate the onset of the Cenozoic 'icehouse', using the latest and most sophisticated climate models, have also progressed. Here we review both observations and the results of modelling experiments of the EOT. From the marine realm, we review records of sea surface temperature, as well as deep sea time series of the temperature and land ice proxy $\delta^{18}O$ and carbon cycle proxy $\delta^{13}C$. From the terrestrial realm we cover plant records and biogeochemical proxies of temperature, $CO_2$ and vegetation change. We summarise the main evidence of temperature, glaciation and carbon cycle perturbations and constraints on the terrestrial ice extent during the EOT, and review indicators of ocean circulation change and deep water formation, including how these changes reconcile with paleogeography, in particular, ocean gateway effects.

Finally, we synthesise existing model experiments that test three major proposed mechanisms driving the EOT: (i) paleogeography changes, (ii) greenhouse forcing and (iii) ice sheet forcing upon climate. We highlight what has been achieved from these modelling studies to illuminate each of these mechanisms, and explain various aspects of the observations. We also discuss the limitations of these approaches, and highlight areas for future work. We then combine and synthesise the observational and modelling aspects of the literature in a model-data intercomparison of the available models of the EOT. This approach allows us to assess the relative effectiveness of the three modelled mechanisms in explaining the EOT observations.

The paper is structured as follows: Section 1.2 defines the chronology of events around the EOT, and clarifies the terminology of associated events, transitions, and intervals, thereby setting the framework for the rest of the review. Section 2 reviews our understanding of palaeogeographic change across the EOT, and discusses proxy evidence for changes in ocean circulation and ice sheets. Section 3 synthesises marine proxy evidence for sea surface temperatures (SST) and deep-ocean temperature

change. Section 4 synthesises terrestrial proxy evidence for continental temperature change, with a focus on pollen-based reconstructions. Section 5 presents estimates of $CO_2$ forcing across the EOT, from geochemical and stomatal-based proxies. Section 6 qualitatively reviews previous modelling work, and Section 7 provides a new quantitative intercomparison of previous modelling studies, with a focus on model-data comparisons to elucidate the relative importance of different forcings across the EOT. Section 8 provides a brief conclusion.

**1.2 Terminology of the Eocene-Oligocene transition**

Palaeontological evidence has long established Eocene (56 to 34 Ma) warmth in comparison to a long term Cenozoic cooling trend (Lyell and Deshayes 1830, p. 99-100). As modern stratigraphic records improved, a prominent step in that cooling towards the end of the Eocene began to be resolved. This became evident in early oxygen isotope records ($\delta^{18}O$) derived from deep-sea benthic foraminifera, which show an isotope shift towards higher $\delta^{18}O$ values (Kennett and Shackleton, 1976;

Shackleton and Kennett, 1975) which was subsequently attributed to a combination of continental ice growth and cooling (Lear et al., 2008). In the 1980s the search was on for a suitable Global Stratotype Section and Point (GSSP) to define the Eocene-Oligocene boundary (EOB). Much of the evidence was brought together in an important synthesis edited by Pomerol and Premoli Silva (1986). The GSSP was eventually fixed at the Massignano outcrop section in the Marche region of Italy in 1992 (Premoli Silva and Jenkins, 1993) at the 19.0 m mark which corresponds to the extinction of the planktonic foraminifer

family Hantkeninidae (Coccione, 1988; Nocchi et al., 1986). By the conventions of stratigraphy, Massignano is the only place where the EOB is defined unambiguously; everywhere else the EOB must be correlated to it, whether by biostratigraphy, magnetostratigraphy, isotope stratigraphy or other methods.

Coxall and Pearson, (2007, p 352) described the EOT as "a phase of accelerated climatic and biotic change lasting 500 kyr that

began before and ended after the E/O boundary". Recognizing and applying this in practice turns out to be problematic due to variability in the pattern of $\delta^{18}O$ between records and on different timescales. Widespread records now show the positive $\delta^{18}O$ shift with increasing detail. A high-resolution record from ODP Site 1218 in the Pacific Ocean revealed two $\delta^{18}O$ and $\delta^{13}C$ 'steps' separated by a more stable 'plateau interval' (Coxall et al., 2005; Coxall and Wilson, 2011). The EOT brackets these isotopic steps with the EOB falling in the plateau between them (Coxall and Pearson, 2007; Coxall and Wilson, 2011; Dunkley

Jones et al., 2008; Pearson et al., 2008). However, while two-step $\delta^{18}O$ patterns have now been interpreted in other deep sea records, thus far largely from the Southern Hemisphere (Figure 1) (Bohaty et al., 2012; Borrelli et al., 2014; Coxall and Wilson, 2011; Langton et al., 2016; Pearson et al., 2008; Wade et al., 2012; Zachos et al., 1996), there is often ambiguity in their

identification. In particular, while the second $\delta^{18}O$ step, 'Step 2' of Coxall and Pearson (2007), is an abrupt and readily correlated feature, the first step (Step 1 of Pearson et al., 2008; EOT-1 of Katz et al. 2008) is often less prominent than at Site 1218 (Figure 1). Furthermore, some records have been interpreted to show more than two $\delta^{18}O$ steps (e.g. Katz et al. 2008). Benthic $\delta^{13}C$ records provide a powerful complementary stratigraphic tool. These also show a correlatable stepped pattern of increase across the EOT, although in detail $\delta^{18}O$ and $\delta^{13}C$ are not synchronous (Coxall et al., 2005; Zachos et al., 1996) and further complications arise in correlation to other sites (Coxall and Wilson, 2011). In attempting to synthesize the pattern across multiple sites we suggest that attempts to define and correlate an initial 'Step-1' are premature at this point and should await better resolved records. Nonetheless, we maintain a tentative Step-1 in our terminology because it is important for differentiating phases of cooling vs ice growth during the EOT.

Settling on a consistent terminology for other features of the EOT is also problematic because usage of certain terms has changed through time. In order to clarify the definition of two key stratigraphic features, we recommend using the following terms: (i) for the basal Oligocene $\delta^{18}O$ increase we suggest the term "**E**arliest **O**ligocene oxygen **I**sotope **S**tep" (**EOIS**) to denote the large isotope step that occurs well after the EOB and within the lower part of Chron C13n (Figure 1); ii) Early Oligocene Glacial Maximum (**EOGM**; Liu et al. 2004; Figure 1) to denote the peak-to-peak isotope stratigraphic interval, corresponding to most of Chron C13n (starting at the top of the EOIS). Other terms for these features have been used inconsistently in the literature (Table 1). For example, the term 'Oi-1' was originally defined (at DSDP Site 522) by Miller et al. (1991) as an isotope stratigraphic 'zone' between one oxygen isotope peak and another, corresponding to a duration of several millions of years. Here there was a distinction between the 'Zone Oi-1' and the 'Oi-1 Event', the latter being equivalent to our EOIS. Subsequent articles variously refer to 'Oi-1' as an extended isotope zone, the peak $\delta^{18}O$ value at the base of that zone, an extended phase of high $\delta^{18}O$ values in the lower Oligocene approximately synonymous with the EOGM, or the 'step' that led up to the peak value (see discussion and references in Coxall and Pearson, 2007, p. 352). The terms 'Oi-1a' and 'Oi-1b', originally defined by Zachos et al. (1996) as "…two distinct, 100 to 150-kyr-long glacial maxima…separated by an 'interglacial' ", have also been inconsistently applied in the literature and are now arguably an impediment to clear communication. Due to this ambiguity, we avoid the term 'Oi-1' here. Katz et al. (2008) referred to prominent oxygen isotope steps within the EOT as 'EOT-1' and 'EOT-2', which might seem a convenient nomenclature for the steps referred to here, but whereas 'EOT-1' arguably corresponds to the 'Step 1' of Coxall and Pearson (2007), 'EOT-2' was a separate feature identified in the St. Stephen's Quarry record some way below the level identified as 'Oi-1' (Katz et al. 2008, p330). Hence it is not appropriate to use 'EOT-2' to denote the second step. Note, for clarity, the EOIS is not instantaneous and several records show some 'intermediate' values; its inferred duration in the records presented here is in the tens of kyr (40 kyr at Site 1218, Coxall et al., 2005).

| Event | Abbr. | Definition | Correlation | Timing | Comment | Also known as |
|---|---|---|---|---|---|---|
| Early Oligocene Glacial Maximum | EOGM | Period of cold climate / glaciation in the early Oligocene corresponding to most of Magnetic Chron C13n | Peak to peak $\delta^{18}O$ stratigraphic interval starting at the top of the EOIS & extending to another peak around the top of C13n | 33.65 to ~33.16 Ma, ~490 kyr duration | Defined by Liu et al. (2004). The end of the EOGM may correspond to a 2nd $\delta^{18}O$ peak sometimes referred to as Oi-1b | Oi-1 (Zachos et al., 1996) including the separate $\delta^{18}O$ maxima Oi-1a and Oi-1b |
| Chron C13n | C13n | Interval of normal magnetic polarity in the early Oligocene broadly correlative with the EOGM | Between specific magnetic reversals | 33.705 - 33.157 Ma (GTS2012) | Very useful for correlation & dating when available | - |
| Eocene - Oligocene Transition | EOT | A phase of accelerated climatic & biotic change that began before and ended after the EOB | Stratigraphic interval between the extinction of *Discoaster saipanensis* & the top of the EOIS | Start = 'Top' *D. saipanensis* extinction event; End = end of EOIS $\delta^{18}O$ maximum event. Duration ~790 kyr. | Definition revised here after Coxall & Pearson (2007) | |
| Earliest Oligocene oxygen Isotope Step | EOIS | Short period of rapid $\delta^{18}O$ increase (0.7‰ or more) that occurred well after the EOB & within the lower part of Chron C13n | Stratigraphically above the EOB & within the lowermost part of Chron C13n | The peak is at ~33.65 Ma (GTS2012) Duration ~40 kyr. | Herein defined. Defines the end of the EOT and the start of the EOGM | The 'Oi1 Event' "… at the base of Zone Oi1", Miller et al. (1991); Oi-1a of Zachos et al. (1996) |
| Eocene Oligocene boundary | EOB | The stratigraphic boundary between the Eocene and Oligocene Epochs defined at the Massignano GSSP | Denoted by the extinction of the planktonic foraminifera *Hantkenina* in the marine realm | 33.9 Ma (GTS2012) 33.7 Ma (CK95) | - | Base Oligocene epoch; base Rupelian stage |
| Step-1 | - | The first step increase in $\delta^{18}O$ occurring shortly before the EOB in some records | Harder to identify & correlate than the EOIS. | ? ~34.15 Ma, Duration ~40 kyr. | - | EOT-1, Katz et al. (2008); 'Precursor glaciation', Scher et al. (2011) |
| Late Eocene Event | - | A transient late Eocene cool or glacial event near the start of the EOT | Transient interval of positive $\delta^{18}O$ seen in some records. | Onset is coincident (within 80 kyr analytical error; Coxall et al., 2005) with the *D. saipanensis* extinction (34.44 Ma) at Site 1218. | Defined by Katz et al. (2008) Defines the start of the EOT as defined herein | - |
| Priabonian oxygen isotope maximum | PrOM | A transient late Eocene cool or glacial event | Transient interval of positive $\delta^{18}O$ seen in some records well below the EOT | ~37.3 Ma. Duration ~140 kyrs, tentatively placed within Chron C17n.1n | Defined by Scher et al. (2014) | - |

**Table 1:** Summary of Eocene Oligocene terminology and approximate timings of events, as interpreted at the time of writing. Timescales referred to are GTS2012 (Gradstein et al., 2012) and CK95 (Cande and Kent, 1995).

This brings us back to the definition of the 'EOT'. Based on the stratigraphic record from Tanzania, Dunkley Jones et al. (2008) and Pearson et al. (2008) placed the base of the EOT at the extinction of the tropical warm-water nannofossil *Discoaster saipanensis,* a reliable bioevent which they regarded as the first sign of biotic extinction associated with the late Eocene cooling. This extinction event has long been used to mark the base of nannofossil Zone NP21 (Martini, 1971) or more recently Zone CNE21 (Agnini et al., 2014). On the timescale used by Dunkley Jones et al. (2008) it was estimated to be 500 kyr before the top of the EOIS. However, a subsequent calibration from ODP Site 1218 (Blaj et al., 2009) placed this event significantly earlier than previously suggested, which is supported by recent work in Java (Jones et al., 2019). In the record from Site 1218 the *D. saipanensis* extinction is coincident (within 80 kyr analytical error; Coxall et al., 2005) with the base of a significant $\delta^{18}O$ increase – possibly a 'failed' glaciation – that seems to be visible in many of the records (including Tanzania) and has been termed the 'Late Eocene Event' by Katz et al., (2008). It seems desirable to include these biotic and climatic events within the definition of the EOT rather than insist on an arbitrary 500 kyr duration. On the most commonly used current timescale, 'Geological Timescale 2012' (GTS2012; Gradstein et al., 2012), the critical levels are calibrated as follows: top of the EOIS at 33.65 Ma, base of Chron C13n at 33.705 Ma, EOB at 33.9 Ma, and extinction of *D. saipanensis* at 34.44 Ma. Hence the

stratigraphic interval of the EOT according to our preferred definition is now given an estimated duration of 790 kyr (Figure 1). This terminology and the alternatives are summarized in Table 1 and illustrated below in Figure 1.

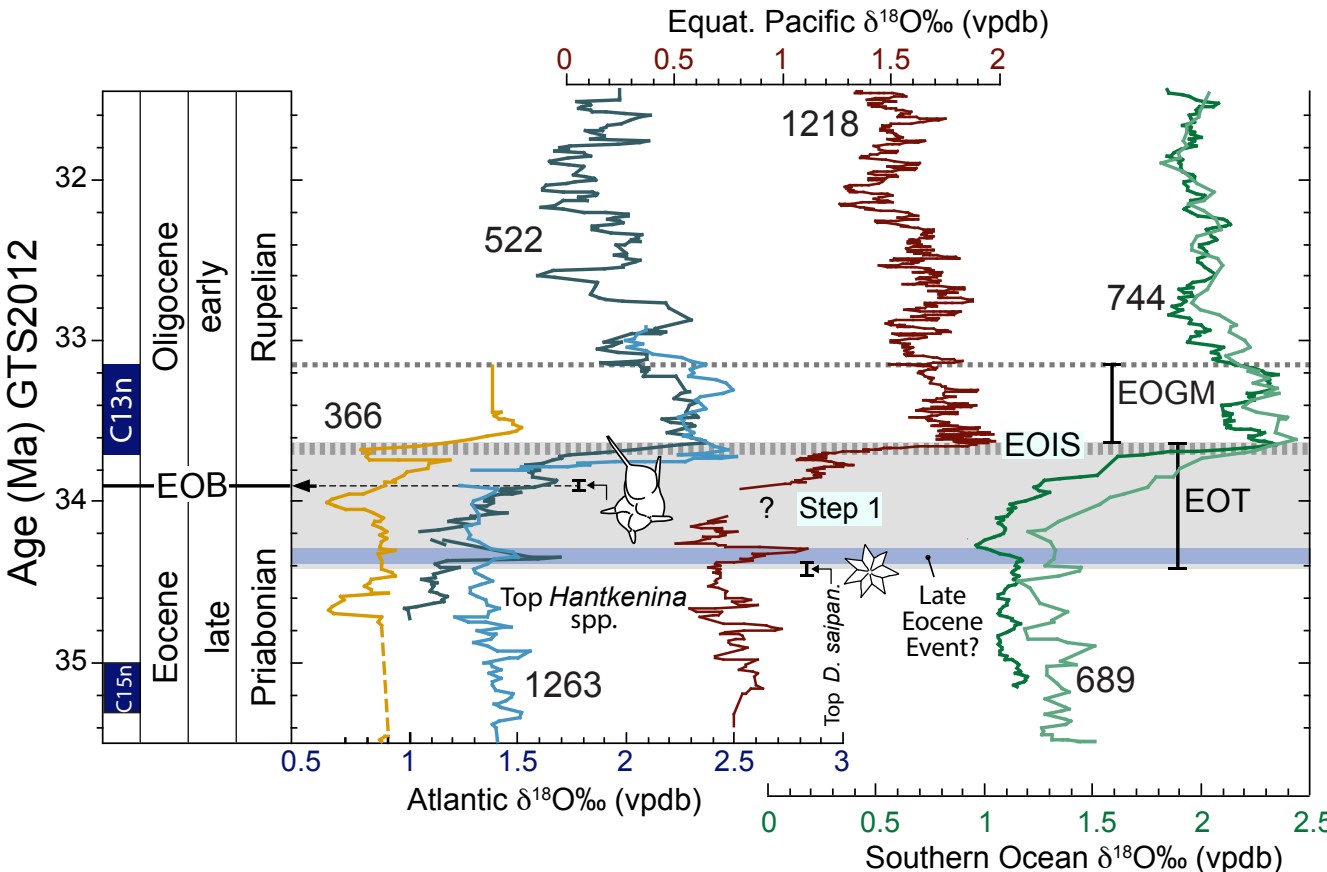

**Figure 1:** Oxygen stable isotope and chrono-stratigraphic characteristics of the Eocene-Oligocene transition (EOT) from deep marine records and EOT terminology on the GTS 2012 time scale. Benthic foraminiferal δ18O from six deep sea drill holes are shown: tropical Atlantic Site 366, South Atlantic Sites 522 and 1263 (Zachos et al. 1996; Langton et al., 2016); Southern Ocean Sites 744 and 689 (Zachos et al. 1996; Diester-Haass and Zahn 1996) and the Equatorial Pacific Site 1218 (Coxall and Wilson 2011). Due to different sample resolutions, running means are applied using a 3-point filter for Sites 522, 689 and 1263, 5-point filter for Sites 366 and 744 and a 7-point filter for Site 1218. Time scale conversions were made by aligning common magneto-stratigraphic tie points. The EOT is defined as a ca. 790 kyr long phase of accelerated climatic and biotic change that began before and ended after the Eocene-Oligocene boundary (EOB) (after Coxall and Pearson (2007)). It is bounded at the base by the 'Top' *D. saipanensis* nannofossil extinction event, and above by the EOIS-δ18O maximum. Benthic data are all *Cibicidoides* spp. or 'Cibs. Equivalent' and have not been adjusted to sea water equilibrium values. 'Step-1' comprises a modest δ18O increase linked to ocean cooling (Lear et al., 2008. Bohaty et al., 2012). The 'Top *Hantkenina* spp.' marker corresponds to the position of this extinction event at DSDP Site 522 (including sampling bracket) with respect to the corresponding Site 522 δ18O curve. That it coincides with the published calibrated age of this event (33.9 Ma) is entirely independent. The 'Late Eocene Event' δ18O maximum (after Katz et al., 2008) may represent a failed glaciation.

Combined δ18O and trace element investigations (see Section 3.2) have led to the suggestion that the δ18O increase commonly referred to as Step-1 (Figure 1) is mostly attributable to ocean cooling, with subordinate ice sheet growth, whereas the more

prominent $\delta^{18}O$ increase at the end of the EOT (i.e., the EOIS in our terminology) largely represents ice growth with a little further cooling (Bohaty et al., 2012; Katz et al., 2008; Lear et al., 2008). Estimates of the combined total sea-level fall across the EOT are on the order of 70 m (Miller et al., 2008; Wilson et al., 2013) and microfacies and paleontological records from shelf environments (Houben et al., 2012) are consistent with this generalization. A recent shallow marine sediment record also indicates the onset of major glaciation at ~33.7 Ma (Gallagher et al., 2020), in agreement with deep sea records. The EOIS is the sharpest feature in most records, culminating with the highest benthic $\delta^{18}O$ values of the Eocene and Oligocene. It is widely suggested that it signifies the initiation of major sustained Antarctic glaciation, most likely an early East Antarctic Ice Sheet (EAIS) (Bohaty et al., 2012; Coxall et al., 2005; Galeotti et al., 2016; Miller et al., 1987; Shackleton and Kennett, 1975; Zachos et al., 1992). The EOGM is interpreted as an approximately 500 kyr-long glacial maximum, with lower values visible in some records Zachos et al., 1996; Liu et al. 2004) (Figure 1). Oxygen isotope maxima in the late Eocene imply substantial ephemeral precursor glaciations in the approach to the EOT (Galeotti et al., 2016; Houben et al., 2012; Katz et al., 2008; Scher et al., 2011, 2014). The oldest and most prominent of these hypothesized transient glacial events occurred at ~37.3 Ma (within magnetochron C17n) and is referred to as the Priabonian Oxygen Isotope Maximum (PrOM) Event (Scher et al., 2014). The 'Late Eocene Event' of Katz et al. (2008) at ~34.15 Ma may be regarded as the second, the third being $\delta^{18}O$ 'Step- 1' at ~34 Ma (the 'precursor glaciation' of Scher et al., 2011). Nevertheless, differences between $\delta^{18}O$ curves from different water depths and ocean regions, combined with increasing detail in individual records afforded by high-resolution sampling emphasizes that the EOT cannot be adequately understood as a series of discrete events because it is clearly imprinted by orbitally paced variability throughout (Coxall et al., 2005).

A detailed discussion of the *Hantkenina* extinction and associated bioevents at the EOB was provided by Berggren et al., (2018, p 30-32). The highest stratigraphic occurrence of the planktonic foraminifera family Hantkeninidae denotes the EOB in its type section (Nocchi et al., 1986). This is thought to have involved simultaneous extinction of all five morphospecies and two genera of late Eocene hantkeninids (Coxall and Pearson, 2007) (Figure 1). Insofar as the principles of biostratigraphy require a particular species to denote a biozone boundary, the commonest species *Hantkenina alabamensis* is used to define the base of Zone O1 (Berggren et al., 2018; Berggren and Pearson, 2005; Wade et al., 2011). The extinction of *H. alabamensis* can be considered the 'primary marker' for worldwide correlation of the EOB. It occurs at a slightly higher (later) level than another set of prominent planktonic foraminifer extinctions, namely *Turborotalia cerroazulensis* and related species. DSDP Site 522 (South Atlantic), thus far, is one of the few deep sea records to have both a detailed $\delta^{18}O$ stratigraphy and planktonic foraminifera assemblages that capture these evolutionary events. Here, the *Hantkenina* extinction horizon occurs approximately two thirds of the way through the EOT (Figure 1). It occurs at a similar relative position in the hemipelagic EOT sequence in Tanzania (Pearson et al., 2008), also in unpublished data from Indian Ocean ODP Site 757 (Coxall et al, unpublished). This finding implies that the extinction of the hantkeninids was approximately synchronous although its cause is currently unknown. Existing constraints on the hantkeninid extinction horizon remain rather coarse in terms of sampling

resolution compared to many isotopic records and the matter will benefit greatly from incoming high-resolution records from Site U1411, which boasts both excellent (glassy) preservation and orbital level sampling.

Dating and correlation of non-marine records, which usually lack $\delta^{18}O$ stratigraphy, is more challenging and there are far fewer well-dated records on land. Here a strict concept of the EOT is difficult to apply and relies on correlations using other stratigraphic approaches, including magnetostratigraphy and palynomorph or mammal tooth biostratigraphy that have been cross calibrated in a few marine and marginal-marine sections (Abels et al., 2011; Dupont-Nivet et al., 2007, 2008; Hooker et al., 2004). Central Asian sections are the exception, where even the step features of the EOT can be identified using magneto, bio- and cyclostratigraphy (Xiao et al., 2010). Moreover, combined $\delta^{18}O$ and clumped isotope analyses on fresh water gastropod shells from a terrestrial EOT section in the south of England has permitted the first direct correlation of marine and non-marine realms and identified coupling between cooling and hydrological changes in the terrestrial realm (Sheldon et al., 2016). This finding suggests a close timing and causal relation between the earliest Oligocene glaciation and a major Eurasian mammalian turnover event called the 'Grande Coupure' (Hooker et al., 2004; Sheldon et al., 2016). During the Grand Coupure, many endemic European mammal species became extinct and were replaced by Asian immigrant species. These changes have been attributed to a combination of climate-driven extinction and species dispersal due to the closing of Turgai Strait; which provided a greater connection between Europe and Asia (Akhmetiev and Beniamovski, 2009; Costa et al., 2011; Hooker et al., 2004).

In shallow water carbonate successions, the EOB has traditionally been approximated by the prominent extinctions of a series of long-ranging larger benthic foraminifers (LBFs) often called orthophragminids (corresponding to the Families Discocylinidae and Asterocyclinidae; Adams et al. 1986). The general expectation was that these extinctions likely occurred at the time of maximum ice growth and sea level regression – in our terminology the EOIS. However evidence from Tanzania (Cotton and Pearson, 2011) and Indonesia (Cotton et al., 2014) suggest that the extinctions occurred within the EOT. In Tanzania the extinctions occur quite precisely at the level of the EOB, hinting that the EOB itself may have had a global cause affecting different environments, possibly independent of the events that caused the isotope increases (Cotton and Pearson, 2011).

The definition of the EOT used here excludes the long-term Eocene cooling trend. That trend began in the Ypresian (early Eocene) and continued through much of the Lutetian and Bartonian (middle Eocene; albeit interrupted by the Middle Eocene Climatic Optimum; MECO; Bohaty and Zachos 2003) and Priabonian (late Eocene) (Cramwinckel et al., 2018; Inglis et al., 2015; Liu et al., 2018; Śliwińska et al., 2019; Zachos et al., 2001). In particular, prominent extinctions in various marine groups occurred around the beginning of the Priabonian (late Eocene), possibly connected with global cooling (e.g. Wade and Pearson 2008; note that the base of the Priabonian has recently been defined in the Alano section in Italy; Agnini et al., 2020). These

data are excluded by our definition from the EOT but may be part of the same general long-term pattern. In some stratigraphic records, especially terrestrial ones, it may not be easy to distinguish these longer-term events from the EOT.

## 2 Proxy evidence for paleogeography, ocean circulation, and terrestrial ice evolution

Here we discuss proxy evidence for the global paleogeography of the EOT (Section 2.1) including the state and evolution of ocean gateways (Section 2.2), and proxy evidence for ocean circulation (Section 2.3) and Antarctic glaciation (Section 2.4). We then briefly discuss the timing of the Northern Hemisphere glaciation (Section 2.5).

### 2.1 Tectonic Reconstruction

The tectonic evolution of the southern continents, opening a pathway for the Antarctic Circumpolar Current, has long been
linked with long-term Eocene cooling and the EOT (Kennett et al., 1975). However, there remain major challenges in reconstructing the paleogeography at or around the EOT, requiring a series of methodological steps (Baatsen et al., 2016; Kennett et al., 1975; Markwick, 2007, 2019; Markwick and Valdes, 2004; Müller et al., 2008). The first step is to use modern geography and relocate the continental and ocean plates according to a plate tectonic evolution model, used in software such as GPlates (Boyden et al., 2011). This software uses the interpretation of seafloor spreading and paleomagnetic data to
reconstruct relative plate motion (e.g. Scotese et al. 1988), and an absolute reference frame to position the plates relative to the Earth's mantle (e.g. Dupont-Nivet et al. 2008). Currently, there are two such absolute reference frames: one based on a global network of volcanic hot-spots (Seton et al., 2012) and one based on a paleomagnetic reference frame (van Hinsbergen et al., 2015; Torsvik et al., 2012). Importantly, these two reference frames give virtually the same continental outlines, but the orientation of the continents is shifted. This results in differences in continental positions between the reference frames of up
to 5-6° (Baatsen et al., 2016) around the EOT, creating an uncertainty in reconstructing paleogeography, especially in southern latitudes between 40 to 70 °S where important land and ocean geological archives exist. This latitudinal uncertainty also impacts the reconstruction of Antarctic glaciation, since glacial dynamics are highly sensitive to latitude.

After the plate tectonic reconstruction has been applied, adjustments are needed to capture the age-depth evolution of the
260 seafloor (Crosby et al., 2006) and seafloor sedimentation rate (Müller et al., 2008). Adjusting land topography is more difficult, and requires knowledge of paleoaltimetry, including processes such as plate collision processes, uplift, subsidence and erosion. Several publicly available reconstructions exist for the Eocene; Markwick (2007) reconstructed paleotopography for the late Eocene (38 Ma), while Sewall et al. (2000) and more recently (Zhang et al., 2011) and Herold et al. (2014) have generated paleotopographies for the early Eocene (~55 to 50 Ma). These are based on the Hot Spot reference frame. Baatsen et al. (2016)
have recently created a paleogeographic reconstruction of the late Eocene using the Paleomagnetic reference frame. Such efforts to develop realistic paleogeography for each time slice represents a major undertaking in blending geomorphic evidence with tectonic evolution, and thus includes many specific details that are beyond the scope of this review.

Recently, a stage-by-stage paleogeographic reconstruction of the entire Phanerozoic has been made publicly available in digital format (Scotese and Wright, 2018). This includes snapshots of the Priabonian (35.9 Ma) and Rupelian (31 Ma). Another stage-by-stage reconstruction of Cenozoic paleogeography evolution originates from Markwick (2007); this reconstruction has been incorporated into a modelling study of climate dependence on paleogeography (Farnsworth et al., 2019; Lunt et al., 2016), and paleogeography changes across the EOT (Kennedy et al., 2015). However, the most recent versions of the (Markwick, 2007) paleogeography reconstructions are proprietary and are thus not included in this paper. Therefore, we present a summary of late Eocene (38 Ma) paleogeography in

Figure 2 from the publicly available datasets of Baatsen et al. (2016) and Scotese and Wright (2018). Our aim here is not to evaluate these reconstructions, but to present them such that their differences can be taken as broadly indicative of the uncertainties in paleogeography at this time.

The reconstructions contain several notable regions of uncertainty which we briefly mention. They include (i) the Tibetan Plateau and the Indian subcontinent, where there are clear disagreements between the reconstructions, (ii) the Turgai strait and Tethys region, which has far greater shallow marine shelf regions in the Baatsen et al. (2016) reconstruction, (iii) the Fram Strait, which is arguably closed by the Eocene-Oligocene transition but is open in both reconstructions (Lasabuda et al., 2018), and (iv) the Rocky Mountains and North American continent exhibit key differences in elevation and coastlines, which has implications for Eocene-Oligocene climate evolution (Chamberlain et al., 2012). A full review of these uncertainties is beyond the scope of this paper, however we briefly discuss some impacts of these paleogeography uncertainties on terrestrial temperature reconstructions in Section 4, while we discuss the impacts on ocean circulation in Section 2.2 and 2.3.

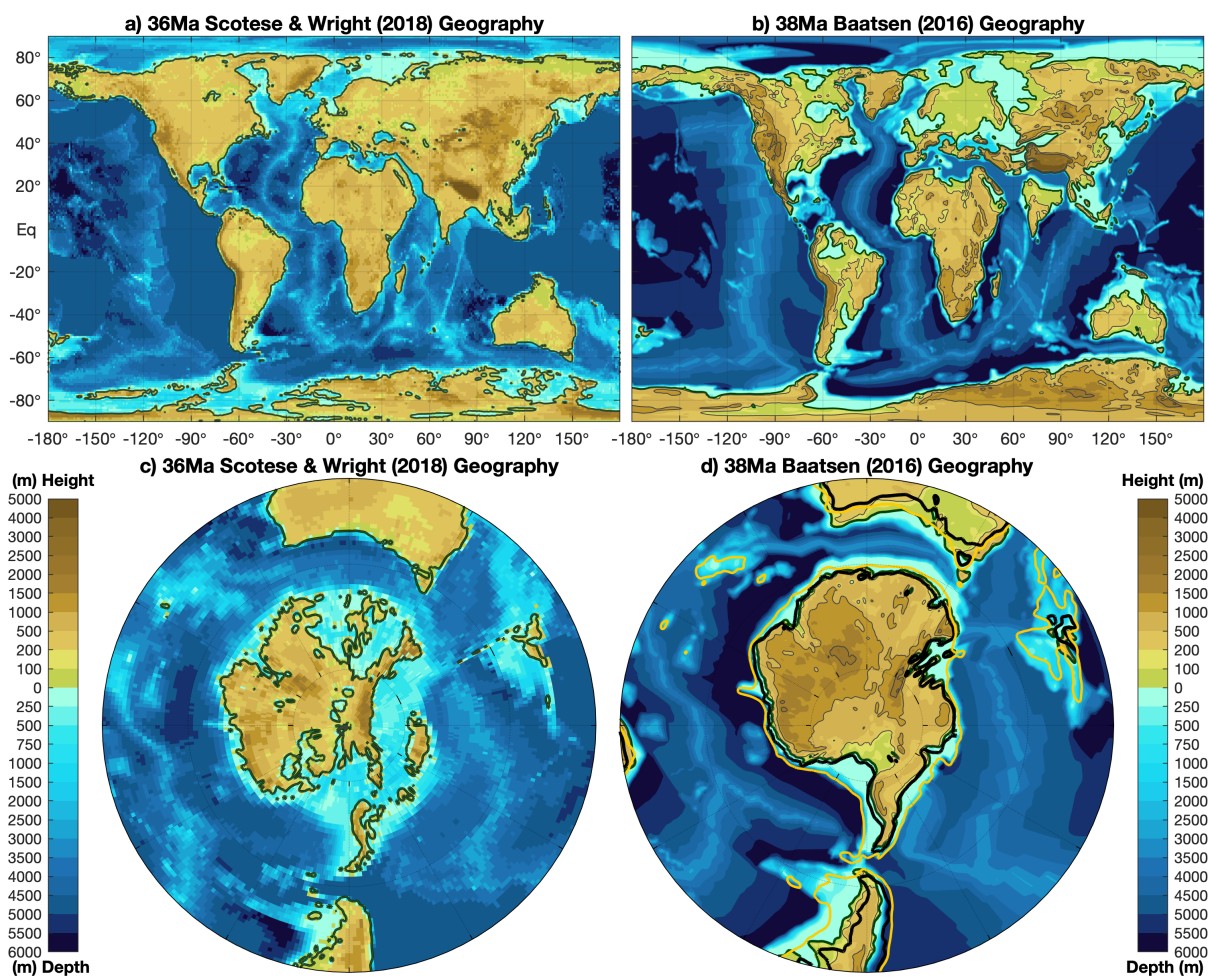

**Figure 2:** Paleogeography at the late Eocene showing two alternative reconstructions. (a) and (c) use the Hot Spot reference frame, using the Scotese and Wright (2018) paleogeography at 36 Ma, while (b) and (d) use the Paleomagnetic reference frame showing the reconstruction of Baatsen et al. (2016) at 38 Ma. These two reconstructions use different methodologies, and are presented as broadly indicative of the uncertainties in paleogeography at this time. Also shown in (d) are post-EOT coastlines at 30 Ma (black contours) and 1000 m depth (orange contours), which illustrate the widening of the Southern Ocean gateways during the 8 Myr interval around the EOT.

## 2.2 Southern Ocean Gateways

A long-held hypothesis on the cause of the EOT glaciation is that Antarctica cooled because of tectonic opening of Southern Ocean gateways (Barker and Burrell, 1977; Kennett, 1977). This mechanism suggests that the onset of the Antarctic Circumpolar Current (ACC) reorganised ocean currents from a configuration of subpolar gyres with strong meridional heat transport to predominantly zonal flow, thereby causing thermal isolation of Antarctica (Barker and Thomas, 2004). The hypothesis is supported by foraminiferal isotopic evidence from deep sea drill cores in the Southern Ocean, which indicate a

shift from warm to cold currents (Exon et al., 2004). As such, there has been considerable effort to reconstruct the tectonic history of the Southern Ocean gateways.


The Drake Passage opening has been dated to around 50 Ma (Livermore et al., 2007) or even earlier (Markwick, 2007); however it was likely shallow and narrow at this time. The timing of the transition to a wide and deep gateway, potentially capable of sustaining a vigorous ACC, occurred on a timescale of tens of millions of years. Even with substantial widening of Drake Passage, several intervening ridges in the region are likely to have blocked the deep circumpolar flow (Eagles et al.,

2005). These barriers may have not have cleared until the Miocene at around 22 Ma (Barker and Thomas, 2004; Dalziel et al., 2013). The evolution of the Tasman Gateway is better constrained. Geophysical reconstructions of continent-ocean boundaries (Williams et al., 2011) place the opening of a deep (greater than ~500 m) Tasmanian Gateway at 33.5 ± 1.5 Ma (Scher et al., 2015; Stickley et al., 2004). Marine microfossil records suggest the circumpolar flow was initially westward (Bijl et al., 2013). Multiproxy based evidence from ODP Leg 189 suggests that the opening of the Tasmanian Gateway significantly preceded

Antarctic glaciation and might therefore not been its primary cause (Huber et al., 2004; Stickley et al., 2004; Wei, 2004). The results also indicate that the gateway deepening at the EOT initially produced an eastward flow of warm surface waters into the southwestern Pacific, and not of cold surface waters as previously assumed. Subsequently, the Tasmanian Gateway steadily opened during the Oligocene, hypothesised to cross a threshold when the northern margin of the ACC aligned with the westerly winds (Scher et al., 2015), triggering the onset of an eastward flowing ACC at around 30 Ma. However, the westerly winds

can also shift position due to changes in orographic barriers or an increase in the meridional temperature gradient after glaciation. Thus, the opening of Southern Ocean gateways approximately coincides with the EOT, but with large uncertainty on the timing and implications. We discuss modelling of this mechanism in Section 6.1.

**2.3 Meridional overturning circulation**

Throughout much of the Eocene, deep water formation is suggested to have occurred dominantly in the Southern Ocean and

the North Pacific (Ferreira et al., 2018), based on numerical modelling and supported by stable and radiogenic isotope work (Cramer et al., 2009; McKinley et al., 2019; Thomas et al., 2014). Compilations of $\delta^{18}O$ and $\delta^{13}C$ throughout the Atlantic basin suggest that the Atlantic meridional overturning circulation (AMOC) either started up or strengthened at the EOT (Borrelli et al., 2014; Coxall et al., 2018; Katz et al., 2011). This view of the AMOC expansion is supported by a decrease in South Atlantic $\epsilon_{Nd}$ around the EOT (Via and Thomas, 2006). Significant seafloor spreading was occurring in the Southern Hemisphere, such

that these changes in ocean circulation have previously been explained by the opening of Southern Ocean gateways (Borrelli et al., 2014). However, studies of deep sea sediment drifts suggest that some kind of North Atlantic overturning operated from the middle Eocene (Boyle et al., 2017; Hohbein et al., 2012). This earlier onset is supported by climate modelling that suggested that AMOC fluctuations in the middle Eocene are linked to obliquity forcing cycles (Vahlenkamp et al., 2018a, 2018b). Moreover, interactions between the Arctic and Atlantic oceans are gaining interest as potential triggers of a late Eocene proto

AMOC (Hutchinson et al., 2019). Data from the Labrador Sea and western North Atlantic margin indicate that North Atlantic

waters became saltier and denser from 37 to 33 Ma (Coxall et al., 2018). This densification may then have strengthened or even triggered an AMOC, suggesting a possible forcing mechanism for Antarctic cooling that predates the EOT and Southern Ocean gateway openings.

Proxy records suggest that the Arctic Ocean was much fresher during the Eocene than the present day, with typical surface salinities around 20-25 psu and periodic excursions to very low salinity conditions (<10 psu) (Brinkhuis et al., 2006; Kim et al., 2014; Waddell and Moore, 2008). Outflow of this fresh surface water into the North Atlantic can potentially prohibit deep water formation (Baatsen et al., 2020; Hutchinson et al., 2018). A new line of evidence suggests that deepening of the Greenland-Scotland Ridge (GSR) around the EOT may have enabled North Atlantic surface waters to become saltier (Abelson

and Erez, 2017; Störz et al., 2017), by allowing a deeper exchange between the basins. A related hypothesis derived from sea level and paleo-shoreline estimates in the Nordic Seas is that the Arctic likely became isolated during the Oligocene (Hegewald and Jokat, 2013; O'Regan et al., 2011). Thus a gradual constriction of the connection between the Arctic and Atlantic presents a newly hypothesised priming mechanism for establishment of a well-developed AMOC (Coxall et al., 2018; Hutchinson et al., 2019).

**2.4 Antarctic glaciation**

Although transient glacial events on Antarctica are proposed for the late Eocene, the most significant long term glaciations likely began on eastern Antarctica on the Gamburtsev Mountains and other highlands (Young et al., 2011) as a result of rapid global cooling in the early Oligocene around 33.7 Ma (EOGM, Figure 1). Evidence for glacial discharge into open ocean basins in the earliest Oligocene is long established, with ice-rafted debris (IRD) appearing in deep-sea Southern Ocean

sediment cores (Zachos et al., 1992). Since these initial results, efforts have continued to document and understand early Cenozoic Antarctic ice dynamics (Barker et al., 2007; Francis et al., 2008; McKay et al., 2016). Combining perspectives from marine geology, geophysics, geochemical proxies and modelling, these efforts have largely focused on the evolution and stability of the early Oligocene Antarctic ice sheets, and estimates of ice volume contributions to sea-level change. Other important developments in the study of Antarctic ice include modelled thresholds for Antarctic glaciation (DeConto et al.,

2008; Gasson et al., 2014), and improved reconstructions of Eocene-Oligocene subglacial bedrock topography (from airborne radar surveys). These bedrock reconstructions are important for reconstructing the nucleation centres of precursor ice sheets (Scher et al. 2011, 2014) and subsequent development of continental-sized ice sheets (Bo et al., 2009; Thomson et al., 2013; Wilson et al., 2013; Wilson and Luyendyk, 2009; Young et al., 2011).

Evidence for glaciation in the Weddell Sea and Ross Sea suggest that there was an increase in physical weathering over West Antarctica around the EOT (Anderson et al., 2011; Ehrmann and Mackensen, 1992; Huang et al., 2014; Olivetti et al., 2015; Scher et al., 2011; Sorlien et al., 2007). However, in the Ross Sea, evidence suggests that an expansion over West Antarctica in Marie Byrd Land occurred after the EOT (Olivetti et al., 2013), while in the Weddell Sea sedimentation rates were still

lower than recent times, suggesting the West Antarctic ice sheet was not expanded to modern proportions (Huang et al., 2014).

This is consistent with approximations of ice volume based upon oxygen isotopes (Bohaty et al., 2012; Lear et al., 2008) and is supported by the record of relatively diverse vegetation around at least coastal regions of Antarctica through the Oligocene (Francis et al., 2008).

Recent evidence has emerged of transient 'precursor' Antarctic glaciations occurred in the late Eocene (Carter et al., 2017;

Escutia et al., 2011; Passchier et al., 2017; Scher et al., 2014), suggesting a 'flickering' transition out of the greenhouse. Importantly, several Southern Ocean sites revealed evidence that Antarctic glaciation induced crustal deformation and gravitational perturbations resulting in local sea level rise close to the young Antarctic ice sheet (Stocchi et al., 2013). Finally, detailed core sedimentary records drilled close to Antarctica in the western Ross Sea invoke a transition from a modestly sized highly dynamic late Eocene-early Oligocene ice sheet, existing from ~34 - 32.8 Ma, to a more stable, continental-scale ice

sheet thereafter, which calved at the coastline (Galeotti et al., 2016).

## 2.5 Northern hemisphere glaciation

While there is clear evidence for Antarctic glaciations at the EOT, the question of contemporaneous northern hemisphere glaciation is contentious. The prevailing view is that the Oligocene represented a non-modern-like state with only Antarctica glaciated (Westerhold et al., 2020; Zachos et al., 2001). Glaciation in mountain areas around the globe is suggested to have

followed through the Miocene and Pliocene, with evidence for the first significant build-up of ice on Greenland (in the southern highlands) traced to the late Miocene, sometime between 7.5 - 6 Ma (Bierman et al., 2016; Larsen et al., 1994; Maslin et al., 1998; Pérez et al., 2018) or as early as 11 Ma (Helland and Holmes, 1997). Northern Hemisphere glaciation intensified during the late Pliocene (~2.7 Ma) when large terrestrial glaciers began rhythmically advancing and retreating across North America, Greenland, and Eurasia (Bailey et al., 2013; Ehlers and Gibbard, 2007; Lunt et al., 2008; Maslin et al., 1998; Raymo, 1994;

De Schepper et al., 2014; Shackleton et al., 1984). It is important to note that a delay in northern hemisphere glaciation relative to Antarctica is predicted by climate models—the stabilizing effect of the hysteresis in the height-mass balance feedback becomes weaker with greater distance from the poles (Pollard and DeConto, 2005), because with decreasing latitude summers become warmer for a given radiative forcing (DeConto et al., 2008).

Nevertheless, a series of studies (Tripati et al., 2005, 2008; Tripati and Darby, 2018) argue that bipolar glaciation was triggered in the Eocene and/or Oligocene. This suggestion is based on two lines of evidence from the sedimentary record: (i) estimates of global seawater $\delta^{18}O$ values (Tripati et al., 2005) and (ii) identification of ice-transported sediment grains inferred to have originated from Greenland in both interior Arctic Ocean and subarctic Atlantic sediment cores associated with the EOT (Eldrett et al., 2007), or earlier (middle Eocene) (St. John, 2008; Tripati and Darby, 2018; Tripati et al., 2008). Certainly, several lines

of evidence provide support for winter sea ice in the Arctic from the middle Eocene (Darby, 2014; St. John, 2008; Stickley et al., 2009) and perennial sea ice from 13 Ma (Krylov et al., 2008). It is possible that small mountain glaciers on East Greenland,

perhaps comparable to the modern Franz Josef and Fox Glaciers of New Zealand (which extend from the Southern Alps through lush rain forest), reached sea level during cooler orbital phases of the Eocene, intensifying in the late Eocene and early Oligocene (Eldrett et al., 2007). Yet, the results of a recent detailed analysis of expanded EOT sections from the North Atlantic's modern day iceberg alley on the Newfoundland margin are inconsistent with the presence of extensive ice sheets on southern and western Greenland and the northeastern Canadian Arctic, contradicting the suggestion of extensive early northern hemisphere glaciation in favour of a unipolar icehouse climate state at the EOT (Spray et al., 2019). Furthermore, it is unlikely that ice growth on land in the Northern Hemisphere was sufficiently extensive to impact global seawater $\delta^{18}O$ budgets sea level at the EOT (Coxall et al., 2005; Lear et al., 2008; Mudelsee et al., 2014). Marine SSTs and floral records from the subarctic and Arctic imply sustained warm temperatures and extensive lowland temperate vegetation well into the middle Miocene (O'Regan et al. 2011), that are not readily reconciled with large continental ice sheets fringing Greenland and other Arctic landmasses then or before this time.

From a theoretical perspective, climate and ice sheet modelling suggest that the $CO_2$ threshold for Northern Hemisphere ice-sheet inception is fundamentally lower than for Antarctica (DeConto et al., 2008; Gasson et al., 2014), implying that the climate must be cooler to glaciate Greenland than Antarctica. This is also consistent with evidence that the modern Greenland ice sheet is highly sensitive to climatic warming and that Greenland may have been almost ice-free for extended periods even in the Pleistocene (Schaefer et al., 2016). This asymmetry between the northern and southern hemispheres in susceptibility to glaciation has been attributed to (i) the lower latitudes of the continents encircling the Arctic Ocean relative to the Antarctic, together with different ocean and atmospheric circulation patterns (DeConto et al., 2008; Gasson et al., 2012) and (ii) the ice sheet carrying capacity of the continents; it has been argued that Greenland topography was low during the Paleogene compared to Antarctica and extensive mountain building, providing high-altitude terrain needed for glaciation, did not occur until the late Miocene-Pliocene (Gasson et al., 2012; Solgaard et al., 2013).

But even in the question of Greenland topography there is uncertainty. Reconstructions of plate kinematics in suspected ice sheet nucleation sites (e.g., northern Greenland, Ellesmere Island) are equivocal. It has been argued that Greenland topography was low during the Paleogene compared to Antarctica and extensive mountain building, providing high-altitude terrain needed for glaciation, did not occur until the late Miocene-Pliocene (Gasson et al., 2012; Japsen et al., 2006; Solgaard et al., 2013). Recent work on the plate kinematic history of the Eurekan orogeny, taking into account crustal shortening (Gurnis et al., 2018), indicate a period of significant compression in northern Greenland and Ellesmere from 55-35 Ma (Gion et al., 2017) that was probably associated with uplift (Piepjohn et al., 2016). These latest tectonic insights are compatible with insights from apatite fission track and helium data that support the onset of a rapid phase of exhumation of the East Greenland margin around 30 +/- 5 Ma (Bernard et al., 2016; Japsen et al., 2015). Together, these approaches support a view of high mountains on Greenland/Ellesmere that began eroding in the late Eocene to early Oligocene with a greater possibility of supporting glaciers.

## 3 Marine observations

### 3.1 Sea surface temperature observations

A key requirement for understanding the cause and consequences of the Eocene-Oligocene climatic transition is good spatial and temporal constraints on global temperatures and our most numerous and well-resolved records of this undoubtedly come from the oceans. Quantitative reconstruction of sea surface and deep ocean temperatures has been ongoing for decades. This requires use of various geochemical proxies, both to provide independent support for absolute temperature estimates and because different proxy options are available for different ocean/sedimentary settings, and deep sea versus surface ocean water masses. Each method has its own limitations and uncertainties resulting in a currently patchy but steadily improving view of global change. Quantitative assemblage-based SST proxies akin to transfer functions are not available because there are no living plankton relatives during the EOT. For a thorough review of pre-Quaternary marine SST proxies, and their strengths and weaknesses, see Hollis et al., (2019).

While more heterogeneous than the deep sea, reconstruction of SST in the EOT is in some ways currently more achievable than bottom water temperatures because more proxies are available, although there are still multiple confounding factors to consider. Classical marine $\delta^{18}O$ paleothermometry extracted from the calcium carbonate shells of fossil planktonic (surface floating) foraminifera is especially complicated because of the combining influences of (i) compromised fossil preservation under the shallow late Eocene ocean calcite compensation depth, limiting the availability of planktonic records, (ii) increasing $\delta^{18}O$ of sea water as a consequence of ice sheet expansion, which enriches ocean water and thus increases calcite $\delta^{18}O$; a signal which can otherwise indicate cooling. However, a growing number of clay-rich hemipelagic marine sequences containing exceptionally well-preserved (glassy) fossil material are yielding $\delta^{18}O$ paleotemperatures that provide useful SST perspectives. $\delta^{18}O$ SSTs derived from glassy foraminifera (Haiblen et al., 2019; Norris and Wilson, 1998; Pearson et al., 2001; Wilson et al., 2002; Wilson and Norris, 2001) contrast greatly from those measured on recrystalized 'frosty' material (Sexton et al., 2006). A detailed compilation of glassy versus recrystalized foraminiferal $\delta^{18}O$ proxies around the EOT is given in Piga (2020).

Planktonic foraminifera Mg/Ca paleothermometry provides another means of quantifying SSTs (Evans et al., 2016; Lear et al., 2008), however such records are even more sparse than $\delta^{18}O$ equivalents due to the scarcity of appropriate EOT fossils. This method is especially useful since, in theory, unlike $\delta^{18}O$ it should be independent of Antarctic glaciation and, when coupled with $\delta^{18}O$ paleothermometry, past variations in the $\delta^{18}O$ composition of seawater, and thus ice volume changes may also be estimated (Lear et al., 2004, 2008; Mudelsee et al., 2014). The two key existing records are from Tanzania (Lear et al., 2008) and the Gulf of Mexico (Evans et al., 2016; Wade et al., 2012). The Tanzanian planktonic Mg/Ca record provides cornerstone evidence for a permanent 2.5 °C tropical surface and bottom water, ocean cooling, and therefore likely global cooling, associated with the Step 1 of the EOT (Figure 1). The Gulf of Mexico Mg/Ca temperature record resembles the

biomarker-derived (i.e. TEX$_{86}$, see below) SST record from this site (Wade et al., 2012). Both imply a distinct and slightly larger surface cooling of 3-4°C limited to Step 1. To what extent secular change in seawater Mg/Ca reconstruction might have influenced these actual numbers is an ongoing question (Evans et al., 2018). Clumped isotope paleothermometry (Ghosh et al., 2006; Zaarur et al., 2013), also independent of seawater $\delta^{18}$O, is still in its infancy, but this is a third method applicable to calcareous microfossils that will help address some of these problems. Thus far only one clumped isotope ($\Delta_{47}$) record from Maud Rise spans the EOT (Petersen and Schrag, 2015). This record shows cooling preceding the EOT, and then relatively minor changes across the EOT. Early to middle Eocene clumped isotope SST records are consistent with other proxies, specifically cooler values at high southern latitudes compared to the early and middle Eocene (Evans et al., 2018). Many new SST records base on Mg/Ca and $\Delta_{47}$ are expected in coming years.

In some regions, Eocene-Oligocene age sediments lack biogenic calcium carbonates (e.g. Bijl et al., 2009). Therefore low- and non-calcareous areas, like the Arctic and high-latitudes of the North Atlantic and North Pacific, have suffered for lack of paleotemperature data. However, the development of independent organic proxies based on biomarkers such as alkenones ($U^{K'}_{37}$ index; Brassell et al. 1986) and glycerol dialkyl glycerol tetraethers (GDGTs) from the membrane lipids of Thaumarchaeota (TEX$_{86}$ index; Schouten et al. 2002), which can be preserved in high sedimentation settings close to continental margins or restricted basins where carbonate is often scarce, have helped fill this gap. Importantly, these organic biomarkers are often the only marine archive for paleothermometry at high latitudes, where SST constraints are particularly useful for model-data comparisons.

While the $U^{K'}_{37}$ index is well established, the TEX$_{86}$ index is relatively new and its accuracy as a paleotemperature proxy is under critical review. There have been several different TEX$_{86}$ indices developed, with different SST calibrations (e.g. TEX'$_{86}$ by Sluijs et al. 2009; TEX$_{86}^{H}$ and TEX$_{86}^{L}$ by Kim et al. 2010; Bayspar by Tierney and Tingley 2015). As suggested by some of the recent studies conducted on cultures of Thaumarchaeota, GDGT composition may be sensitive not only to SST but also other factors such as oxygen ($O_2$) concentration (Qin et al., 2015) or ammonia oxidation rate (Hurley et al., 2016). Furthermore, there is uncertainty in the source of the GDGTs used for SST estimations, i.e. their production level in the water column and possible summer biases, and therefore their value as an SST proxy. Recent reviews are available for both the paleotemperatures $U^{K'}_{37}$ (Brassell, 2014) and for TEX$_{86}$ (Hurley et al., 2016; Pearson and Ingalls, 2013; Qin et al., 2015; Tierney and Tingley, 2015). Despite these issues, in some studies where both $U^{K'}_{37}$ and TEX$_{86}$ indices were applied, temperature estimations show remarkably similar results (Liu et al., 2009) suggesting that TEX$_{86}$, after an evaluation of the source and the distribution of GDGTs (Inglis et al., 2015), can successfully be applied as a paleotemperature proxy. TEX$_{86}$ is especially useful at lower latitudes, since the $U^{K'}_{37}$ index saturates at about 29°C (Müller et al., 1998).

Cross-latitude biomarker proxy records ($U^{K'}_{37}$ and TEX$_{86}$) suggest that SSTs were higher than today in both the late Eocene and early Oligocene SSTs with annual means of up to 20 °C at both 60 °N and 60 °S respectively and low meridional

temperature gradients (Hollis et al., 2009; Liu et al., 2009; Wade et al., 2012). One record from the Gulf of Mexico (Wade et al., 2012) suggests consistently higher SSTs derived from $TEX_{86}$ than from inorganic proxies (Hollis et al., 2009, 2012; Liu et al., 2009). Where records span the EOT (i.e. ~33-34 Ma), between 1 to 5 °C of surface cooling in both hemispheres is found. To date, temperature records from the high northern latitudes are sparse but coverage from the high southern latitudes is richer where several records suggest a cooling of subantarctic waters across the EOT of 4 to 8°C, although some records are indistinguishable from 0°C change (Figure 3). In the low latitude Pacific, Atlantic and Indian Ocean tropical SSTs were significantly warmer than today in the late Eocene, with SSTs up to 31°C (Liu et al., 2009) or even ~33°C (Lear et al., 2008; Wade et al., 2012). One $TEX_{86}$ record from the Gulf of Mexico implies gradual surface cooling of 3-4 °C between ~34 and 33 Ma (Wade et al., 2012). $TEX_{86}$ data from Site 803 in the tropical Pacific shows a large transient cooling of up to 6°C across the EOT, however, such a large change in tropical temperatures is regarded as unrealistic and is more likely caused by a reorganization of the water column (Liu et al., 2009). We therefore do not include Site 803 in our compilation of temperature change across the EOT (Figure 3).

Newly available records from the North Atlantic region are starting to challenge earlier evidence of homogeneous bipolar cooling (Liu et al., 2018; Śliwińska et al., 2019). Furthermore, comparison of a uniquely well-resolved record from the Newfoundland margin, western North Atlantic (Liu et al., 2018), with data from the subantarctic South Atlantic, confirms this in new detail, leading the authors to the conclusion that surface ocean cooling during the EOT was strongly asymmetric between hemispheres. Liu et al. (2018) interpret this finding as evidence for 'transient thermal decoupling of the North Atlantic Ocean from the southern high latitudes', as a result of changes in ocean circulation-driven heat transport associated with Antarctic glaciation. Recent $TEX_{86}$ data spanning the Oligocene suggests that the low meridional gradient similar to the late Eocene persists well after the EOT, and that warming occurs in the late Oligocene despite an apparent decrease in $CO_2$ (O'Brien et al., 2020).

Here we present a new compilation of SST change across the EOT (Figure 3; Supplementary Table S1). For this compilation, we define two windows of time-averaging: one for the late Eocene (38 to 33.9 Ma) and one for the early Oligocene (33.9 to 30 Ma), with the change across the EOT defined as the difference between the two windows. The compilation includes only SST proxy records that record a signal in both the late Eocene and early Oligocene. We chose these broad averages in order to incorporate data from as wide a geographical region as possible, and to apply a consistent methodology to both SST and terrestrial temperature change. A consequence of this choice is that the averaging may dampen the peak-to-peak signal of EOT SST change in high-resolution records, or increase uncertainty in certain records. However, by choosing longer windows, our averaging method provides a clear picture of the lasting climate change from the late Eocene to the early Oligocene. A summary of SST records across the EOT is shown in Figure 3. The data are plotted against their paleolatitude at 34 Ma, derived using the paleomagnetic reference frame of Torsvik et al., (2012) and van Hinsbergen et al., (2015).

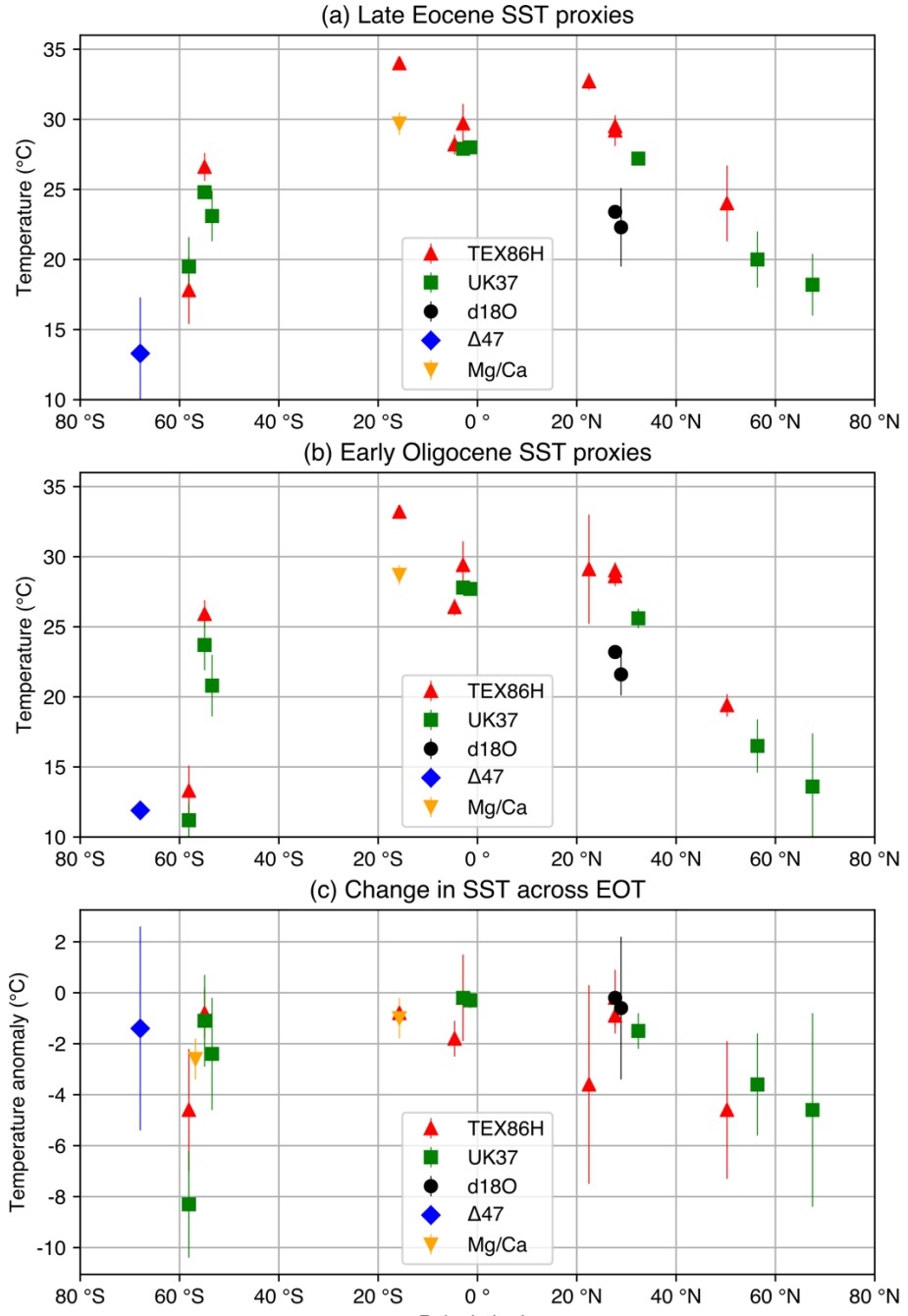


**Figure 3:** Summary of sea surface temperature (SST) change across the EOT from proxies, $TEX_{86}^H$, $U^{K'}_{37}$, $\delta^{18}O$, $\Delta_{47}$, and Mg/Ca. (a) Late Eocene, (b) early Oligocene and (c) change in SST across the EOT. Data shown in (a) and (b) are only from locations that record a temperature signal on both sides of the EOT. Data compiled from (Bohaty et al., 2012; Cramwinckel et al., 2018; Inglis et al., 2015; Kobashi et al., 2004; Lear et al., 2008; Liu et al., 2009, 2018; Pearson et al., 2007; Petersen and Schrag, 2015; Piga, 2020; Śliwińska et al., 2019; Wade et al., 2012; Zhang et al., 2013). The late Eocene value was calculated as an average between 38 and 33.9 Ma (pre-EOT), while the early Oligocene value was calculated as the average between 33.9 and 30 Ma (post-EOT), and the change across the EOT is the difference between these values. The data compilation is provided in digital form in Supplementary Table S1.


### 3.2 Deep sea temperature changes


As described in Section 1.2, the Eocene-Oligocene climate transition is defined by high-resolution benthic foraminiferal oxygen isotope ($\delta^{18}$O) records from deep sea sites (Coxall et al., 2005; Zachos et al., 1996). These records describe a benthic $\delta^{18}$O increase of about 1.5‰, a combination of deep sea cooling and terrestrial ice growth. While surface ocean temperature changes have been constrained using organic and inorganic proxies (Section 3.1), there are fewer proxies for deep sea

temperature and, thus, the picture of deep ocean cooling remains uncertain. This is because, on its own, it is impossible to deconvolve the temperature and ice volume components of $\delta^{18}$O records, and hence quantify the timing, magnitude and spatial distribution of deep ocean temperature change through the climate transition. Indeed, an early interpretation of the Cenozoic benthic oxygen isotope record suggested that the $\delta^{18}$O increase at the EOT represented a pure cooling signal (Shackleton and Kennett, 1975), whereas numerous lines of evidence have since shown that a substantial component of the $\delta^{18}$O shift reflects

the glaciation of Antarctica (e.g. Zachos et al. 1996). Independent paleotemperature proxies provide a potential means to deconvolve the two contributors to $\delta^{18}$O records, and benthic foraminiferal Mg/Ca paleothermometry has been applied to several marine EOT sections (Billups and Schrag, 2003; Bohaty et al., 2012; Katz et al., 2008; Lear et al., 2000, 2004, 2008, 2010; Peck et al., 2010; Pusz et al., 2011; Wade et al., 2012). Yet, calculating absolute bottom water temperatures from benthic foraminiferal Mg/Ca ratios requires an estimate of the Mg/Ca ratio of seawater, while Mg-partitioning into foraminiferal calcite

shows modest sensitivity to temperature at low temperatures and is subject to the competing influence of seawater carbonate chemistry (Evans et al., 2018; Lear et al., 2015). Relative temperature trends over short time intervals are generally considered more robust than absolute temperatures, although the residence time of calcium in seawater (~ 1 Myr; Broecker and Peng 1982) compared with the duration of the entire climate transition (~500 kyr) adds some uncertainty to calculated relative temperature changes across the EOT. High resolution reconstructions of seawater Mg/Ca are therefore required to improve

both absolute and relative temperature changes using Mg/Ca paleothermometry.

Furthermore, although the benthic foraminiferal Mg/Ca paleothermometer appears to capture the long-term cooling trend since the early Eocene Climatic Optimum, the concomitant ~1km deepening of the calcite compensation depth (CCD) hinders its use across the EOT (Coxall et al., 2005; Lear et al., 2004). Specifically, the increase in bottom water calcite saturation state across the EOT acts to increase benthic foraminiferal Mg/Ca, and mask the deep sea cooling signal (Coxall et al., 2005; Lear

et al., 2004). Attempts have been made to use Li/Ca to correct this $\Delta CO_3^{2-}$ effect from Mg/Ca records (Lear et al., 2010; Peck et al., 2010; Pusz et al., 2011), but this approach brings with it additional uncertainties including the species-specific sensitivities to both temperature and $\Delta CO_3^{2-}$. An alternative, and perhaps more robust approach at present, is to combine planktonic $\delta^{18}$O records with salinity-independent sea surface paleotemperature records to calculate the change in the surface

water $\delta^{18}O$ ($\delta^{18}O_{sw}$). The overall change in surface $\delta^{18}O_{sw}$ across the EOT has been estimated using planktonic $\delta^{18}O$ and

Mg/Ca at many sites, including a section in Tanzania containing exceptionally well-preserved (glassy) foraminifera (Lear et

al., 2008). The similarity between this $\Delta\delta^{18}O_{sw}$ estimate from the Indian Ocean (~0.6‰; Lear et al. 2008) and those from

other sites, e.g., the Southern Ocean (Bohaty et al., 2012) and the southeast Atlantic (Peck et al., 2010) suggests that the surface

$\delta^{18}O_{sw}$ change is dominated by a global (ice volume) signal. If we can assume that the surface $\delta^{18}O_{sw}$ signal is dominated

by the ice volume signal (as opposed to a local change in the salinity) then these records can be used in conjunction with the

benthic $\delta^{18}O$ records to estimate changes in bottom water temperature across the climate transition (Kennedy et al., 2015). As

noted above, inter-basin similarities suggest this assumption holds true in Indian Ocean, Southern Ocean and southeast Atlantic

sites, whereas sites in the Pacific and North Atlantic are not as clearly constrained. The associated estimated volume of

Antarctic ice depends upon the assumed isotopic composition of the ice sheet, but was likely between 70 and 110% of the size

of the modern day Antarctic ice sheet (Bohaty et al., 2012; Lear et al., 2008), representing a sea level difference to modern of

approximately minus 18 metres to plus 6 metres. Spatial heterogeneities in the deep ocean temperature history may therefore

be inferred by calculating inter-site offsets in benthic foraminiferal $\delta^{18}O$ records (Abelson and Erez, 2017; Bohaty et al., 2012;

Cramer et al., 2009).

There is a growing consensus that Step 1 of the EOT was associated with a cooling of both deep waters and low latitude surface

waters on the order of 2 °C, while the increase in global ice volume was relatively minor (Bohaty et al., 2012; Lear et al., 2004,

2008, 2010; Peck et al., 2010; Pusz et al., 2011). We note that the combination of this magnitude of cooling and an overall

increase in $\delta^{18}O_{sw}$ of ~0.6‰ is enough to account for the average ~1.0‰ shift in benthic foraminiferal $\delta^{18}O$ observed in deep

sea records (Mudelsee et al., 2014). However, this overall shift across the entire climate transition ignores the apparent $\delta^{18}O$

"overshoot" (Zachos et al., 1996) observed in some high resolution records, at the base of the EOGM (Coxall and Pearson,

2007). Determining whether the "overshoot" reflects deep sea cooling, a transient further increase in global ice volume, or a

combination of the two has implications for our understanding of Antarctic ice sheet dynamics and indeed the cause of the

EOT itself. Unfortunately, it is Step 2 (EOIS) of the transition, into the EOGM where the CCD reaches its maximum depth

and benthic foraminiferal Mg/Ca records appear most compromised by the $\Delta CO_3^{2-}$ effect (Lear et al., 2004, 2010), even at

depths above the implied depth of CCD deepening (Peck et al., 2010), so we currently have no robust and direct evidence of

deep ocean cooling across this step. Future work may go some way to address these problems using clumped isotopes or by

generating high resolution B/Ca records across the transition, and by using deep infaunal benthic species (e.g. Elderfield et al.

2012). However, by combining benthic and planktonic records, it appears that the EOGM in the deep Pacific Ocean reflects,

at least in part, a transient cooling of deep waters associated with the major expansion of the Antarctic ice sheet (Kennedy et

al., 2015).

An additional complication is that the Mg/Ca composition of seawater may itself have shifted during the EOT, as suggested by incoming constraints from other proxies, including paired Mg/Ca and clumped isotope temperature constraints in shallow living larger benthic foraminifera (Evans et al., 2018). Further investigation into this possibility is required, which could ultimately help identify Mg/Ca adjustment factors needed to improve the ability to extract paleotemperature estimates for this geological time interval.

## 4 The terrestrial realm at the EOT

There are several proxy indicators of past terrestrial climate change. These include geochemical indices, leaf margin analysis, Climate Leaf Analysis Multivariate Program (CLAMP) (Yang et al., 2011), and pollen assemblages (see review in Hollis et al. 2019). Here we focus on pollen assemblages as a broad indicator of terrestrial change across the EOT, because an EOT synthesis of these data already exists (Pound and Salzmann, 2017). This dataset, a global paleo-biome reconstruction of pollen and spore assemblages, indicates that the terrestrial realm of the late Eocene and early Oligocene has a vegetation distribution that in general indicates a warmer and wetter world than today. The response of the terrestrial realm to the EOT is more heterogeneous than the marine realm, and biome changes do not record a uniform global response (Pound and Salzmann, 2017). Terrestrial biomes record not only global climate change but also regional changes due to local factors. These include orographic uplift, which reduces local temperature and changes regional precipitation patterns. Further changes in precipitation are induced by the retreat of a number of inland seaways due to sea-level changes and tectonics (Chamberlain et al., 2012; Dupont-Nivet et al., 2008; Kocsis et al., 2014; Sheldon et al., 2016). These complicating factors mean that changes in vegetation must be interpreted within the context of local paleoenvironmental changes. However, there are some emerging terrestrial records that record a significant temperature drop and perturbation of the hydrological cycle, consistent with global cooling. Thus, we present the terrestrial records on a continent-by-continent basis below, with a summary of temperature change across the EOT shown in Figure 4. As for the marine data, we derive a temperature change across the EOT by taking the difference between a late Eocene window (38 to 33.9 Ma) and an early Oligocene window (33.9 to 30 Ma). The data in Figure 4 are plotted against paleolatitude at 34 Ma, using the paleomagnetic reference frame of Torsvik et al., (2012) and van Hinsbergen et al., (2015). For a summary of strengths and limitations of deriving quantitative climate estimates from the pre-Quaternary plant record, see Hollis et al. (2019).

### 4.1 North America

In North America, the paleobiome distribution of the EOT ranges from tropical mangroves, swamps and forests in the south of the continent to cool-temperature forests at the high latitudes (Breedlovestrout et al., 2013; Pound and Salzmann, 2017; Wolfe, 1985, 1994). Gradual cooling and drying from the middle Eocene until the late Oligocene allowed the mixed coniferous and deciduous broadleaf forests to become more dominant (Wing, 1987). Fossil leaves found in Washington State (Breedlovestrout et al., 2013) indicate no clear temperature trend from the middle Eocene to the EOT. Instead variations are

attributed to differing paleoaltitude, combined with a gradual long-term cooling. Pollen records from Texas indicate a long term cooling and aridification from the middle Eocene to the early Oligocene (Yancey et al., 2003), whereas pollen records from 5° longitude further east show no turnover at the EOT boundary (Oboh-Ikuenobe and Jaramillo, 2003). Pollen from the far north Yukon Territory show a transition from warmer adapted angiosperm forests in the Late Eocene to cooler adapted gymnosperm forests during the Early Oligocene (Ridgway and Sweet, 1995).

In Oregon, well dated floras and marine invertebrates show no evidence for a rapid change at the EOT (Retallack et al., 2004), but rather a gradual cooling during the early to middle Oligocene. By contrast, paleosols indicate a 2.8±2.1°C drop across the EOT in the same region (Gallagher and Sheldon, 2013). Isotopic data from horse teeth indicate a 8±3.1°C drop in mean annual temperature (MAT) across the EOT, but with a 400 kyr lag behind the marine realm (Zanazzi et al., 2007), though part of this shift is due to changes in the hydrological cycle (Chamberlain et al., 2012; Hren et al., 2013). Moreover, a clumped isotope study of Fan et al., (2017) records a decrease of ~7°C across the EOT in the north central USA, similar to the findings of Zanazzi et al. (2007). Conversely a study on White River mammals interprets no significant change in MAT, but an aridification of the local environment (Boardman and Secord, 2013). This conclusion is in line with paleosol studies, which suggest a change in vegetation structure from a forest to a more open environment (Retallack, 1983).

Oxygen isotope analyses in the western North American Cordillera suggest the changing hydrological regime of North America (Chamberlain et al., 2012) was influenced by factors other than large-scale climate. Rising orography, starting in British Colombia at ~50 Ma and moving south to Nevada by ~23 Ma shifted the North American Monsoon further south during this time. This not only impacts terrestrial oxygen isotopes, but also the regional vegetation – creating an aridification not linked to global climatic events (Chamberlain et al., 2012). There is a significant increase in dust deposition in the foothills of the North American cordillera (Fan et al., 2020), suggesting cooling and aridification. No response to the EOT is evident from North American mammals (Figueirido et al., 2012; Prothero, 2012, 2004), while fossil Equidae analyses from North America indicate that horses had a browsing diet before, at and after the EOT (Mihlbachler et al., 2011). One study argues that North American mammals had already adapted to Oligocene-like cold and arid conditions prior to the EOT (Eronen et al., 2015), suggesting that any further environmental change at the EOT would not be expressed in changes to these species.

## 4.2 South America

Late Eocene and early Oligocene paleo-biome distributions of South America indicate tropical evergreen rainforest in the north and cool-temperate biomes in the south (Pound and Salzmann, 2017). In South America there was a greater change in vegetation from the middle Eocene into the late Eocene, rather than at the EOT (Barreda and Palazzesi, 2007). Patagonian pollen floras from the middle Eocene to the end of the early Oligocene are termed the "Mixed Paleoflora". These show a long-term cooling trend rather than a step change at the EOT (Quattrocchio et al., 2013). Phytolith and oxygen isotope records from

Patagonia show no change in vegetation across the EOT (Kohn et al., 2004, 2015; Strömberg et al., 2013). However, this view has recently been challenged by a higher stratigraphic resolution study of phytoliths and magnetic properties, pointing to a clearer ecosystem change (Selkin et al., 2015). A recent stable isotope hydrology study from Patagonia indicates rapid cooling during the EOT (Colwyn and Hren, 2019). Faunal turnovers in South America began at approximately 42 – 39 Ma (Woodburne et al., 2014). This relates to the end of the MECO, but also correlates with the appearance of rodents from Africa. The mammal turnover associated with the EOT is no more dramatic than those during the late Eocene or late Oligocene (Woodburne et al., 2014). The Amazonian region had a diverse, primarily frugivorous fauna during the EOT, suggesting productive stable forest (Negri et al., 2009). To summarise, there are some indications of significant cooling in South America at the EOT, but overall the signal is mixed, with both faunal and plant-based proxies suggesting a heterogeneous response.

### 4.3 Africa

Vegetation in Africa shows little change in structure from the Late Eocene into the Early Oligocene, but there is a documented drop in palm diversity (Jacobs et al., 2010; Pan et al., 2006; Pound and Salzmann, 2017). There are significant gaps in the paleobotanical record for Africa over this time interval, with most information coming from the region between 10° north and south of the Equator (Jacobs et al., 2010; Pound and Salzmann, 2017). One exception is the Fayum Depression in Egypt which contains macrofossil and microfossil evidence for tropical vegetation in the late Eocene (Tiffney and Wing, 1991; Wing et al., 1995).

### 4.4 Eurasia

In Eurasia there was a progressive change from para-tropical evergreen forests in the middle Eocene to warm-temperate evergreen and deciduous mixed forests by the early Oligocene (Collinson and Hooker, 2003; Teodoridis and Kvaček, 2015). The paleo-biome reconstructions show a dominance of subtropical and warm-temperate mixed forests throughout Eurasia with seasonal biomes in the Iberian Peninsula and arid biomes in central Asia (Pound and Salzmann, 2017). A change from a diverse mixed broad-leaved to a cooler conifer- dominated pollen flora in North Atlantic cores through the Eocene indicates increasing seasonality in Europe (Eldrett et al., 2009). However, leaf floras from Bulgaria show no significant change in vegetation at the EOT (Bozukov et al., 2009). There is a greater change in Iberian pollen floras from the early to late Oligocene than at the EOT (Postigo Mijarra et al., 2009). Between the late Eocene and early Oligocene no change in MAT or precipitation is reconstructed in the Ebro Basin in Spain, but there is a decrease in chemical weathering across the EOT (Sheldon et al., 2012).

In Germany and Czechia, macrofloras show a stepwise disappearance of subtropical species and immigration of evergreen and deciduous warm-temperate species during the late Eocene (Kunzmann et al., 2016). The first mixed evergreen/deciduous forest in azonal biomes is recorded prior to the EOT from Roudniky (35.4 ± 0.9 Ma; Kvaček et al., 2014) referring to a latest Eocene cooling event (Teodoridis and Kvaček, 2015). However, evergreen broadleaved forests were still present in the early Oligocene (Kovar-Eder, 2016; Teodoridis and Kvaček, 2015) indicating the low impact of global EOT changes in terrestrial central

Europe. Most of the subtropical to warm-temperate genera survived in that region until the Miocene Climatic Optimum (Mai, 1995). Based on proxies from macrofloras, MAT was almost stable at the EOT (Teodoridis and Kvaček, 2015) with ongoing

prevailing seasonality in precipitation and a curtailment of the growing season (Moraweck et al., 2019). While cold month mean temperatures (CMMT) in the Priabonian mostly exceed 10°C, the lower limit for the growing season, the earliest Oligocene floras from Schleenhain and Haselbach (Germany) indicate CMMT below 10°C and a growing season length of 9-11 months (Moraweck et al. 2019).

Recent investigations on late Eocene and earliest Oligocene macrofloras in SE Tibet and Yunnan revealed multiple lines of evidence for the modernization of the vegetation by establishment of present-day genera and families (Linnemann et al., 2017; Su et al., 2018). Regional vegetation change across EOT from subtropical to temperate and partly cool temperate in SW China has been argued to be influenced by the uplift of the Tibetan Plateau (Su et al., 2018). An Eocene appearance of a modern subtropical / tropical aspect of vegetation is also recorded from Chinese low latitude floras (Hainan; Guangdong) indicating

an Eocene establishment of monsoonal climate linked to Tibetan uplift (Jin et al., 2017). However, the evolution of the Tibetan Plateau at the EOT is currently under debate. Earlier studies suggested that a proto-Tibetan highland of more than 4000 m elevation existed in the late Eocene based on stable isotope paleoaltimetry (Cyr et al., 2005; Quade et al., 2011; Rowley and Currie, 2006). New data-model comparisons have cast doubt on these estimates, finding that the stable isotope paleoaltimetry is influenced by different atmospheric circulation patterns than previously thought (Botsyun et al., 2019; Quade et al., 2020).

These studies suggest a lower paleoaltimetry of the Tibetan Plateau (less than 3000 m) in the late Eocene (Botsyun et al., 2019; Quade et al., 2020).

Aside from paleoaltimetry, the timing of environmental changes suggests that climate change at the EOT had a distinct impact on Tibetan environments (Dupont-Nivet et al., 2007). Northeastern Tibet (Xining Basin) shows significant changes at the EOT

in the depositional environments (Dupont-Nivet et al., 2008), pollen and clumped isotopic temperatures (Hoorn et al., 2012; Page et al., 2019) and accumulation rates (Abels et al., 2011). Furthermore, temperature changes in the Xining Basin are too sudden to be driven by changes in basin altitude (Page et al., 2019). The timing of the large temperature drop suggests a coeval decrease in regional temperature linked to EOT glaciation and monsoonal rainfall (Page et al., 2019). Mongolian and northwestern Chinese faunal records indicate a large mammal turnover at the EOT: the "Mongolian Remodelling" (Kraatz and

Geisler, 2010; Meng and McKenna, 1998; Sun et al., 2014), synchronous with the Grand Coupure in Europe. Significant depositional environment change in southwestern Mongolia is also shown by (Sun and Windley, 2015).

Freshwater gastropods from southern Britain show that growing season temperatures (Spring – Summer) may have dropped from around 34°C to about 20°C across the Eocene – Oligocene boundary (Hren et al., 2013). This has been translated into a

MAT drop of 4 - 6°C (Hren et al., 2013), which is comparable to the $U^{K'}_{37}$ estimated SST change from the high latitude North Atlantic ODP Site 913, but not the smaller SST change at the more comparable latitude ODP Site 336 (Liu et al., 2009).

Summer temperatures for the Hampshire Basin fell by around 4°C during the EOT (Grimes et al., 2005), but did not drop again during the EOIS (see section 1.2). Paleosols of the Hampshire Basin show minimal changes in temperature but an increase in precipitation (Sheldon and Tabor, 2009). Some of the discrepancies between these temperature signals may be due to
differences in sampling rates during key events of the EOT.

## 4.5 Australia and New Zealand

In Australia, the EOT is associated with the loss of rarer taxa in pollen records rather than significant turnovers (Macphail, 2007). There is a diversity drop from the middle to late Eocene into the latest Eocene – early Oligocene (Martin, 2006). A recent review of the distribution of paleo-biomes in Australia showed no change between the late Eocene and the early
Oligocene (Pound and Salzmann, 2017), though data coverage is sparse apart from in the south of the continent. However, a recent study of rainforest flora in south-eastern Australia shows a transition from warm-temperature rainforests in the late Eocene to cool-temperature rainforests in the early Oligocene (Korasidis et al., 2019). Those biome flora suggest a shift in MAT from 14-20 °C in the late Eocene to 10-14 °C in the early Oligocene (Korasidis et al., 2019). Terrestrial paleoclimate reconstructions of temperature also show a cooling at around 36 Ma (Pound and Salzmann, 2017). The New Zealand records
show a warm humid forest with a gradual turnover of palynomorphs through the late Eocene and the early Oligocene (Homes et al., 2015; Pocknall, 1991).

## 4.6 Antarctica

On Antarctica it is known that from the equable climates of the middle Eocene there was a progressive drop in plant diversity and stature, from evergreen forests to low-lying vegetation (Francis et al., 2008; Pound and Salzmann, 2017). Changing $\delta^{13}C$
measurements from late Eocene leaves and pollen have been interpreted as decreasing moisture availability on the Antarctic Peninsula (Griener et al., 2013). Vegetation at Wilkes Land, East Antarctic, changed from an early Eocene subtropical to a cool temperate forest, indicating a 5°C decline in MAT (Pross et al., 2012). A further change towards a less diverse, cool-temperate shrubland and forest indicates further cooling at Wilkes Land at the onset of the Oligocene (Strother et al., 2017). Other evidence supporting decreasing moisture availability is demonstrated by a shift from chemical weathering in a humid
environment, to physical weathering associated with a colder more arid regime (Basak and Martin, 2013; Dingle et al., 1998; Ehrmann and Mackensen, 1992; Robert and Kennett, 1997; Wellner et al., 2011). This aridification of the Antarctic continent is attributable to a partly glaciated continent in the late Eocene. A new bedrock topography for Antarctica allows for an early Oligocene ice sheet of greater areal extent than today (Wilson et al., 2012), raising the possibility that the temperature component of the EOT $\delta^{18}O$ increase was more modest than previously suggested (Wilson et al., 2013).


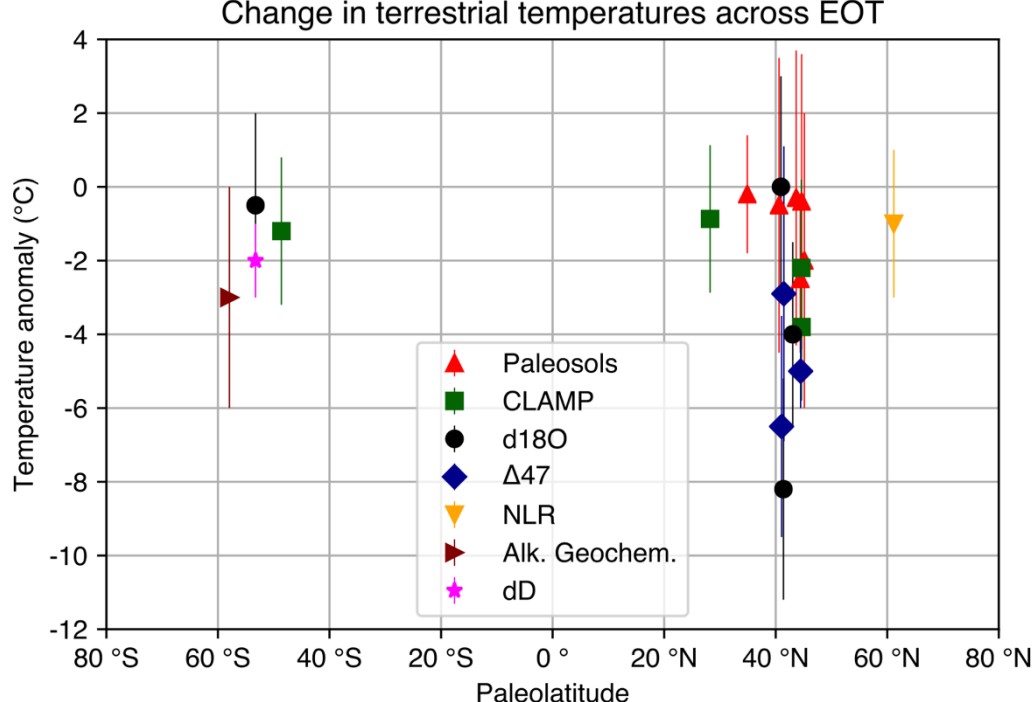

**Figure 4:** Summary of terrestrial air temperature change across the EOT from proxies Paleosols, CLAMP, $\delta^{18}O$, $\Delta_{47}$, Nearest Living Relative (NLR), alkaline geochemistry and $\delta D$ (hydrogen isotopes). Data is compiled from (Boardman and Secord, 2013; Colwyn and Hren, 2019; Eldrett et al., 2009; Fan et al., 2017; Gallagher and Sheldon, 2013; Héran et al., 2010; Herman et al., 2017; Hinojosa and Villagrán, 2005; Hren et al., 2013; Kohn et al., 2004; Kvaček et al., 2014; Lielke et al., 2012; Meyers, 2003; Page et al., 2019; Passchier et al., 2013; Roth-Nebelsick et al., 2017; Sheldon and Tabor, 2009; Zanazzi et al., 2007). Where possible, we apply the same method as in Figure 3, i.e. the 'late Eocene' is taken the average temperature from 38 to 33.9 Ma and the 'early Oligocene' is taken as the average from 33.9 to 30 Ma, and the temperature change shown here is the difference. However, in a number of cases, only a relative temperature change across the EOT was given in the original literature. We therefore limit our compilation to temperature anomaly only. The compilation shown above is provided in digital form in Supplementary Table S2.

## 5 CO₂ and carbon cycle dynamics

The concentration of carbon dioxide in the atmosphere ($p$CO₂) is a primary driver of global climate change on geological timescales (Berner and Kothavala, 2001; Foster et al., 2017; Royer et al., 2004), and changes in $p$CO₂ have been linked to the phase of acute climate change at the EOT (DeConto and Pollard, 2003; Heureux and Rickaby, 2015; Pearson et al., 2009; Steinthorsdottir et al., 2016). However atmospheric $p$CO₂ reconstructions for the EOT are sparse, variable and, in some cases, contradictory and not readily reconciled with paleo-temperature proxy records or numerical model hindcasts (Beerling and Royer, 2011; Heureux and Rickaby, 2015; Pagani et al., 2005; Pearson et al., 2009; Royer et al., 2004; Zhang et al., 2013). New well-resolved $p$CO₂ records with strong age control are pressingly needed. Four proxies have been identified as particularly useful for Cenozoic $p$CO₂ reconstructions by the Intergovernmental Panel on Climate Change (IPCC, 2013). These are the marine carbon and boron isotope proxies, and the terrestrial paleosol carbon and stomatal density proxies (Beerling and Royer, 2011). Below, we discuss the development and state of the art of existing EOT $p$CO₂ records constructed using marine

and terrestrial proxies.

## 5.1 Marine $p$CO$_2$ proxies

To date, the most detailed pre-Pleistocene climate records are derived from marine geochemical proxies, including boron
isotopes ($\delta^{11}$B) in planktonic foraminiferal calcite (Anagnostou et al., 2016; Foster et al., 2012; Greenop et al., 2017; Pearson
et al., 2009; Pearson and Palmer, 1999, 2000) and carbon isotopes ($\delta^{13}$C) in marine organic biomarkers (Heureux and Rickaby,
2015; Liu et al., 2009; Pagani et al., 2005, 2011; Zhang et al., 2013). Each proxy has its own limitations and uncertainties,
which initially led to divergent estimates of $p$CO$_2$ using these different proxies. However, recent efforts to address such
uncertainties and limitations have led to a more coherent picture of the evolution of $p$CO$_2$ through the Cenozoic from marine
proxies.

While the theoretical basis of the boron isotope proxy is well understood, a major uncertainty in reconstructing surface ocean
pH is estimating the boron isotopic composition of seawater (Greenop et al., 2017). A further major uncertainty comes into
play when a second carbonate system parameter (e.g. total alkalinity) is required to calculate $p$CO$_2$ from seawater pH, as well
as the major ion composition of seawater, which impacts key dissociation constants. Nevertheless, significant progress has
been made to reduce these uncertainties (Anagnostou et al., 2016; Greenop et al., 2017; Sosdian et al., 2018). For the Eocene
$p$CO$_2$ estimates, seawater $\delta^{11}$B has been estimated using the $\delta^{11}$B-pH relationship, while self-consistent estimates of the second
carbonate parameter have been determined using Earth System modelling (Anagnostou et al., 2016). For the alkenone $\delta^{13}$C
proxy, there are many factors that can impact algal growth conditions and inaccurate temperature reconstructions have also
been known to bias $p$CO$_2$ reconstructions (Pagani et al., 2011; Zhang et al., 2013). Algal carbon-concentrating mechanisms
may also lead to biased pCO$_2$ reconstructions when using the alkenone $\delta^{13}$C proxy in low-CO$_2$ intervals of the Neogene, but
is unlikely to be a significant issue at the EOT (Zhang et al., 2013).

The boron isotope proxy suggests atmospheric $p$CO$_2$ was 1400 ± 470 ppm in the early Eocene, and decreased by several
hundred ppm through the Eocene over several million years (Anagnostou et al., 2016). In the late Eocene (Bartonian-
Priabonian), $p$CO$_2$ reconstructions are variable, but the boron isotope and alkenone proxy both indicate $p$CO$_2$ concentrations
around 1000 ppm (Anagnostou et al., 2016; Zhang et al., 2013). The EOT itself appears to be associated with a further, and
perhaps steeper, decline in $p$CO$_2$, with both proxies supporting the passing of a modelled glaciation threshold of ~750 ppm
(DeConto and Pollard, 2003; Pagani et al., 2011; Pearson et al., 2009; Zhang et al., 2013), although this modelled threshold
itself is highly uncertain (Gasson et al., 2014). A $\delta^{11}$B-based record from Tanzania sediments also suggests an intriguing
transient pCO$_2$ increase associated with the second $\delta^{18}$O step (Pearson et al., 2009).

**5.2 Terrestrial proxies**

The stomatal $CO_2$ proxy is based on the empirically and experimentally demonstrated inverse relationship between the density of stomata on the leaf surfaces of most land plants and $pCO_2$ (Beerling, 1998; Franks et al., 2014; Hincke et al., 2016; Konrad et al., 2008; Kürschner et al., 2008; McElwain and Chaloner, 1995; Royer et al., 2001; Steinthorsdottir et al., 2016, 2019, 2020, 2011, 2013; Steinthorsdottir and Vajda, 2015; Wagner et al., 1996; Woodward, 1987). Previous studies using the stomatal proxy method of $pCO_2$ reconstructions for the time intervals either side of the EOT (here Bartonian-Rupelian) are still relatively few, derived from mostly low-resolution datasets consisting of a variety of fossil plant taxa, and marred by limitations in chronological accuracy and correlation to marine records. Consequently, the $pCO_2$ records have so far been highly heterogeneous.

The three current methods of stomatal $pCO_2$ reconstruction (e.g. McElwain and Steinthorsdottir, 2017; Steinthorsdottir et al., 2020) include: 1) the semiquantitative stomatal ratio method, which compares the stomatal density (SD) or stomatal index (SI – the % of stomata relative to all leaf epidermal cells) of fossil plants with the SD/SI of its nearest living relative (NLR), to estimate paleo- $pCO_2$ (McElwain and Chaloner, 1995), 2) the empirical transfer function method, using experimental data sets of NLR responses to variations in $pCO_2$ (e.g. Kürschner et al., 2008; Wagner et al., 1996), and 3) mechanistic gas exchange modelling which requires numerous additional parameters such as paleotemperature and leaf $\delta^{13}C$ (Franks et al., 2014; Konrad et al., 2008). All of these methods have been applied to reconstruct $pCO_2$ records spanning the Eocene-Oligocene (Foster et al., 2017; Figure 5), with the highest-resolution dataset produced using the stomatal ratio method (Steinthorsdottir et al., 2016). Due to the semiquantitative approach and potential inter-method variability, these estimates are considered to be less robust in their absolute values of $pCO_2$ than the marine estimates (Section 5.1). However, stomatal records are a valuable indicator of relative change of $pCO_2$ in the terrestrial realm, especially in data-poor intervals including several stages of the Eocene and Oligocene (Beerling and Royer, 2011; Foster et al., 2017).

Early results based on datasets of the gymnosperms *Ginkgo biloba* and *Metasequoia glyptostroboides* from the USA suggested that $pCO_2$ was more or less stable between 300 and 450 ppm during the Eocene and Oligocene (Royer et al., 2001) - however, the fossil leaf record was of too low resolution to draw strong conclusions. Another early study, based on data gathered mainly from published images of fossil *Ginkgo* specimens from Russia and USA, suggested a decrease in $pCO_2$ across the EOT (Retallack, 2001), from ~1300 ppm in the Bartonian, to ~420 ppm at the EOT, and ~330 ppm in the Rupelian. A further stomatal dataset from Germany, based on fossil angiosperm leaves from the species *Eotrigonobalanus furcinervis* (Fagaceae) and *Laurophyllum acutimontanum* (Lauraceae), again of low temporal resolution, suggested that $pCO_2$ was higher before than after the EOT (Roth-Nebelsick et al., 2004). A more recent study from a temporally restricted sedimentary succession in Canada suggested high but decreasing $pCO_2$ at the Bartonian/Priabonian boundary (from ~1000–700 ppm to ~450 ppm), based on a dataset of *Metasequoia* fossil needles (Doria et al., 2011), but does not include the EOT or the Rupelian. In contrast, a

study based on various angiosperm species using a leaf gas exchange model suggested more modest as well as stable $p$CO$_2$ of ~470 ppm during the Bartonian and Priabonian, decreasing to ~400 ppm after the EOT in the Rupelian (Grein et al., 2013). A subsequent study from the same region compiled all data in broad temporal bins and reconstructed early Oligocene to early Miocene $p$CO$_2$ to ~400 ppm throughout, despite significant changes in stomatal densities (Roth-Nebelsick et al., 2012, 2014). Two studies with restricted temporal ranges reconstructed $p$CO$_2$ in the Bartonian, indicating 400-500 ppm using *Metasequoia*

from Canada (Maxbauer et al., 2014) and ~390 ppm using the podocarp conifer *Nageia maomigensis* from China (Liu et al., 2016).

Recently, a new relatively high-resolution dataset consisting of *Eotrigonobalanus furcinervis* from Germany was published, including data points thought to be temporally located immediately before and after the EOT (Steinthorsdottir et al., 2016).

The results show $p$CO$_2$ of ~650 ppm in the Bartonian, decreasing to ~550–400 ppm in the Priabonian, and ~410 ppm at the EOT and the earliest Rupelian (Steinthorsdottir et al., 2016). This higher-resolution record shows a distinct ~40% Bartonian-Priabonian decrease in $p$CO$_2$, highly comparable to the marine isotope temperature records, but reaching stable levels by the EOT, and not recording a significant $p$CO$_2$ decrease at the EOT proper, unlike in the marine temperature records (Steinthorsdottir et al., 2016; Zachos et al., 2001, 2008). This discrepancy between $p$CO$_2$ and temperatures suggests that there

are factors other than greenhouse forcing that contribute to the threshold climate response of glaciation. New results based on Lauraceae leaf fragments from the Southern Hemisphere (Australia and New Zealand) further confirm Late Eocene $p$CO$_2$ mostly in the order of 550–450 ppm, but are not sufficiently chronologically well-constrained to confirm falling $p$CO$_2$ prior to the EOT (Steinthorsdottir et al., 2019).

Other results show a significant decrease in stomatal density values (indicating increasing $p$CO$_2$) of extinct *Platanus neptuni* before the EOT (Moraweck et al., 2019). This unexpected trend, contradictory to the stomata density-$p$CO$_2$ relation previously recorded, is not yet understood but is consistent with the suggestion that $p$CO$_2$ change prior to the EOT caused plant responses. Further, $p$CO$_2$ reconstructed using a mechanistic gas exchange model of Konrad et al. (2008) applied to two fossil species from northern central Europe, *Rhodomyrtophyllum reticulosum* and *Platanus neptuni*, records no significant decrease across

the EOT (Moraweck et al., 2019), in agreement with Steinthorsdottir et al. (2016), but not recording decreasing $p$CO$_2$ before the EOT. It should be noted that the gas-exchange model of Konrad et al. (2008) has recently been tested with modern material and was shown to produce the most accurate $p$CO$_2$ estimates when used with multiple species, to derive a consensus $p$CO$_2$ (Grein et al., 2013).

When focusing on datasets from central Europe, terrestrial plant based $p$CO$_2$ records support the plant-derived temperature records by indicating no abrupt decrease or environmental change across the EOT (Kunzmann et al., 2016; Teodoridis and Kvaček, 2015). A detectable but not fundamental change in vegetation, temperatures and $p$CO$_2$ is however evident from the interval prior to the EOT (Kunzmann et al., 2016; Kvaček et al., 2014; Steinthorsdottir et al., 2016; Teodoridis and Kvaček,

2015). Preliminary results of MAT estimations based on sedimentary GDGT values from some central German sites are in accordance with temperature estimates from plant fossils, i.e. based on CLAMP and the NLR approach. In combination these data refer to a successive decrease in MAT across the Priabonian and the EOT but not to a significant drop during the EOT.

## 5.3 Synthesis of EOT $pCO_2$ change

The most recent marine records indicate $pCO_2$ of ~1000 ppm in the Bartonian-Priabonian, decreasing to ~700–800 ppm into the Rupelian. Stomatal proxy-based $pCO_2$ records generally indicate elevated Bartonian-Priabonian $pCO_2$ of ~500–1000 ppm, decreasing ~40% before the EOT to $pCO_2$ of ~400 ppm, and continuing in the Rupelian with $pCO_2$ of ~400 ppm or lower. The direction and approximate magnitude of $pCO_2$ change leading up to the EOT is therefore consistent between proxies, even though the stomatal records consistently yield lower $pCO_2$ levels than the marine proxies. We consider the higher $pCO_2$ estimates based on marine proxies to be most robust indicator of $pCO_2$ at this time, since they have been shown to reproduce ice-core $CO_2$ well (Foster and Rae, 2016), and to agree better with the available modelling evidence from warm climate simulations of the Eocene, and estimated thresholds for glaciation of Antarctica (section 6.2). Some terrestrial records also indicate a decrease in $CO_2$ but the decrease is more gradual and long term than in the marine records.

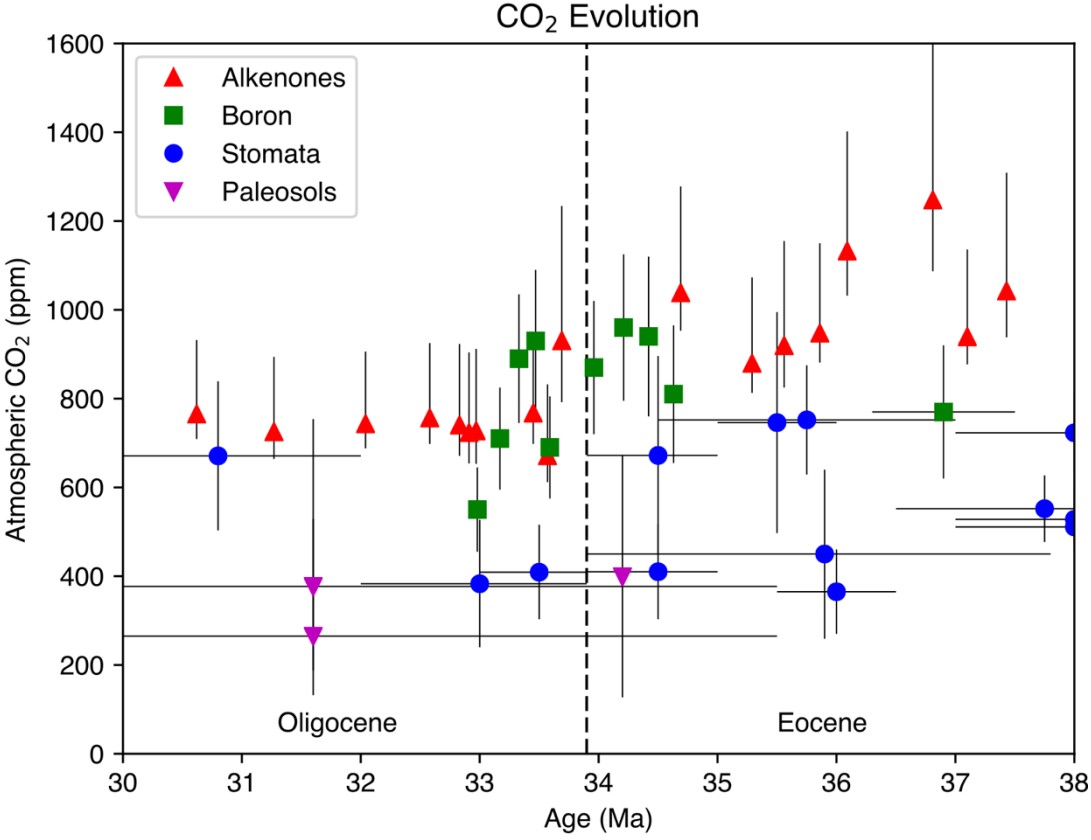

**Figure 5:** Atmospheric $CO_2$ evolution from 44 to 24 Ma from the compilation of Foster et al., (2017), incorporating data from its original data sources (Anagnostou et al., 2016; Doria et al., 2011; Erdei et al., 2012; Franks et al., 2014; Pearson et al., 2009; Roth-Nebelsick et al., 2012, 2014; Steinthorsdottir et al., 2016; Zhang et al., 2013), and Steinthorsdottir et al., (2019).

## 6 Insights into the EOT from modelling studies

In this section we qualitatively synthesise previous modelling studies that have focussed on the EOT. In particular, we discuss the modelled response to changing paleogeography (Section 6.1) and to changes in $CO_2$ (Section 6.2). Finally, we describe carbon-cycle models that have explored mechanisms behind $CO_2$ changes at the EOT (Section 6.3).

### 6.1 Modelling the response to changing paleogeography at the EOT

The widening of the Southern Ocean Drake Passage and Tasman gateways has long been considered as a primary driver for the initiation of the AMOC and Antarctic glaciation at the EOT (Section 2.2). Many climate modelling studies have tested the effect of opening these Southern Ocean gateways and found cooling effects on the southern high latitudes (Cristini et al., 2012; Elsworth et al., 2017; England et al., 2017; Mikolajewicz et al., 1993; Sijp et al., 2009, 2014; Sijp and England, 2004; Toggweiler and Bjornsson, 2000; Viebahn et al., 2016; Yang et al., 2014). These studies have variously found that opening Southern Ocean gateways can decrease southward heat transport (e.g. Sijp et al., 2009), trigger the onset of an AMOC (e.g. Yang et al. 2014), and enable some degree of cooling over Antarctica. However, these approaches do not reconcile the timing and evolution of the gateway evolution of the EOT, since they employed either modern-day or idealised geography with specific gateway perturbations. In contrast, climate model simulations that do employ Eocene boundary conditions indicate that Southern Ocean gateway opening caused only a modest change in ocean poleward heat transport and could therefore not be directly responsible for the initiation of the AIS (Goldner et al., 2014; Huber et al., 2004; Huber and Nof, 2006; Huber and Sloan, 2001; Sijp et al., 2011; Zhang et al., 2011). Furthermore, opening the Southern gateways under Eocene-like $CO_2$ forcing may result in a weaker ACC than under pre-industrial conditions (Lefebvre et al., 2012). The long-term evolution of Southern Ocean gateway opening has been found to cause ~3°C of bottom water cooling (Sijp et al., 2014), which may explain some of the observed benthic cooling (Section 3.2). However, Drake Passage opening probably affected deep ocean temperatures and the strength of the ACC. Hill et al. (2013) showed that despite deep water connections through both Drake Passage and the Tasman Gateway, a coherent ACC could not develop until the Australian continent was sufficiently equatorward such that it no longer inhibited strong zonal flow in the Southern Ocean.

Imposing an ice sheet in a climate model has been shown to have a significant impact on the ocean circulation and deep water formation regions (Goldner et al., 2014; Kennedy et al., 2015). In particular, the presence of an Antarctic ice sheet may enhance westerly winds over the Southern Ocean, leading to enhanced Southern Ocean deep ocean formation and benthic cooling (Goldner et al., 2014). This result suggests that Southern Ocean gateway changes play a secondary role to radiative forcing, since the ocean circulation change are a consequence of the glaciation, rather than a cause. Other ocean gateways may also

play an important role. In the late Eocene continental configuration, the Central American Seaway and the Tethys gateway, connecting the Indian and Atlantic oceans, were wider than today. The importance of the open Tethys gateway for the EOT circulation has not received much attention but Zhang et al. (2011) found that the tropical seaways need to be sufficiently constricted before the southern high latitudes can cool substantially. This cooling is related to a transition from an ocean circulation with southern hemisphere deep water formation to the modern-like circulation with deep water formation in the

North Atlantic.

Recently, focus has shifted to the role of Arctic-Atlantic gateways around the EOT. The evolution of the Arctic-Atlantic gateways has been shown to have a strong influence on the salinity of the North Atlantic and therefore on the AMOC (Hutchinson et al., 2019; Roberts et al., 2009; Stärz et al., 2017; Vahlenkamp et al., 2018b). The deepening of the Greenland-

Scotland ridge at the EOT has been proposed as a trigger for the AMOC (Abelson and Erez, 2017; Stärz et al., 2017). According to this hypothesis, the deepening changes the flow across the ridge from a shallow unidirectional flow to a deeper bi-directional flow which allows salty subtropical water to penetrate further north and enable North Atlantic sinking. Hutchinson et al. (2019) recently proposed that it is the tectonic closing of the shallow Barents Sea gateway, just prior to the EOT, that initiated the AMOC, by closing off the pathway of extremely fresh Arctic water to the North Atlantic. This theory suggests that the North

Atlantic reconnected to the Arctic when the Fram Strait opened in the early Miocene (Jakobsson et al., 2007). In the Hutchinson et al. (2019) study, neither Southern Ocean gateways changes, Greenland-Scotland Ridge changes, nor $CO_2$ forcing changes could similarly overcome the freshening effect of the Arctic to allow an AMOC. However, their study did not test the feedback of these circulation changes on the carbon cycle, making it unclear what the climatic impact of the Arctic closure would have been. Using an Earth System Model, (Vahlenkamp et al., 2018a, 2018b) experimented with similar changes in the North

Atlantic gateways to investigate an alternative idea of significant AMOC behaviour since the early–middle Eocene. They were able to initiate an AMOC when the Greenland Scotland Ridge reached a threshold depth of deeper than 200 m but, only when the Arctic Ocean brackish water outlets were shut off from the North Atlantic. Timing of changes in Arctic-Atlantic 'plumbing', thus, appear to be a critical factor in allowing an AMOC to start up in the Warm Paleogene, and a key area for future research.


In all these studies, it is assumed that the final state of the simulation is the only steady solution for that climate and boundary conditions. While there is no consensus yet (Nof et al., 2007), the present day climate is thought to have two global circulation modes; the observed AMOC with sinking in the north, and a southern-sinking only mode with no AMOC (Liu et al., 2017; Srokosz and Bryden, 2015). In continental geometries other than the present day different circulation patterns and co-existing

equilibria may be possible but have not been systematically searched for so far (Baatsen et al., 2018). In coupled Eocene simulations, centers of deepwater formation include the North and South Pacific (Hutchinson et al., 2018; Thomas et al., 2014) and the North and South Atlantic (Huber et al., 2003; Huber and Sloan, 2001). Conceptual climate models have suggested a potential role for meridional overturning circulation transitions in the EOT (Tigchelaar et al., 2011).

## 6.2 Modelling the response to $CO_2$ decrease at the EOT

A reduction in atmospheric $CO_2$ is hypothesised to be a primary cause of the EOT, because it can plausibly explain both long term cooling during the Eocene, and provide a trigger for the glaciation of Antarctica (DeConto and Pollard, 2003). Although proxy reconstructions of atmospheric $CO_2$ during the Eocene have large uncertainties (Section 5), there is general agreement that climate cooled and $CO_2$ declined through the Eocene (Anagnostou et al., 2016; Foster et al., 2017), making long-term $CO_2$ drawdown from the atmosphere a prime underlying forcing mechanism for the EOT. Furthermore, $CO_2$-forced climate-ice

sheet model experiments yield $\delta^{18}O$ series (DeConto and Pollard, 2003) that closely match the overall form of our best-resolved EOT data sets (Coxall et al., 2005; Coxall and Wilson, 2011).

A long-standing problem in modelling the Eocene climate is to reproduce the low meridional temperature gradient recorded in observations. Proxies suggest that high latitude SSTs were more than 20°C warmer than present day during the early Eocene

(Bijl et al., 2009), terrestrial anomalies were 20-40°C (Huber and Caballero, 2011), while tropical temperatures were some 5-10°C warmer than modern (Huber, 2008; Huber and Sloan, 2000). Evidence of frost intolerant flora and fauna in high latitudes (Greenwood and Wing, 1995) provides a challenge to explain how the climate maintained such a low meridional temperature gradient.

When climate models are forced using proxy data-based estimates of Eocene $CO_2$, they generally fail to capture these flatter meridional temperature gradients (Huber et al., 2003; Roberts et al., 2009; Shellito et al., 2003). One method that has been used to address this high-latitude cold bias is to increase the $CO_2$ to extremely high values (2240 or 4480 ppm) (Cramwinckel et al., 2018; Eldrett et al., 2009; Huber and Caballero, 2011; Winguth et al., 2010). These extremely high $CO_2$ experiments yield an improved match to high-latitude temperature proxies and temperature gradients (Huber and Caballero, 2011; Lunt et

al., 2012). Because these extremely high $CO_2$ concentrations are greater than those implied by proxies, this finding also suggests that modelled climate sensitivity to $CO_2$ forcing may be too low, probably because of positive feedbacks that are either missing or too weak in the models. Several missing feedbacks suggested recently are those associated with cloud physics and/or greenhouse gases in addition to $CO_2$ (Beerling et al., 2011; Kiehl and Shields, 2013; Sagoo et al., 2013; Zhu et al., 2019). We also note that recent proxy data from the warmest regions of the tropics (Tanzania, Java) indicate tropical

temperatures of up to 35°C in the middle-late Eocene (Evans et al., 2018; Pearson et al., 2007). These temperatures imply a somewhat larger meridional gradient than previously suggested, helping to reduce the magnitude (but not eliminate) the model-data mismatch. In addition, several models have now achieved lower meridional temperature gradients through a combination of higher resolution, which tends to increase poleward heat transport, and improved Eocene boundary conditions (Baatsen et al., 2020; Hutchinson et al., 2019; Zhu et al., 2019). A recent model-data comparison demonstrates that the Oligocene retains

a low meridional temperature gradient similar to the late Eocene, which is not well explained by currently available climate models (O'Brien et al., 2020). In the Oligocene, $CO_2$ proxy estimates are lower than in the Eocene, making it arguably more

difficult to model the low temperature gradient with realistic $CO_2$ forcing (O'Brien et al., 2020), although the $CO_2$ proxies carry large uncertainty.

Despite the challenges faced in modelling the early Eocene, the observed cooling during the Eocene of bottom waters (Zachos et al., 2001) and high latitude SSTs (Bijl et al., 2009) and terrestrial temperatures can plausibly be explained by a reduction in $CO_2$ in climate model simulations (Eldrett et al., 2009; Liu et al., 2009). Furthermore, crossing a $CO_2$ threshold of Antarctic glaciation may also explain several degrees of bottom water cooling, through consequent shifts in Southern Ocean winds and changes to Southern Ocean circulation (Goldner et al., 2014). A key challenge to adequately testing the $CO_2$ forcing hypothesis

is to derive a threshold level of $CO_2$ for glaciation from climate model reconstructions. The first study to do so found a glaciation threshold of around 780 ppm (DeConto and Pollard, 2003), in approximate agreement with $CO_2$ proxies. However, a recent inter-comparison of Eocene climate models used to force an ice sheet model found that this threshold varied significantly between models, from roughly 560 to 920 ppm (Gasson et al., 2014). Differences in the lapse-rate feedback were identified as the leading cause of this spread, although there were also differences in the paleogeographic boundary conditions.


All ice-sheet modelling studies of the EOT to date have used prescribed climate states to force the glaciation. Likewise, coupled ocean-atmosphere-sea ice models currently prescribe ice sheets as either present or absent. Running a full-complexity climate model synchronously with an ice sheet model remains a major technical challenge, and has yet to be implemented for the Eocene or Oligocene. However, innovative asynchronous coupling, such as the "matrix-method" (Pollard, 2010), has shown

some promise by allowing a better representation of the ice albedo feedback, leading to a similar yet slightly revised upward glaciation threshold of ~900 ppm (Ladant et al., 2014b).

**6.3 Carbon Cycle Modelling**

A slow decline in atmospheric $CO_2$ remains a likely priming mechanism for the inception of large ice sheets on Antarctica and

this pivotal transition in Cenozoic climate was associated during the EOT with pronounced rapid perturbation to the global carbon cycle as indicated by a transient increase in ocean $\delta^{13}C$ and a permanent deepening of the CCD (Coxall et al., 2005). Thus, numerical carbon cycle model experiments provide useful insight to forcing mechanisms and feedback processes involved.

Many hypotheses have been posited to explain the carbon cycle perturbations at the EOT (Armstrong McKay et al., 2016; Coxall et al., 2005; Coxall and Wilson, 2011; Merico et al., 2008). Some of the leading hypotheses include a shift from shelf to basin carbonate fractionation (Opdyke and Wilkinson, 1988), increases in organic carbon burial (Olivarez Lyle and Lyle, 2006), feedbacks between ice sheet coverage and silicate weathering (Zachos and Kump, 2005), and an ecological shift from calcareous to siliceous plankton (Falkowski et al., 2004). Carbon cycle box models suggest that the best fit to observations is

achieved by a shift from shelf-to-basin carbonate fractionation (Armstrong McKay et al., 2016; Merico et al., 2008). In this interpretation of events, the fall in sea-level due to Antarctic glaciation (i) reduces the global flux of carbonate into shallow water (reef, bank and shelf) sediments and (ii) exposes fresh, readily dissolved shelf carbonate sediments around the world to rapid subaerial weathering (Merico et al., 2008). The first of these two mechanisms drives the sustained CCD deepening from Eocene to Oligocene and the second drives a one-off dump of carbonate into the ocean that explains the observed initial

transient overshoot behaviour (Zachos and Kump, 2005) and, because the shelf carbonate reservoir is enriched in $^{13}$C relative to pelagic carbonate reservoir (Swart, 2008; Swart and Eberli, 2005), the transient increase in benthic $\delta^{13}$C (Armstrong McKay et al., 2016; Merico et al., 2008). If the isotopic fractionation between these two carbonate sediment reservoirs is modest, however, shelf-basin fractionation can only fully explain the transient increase in oceanic $\delta^{13}$C if the one-off dump of weathered shelf carbonate is questionably large (Merico et al., 2008). In their follow up study, Armstrong McKay et al. (2016) considered

this problem in detail and concluded that, unless shelf carbonates were substantially enriched in $^{13}$C relative to pelagic carbonates (by ~3‰), an additional process must also have contributed, with sequestration of $^{12}$C-enriched carbon into carbon capacitors, and possibly increased ocean ventilation, offering the best fit to the paleorecords when combined with shelf basin fractionation.

Palike et al. (2012) investigated causes of carbon cycle changes over the Eocene using the intermediate complexity climate model cGENIE. They suggest several mechanisms are needed to explain the CCD change in addition to the shelf-basin fractionation hypothesis above: (i) perturbations to continental weathering and solute input to the deep ocean, or (ii) changes in the partition of organic carbon flux between labile (organic carbon that is readily available for oxidation and driving carbonate dissolution) and refractory (carbon that is more resistant to degradation and largely preserved and buried).


The longer-term decline in $CO_2$ over the Eocene needs to be reconciled with the negative feedback between silicate weathering and surface temperature (Walker et al., 1981). Higher $CO_2$ causes warming and enhances the hydrological cycle, which leads to an increase in silicate weathering. The increase in weathering eventually lowers $CO_2$ and subsequent cooling, creating a dynamic equilibrium. This silicate weathering feedback is regulated by tectonic processes (Raymo and Ruddiman, 1992), since

mountain ranges give rise to greater weathering than low-lying regions (Maher and Chamberlain, 2014).

A climate model study suggests that opening and deepening of the Drake Passage could lower atmospheric $CO_2$ via the silicate weathering feedback (Elsworth et al., 2017). They suggested that the gateway transition enhanced the AMOC, leading to greater precipitation over land regions and a warmer northern hemisphere, both of which enhance silicate weathering and thus

drawdown of $CO_2$ (Maher and Chamberlain, 2014). However, this study used modern geography with selected gateway perturbations, whereas climate models using paleogeography from the late Eocene have yielded different patterns of overturning (Baatsen et al., 2020; Hutchinson et al., 2019). Furthermore, a hypothesised change in silicate weathering must be weighed against the CCD record, because silicate weathering changes have implications for carbonate weathering and

bicarbonate ion supply to the ocean (Armstrong McKay et al., 2016; Merico et al., 2008). Fyke et al. (2015) found opening Drake Passage led to a decrease in Atlantic carbon storage and an increase in Pacific and Southern Ocean storage, due to the enhancement of a modern-like AMOC. This led to an overall increase in global carbon storage in the ocean, though their implied drop in atmospheric $CO_2$ is relatively small (10-30 ppm). Incorporating carbon cycle processes into full complexity climate models with Eocene or Oligocene paleogeography thus remains an outstanding challenge (e.g. Goddéris et al., 2014).

Experiments using cGENIE report an increase in carbon re-mineralisation near the ocean surface when temperatures are very warm, such as in the early Eocene (John et al., 2013, 2014). The more temperature dependent re-mineralisation results in a shallower CCD, and a decrease in organic carbon burial, an effect which then decreased over the Eocene as temperatures decreased. This modelled mechanism is consistent with tropical records of $\delta^{13}C$ during the Eocene (John et al., 2013, 2014), providing a positive feedback on carbon dioxide changes, in opposition to the silicate weathering feedback.

## 7 Model-data intercomparison of temperature change across the EOT

Until this point, this review paper has synthesised the existing literature but has not presented any new quantitative analysis. Furthermore, we have in general presented the proxy and model-derived insights separately. Here, we combine the information from proxies and models and present a new model-data comparison and quantitative analysis of the mechanisms behind temperature change at the EOT. This section is in two parts: (7.1) a quantitative intercomparison of temperature change across the EOT from a subset of these previous studies, in which we identify those changes that are robust across models; and (7.2) a comparison of the modelled temperature changes with proxy SST and surface air temperature (SAT) data, in which we assess which models best fit the proxy reconstructions, and which mechanisms most likely explain the observed proxy temperature changes.

### 7.1 Intercomparison of modelled SAT change across the EOT

Here we present an intercomparison of some previous model results of SAT change across the EOT. We use SAT data because it provides a consistent surface temperature over both ocean and land regions that reflects changes across the globe. It also enables comparison with proxies of both marine and terrestrial data to be readily included. We include those models and studies for which the authors have provided their model results in digital form. The models and simulations included in this intercomparison are shown in Table 2. This itself is a subset of the simulations that were available – here we show only those simulations that allow us to compare the response of the models to a consistent forcing, for as many models as possible.

We first consider pairs of simulations that represent the response of the climate system to a perturbation in forcing that may have occurred across the EOT. These pairs can broadly be divided into three categories corresponding to three forcings: a $CO_2$ decrease, an increase in the volume and extent of the Antarctic ice sheet, and a paleogeographic change. Although these forcing

factors are in reality interdependent (for example the ice sheet change may itself be caused by a $CO_2$ change), for the purposes of modelling they are treated as independent mechanisms.

For the $CO_2$ forcing, we consider a halving of atmospheric $CO_2$, which for most models is from 1120 to 560 ppmv. However, for GFDL this is from 800 to 400 ppmv, and for NorESM-L there are $CO_2$ simulations at 980 and 560 ppmv. For the NorESM-L case, we scale the anomaly by a factor of $\log(2)/\log(980/560)$ in order to approximate the radiative forcing of halving $CO_2$. We emphasise here that halving $CO_2$ is not intended to be a realistic forcing perturbation for the EOT, but rather a standardised experimental protocol that can be used to establish the $CO_2$ climate sensitivity of the models. For the ice sheet forcing, we consider a change from an ice-free Antarctic, to an ice sheet similar in volume and area to that of today. However, the configuration of these ice sheets, and the ice-free state, does vary from model to model (see Supplementary Figure S1). The paleogeographic forcing is less consistent across the models, and includes modelled changes to gateways only (CESM and UVic), to west Antarctic geography (FOAM), or to global paleogeography (HadCM3BL) (see Supplementary Figure S3).

Before examining the SAT response of the system to these three forcings, it is useful to explore the absolute temperatures in the model simulations. The annual global mean SAT in each simulation is shown in Figure 6, while the spatial patterns for each individual model are shown in Supplementary Figures S1-S3. In terms of global mean surface temperature, the models fall approximately into two groups: (i) a cooler group, consisting of CESM_H, FOAM, HadCM3BL, NorESM-L, have global mean surface temperatures of around 17-19 °C at 560 ppm, and 21 to 23 °C at 1120 ppm; and (ii) a warmer group consisting of CESM_B, CESM_H (x2) and GFDL CM2.1, where temperatures are roughly 4 °C warmer for the equivalent level of $CO_2$ (Figure 6). A common factor in this split is that the warmer models have higher horizontal resolution (~1° ocean for CESM_B and GFDL CM2.1; ~2° atmosphere for CESM_H (x2) and CESM_B), although this is likely to be depend strongly on the individual model and boundary conditions used. It is also clear from Figure 6 and Supplementary Figures S1-S3 that the $CO_2$ forcing has a much greater effect on global mean SST than the ice or paleogeographic forcing.

| Model | Atmos resolution | Ocean resolution | Publication | Paleogeography | CO2 [ppm] | Antarc. ice | Tasman | Drake | CO2 expts | ICE expts | GEO expts | Model years |
|---|---|---|---|---|---|---|---|---|---|---|---|---|
| CESM_B (CESM 1.0.5) | 144 x 96 x 26 | 384 x 320 x 60 | Baatsen et al (2020) | 38 Ma | 560 | N | open | open | L | | | 3600 |
| | | | | Baatsen et al (2016) | 1120 | N | open | open | H | | | 4600 |
| CESM_H (CESM1.1) | 96 x 48 x 26 | 122 x 100 x 25 | Goldner et al (2014) | 45 Ma | 560 | N | closed | open | L | NI | | 3400 |
| | | | | Sewall et al (2000) | 560 | Y | closed | open | | I | | 3000 |
| | | | | | 1120 | N | closed | open | H | | | 3300 |
| | | | | | 1120 | N | closed | closed | | | B | 1300 |
| | | | | | 1120 | N | open | open | | | A | 1000 |
| CESM_H (x2) | 144 x 96 x 26 | 122 x 100 x 25 | * 2° atmosphere | | 560 | N | closed | open | | | | 1500 |
| FOAM | 48 x 40 x 18 | 128 x 128 x 24 | Ladant et al (2014a) Ladant et al (2014b) | 34Ma | 560 | N | open | open | L | NI | B | 2000 |
| | | | | | 560 | Y | open | open | | I | | 2000 |
| | | | | | 1120 | N | open | open | H | | | 2000 |
| | | | | 30Ma | 560 | N | open | open | | | A | 2000 |
| GFDL CM2.1 | 96 x 60 x 24 | 240 x 175 x 50 | Hutchinson et al (2018) | 38 Ma | 400 | N | open | open | L | | | 6500 |
| | | | | Baatsen et al (2016) | 800 | N | open | open | H | | B | 6500 |
| | | | Hutchinson et al (2019) | * Arctic closed | 800 | N | open | open | | | A | 6500 |
| HadCM3BL | 96 x 73 x 19 | 96 x 73 x 20 | Kennedy et al (2015) | Rupelian (28-34Ma) | 560 | N | open | open | L | NI | | 1422 |
| | | | | | 1120 | N | open | open | H | | | 1422 |
| | | | | | 560 | Y | open | open | | I | | 1422 |
| | | | | Chattian (23-28Ma) | 560 | N | open | open | | | A | 1422 |
| | | | | Priabonian (34-38Ma) | 560 | N | open | open | | | B | 1422 |
| NorESM-L | 96 x 48 x 26 | 100 x 116 x 32 | Zhang et al (2012, 2014) | 35 Ma (Scotese, 2001) | 560 | N | open | open | L | | B | 2200 |
| | | | | | 980 | N | open | open | H | | | 2200 |
| | | | | 33 Ma | 560 | N | open | open | | | A | 2200 |
| UVic | 150 x 100 x 1 | 150 x 140 x 40 | Sijp et al (2016) | 45 Ma | 1600 | N | open | open | | | A | 9000 |
| | | | | Sewall et al (2000)** | 1600 | N | open | closed | | | B | 9000 |

**Table 2:** Details of model simulations that are included in this intercomparison. For each model, the response to CO2, ΔTCO2, is given by
H-L; ΔTICE is given by I-NI; and ΔTGEO is given by A-B.

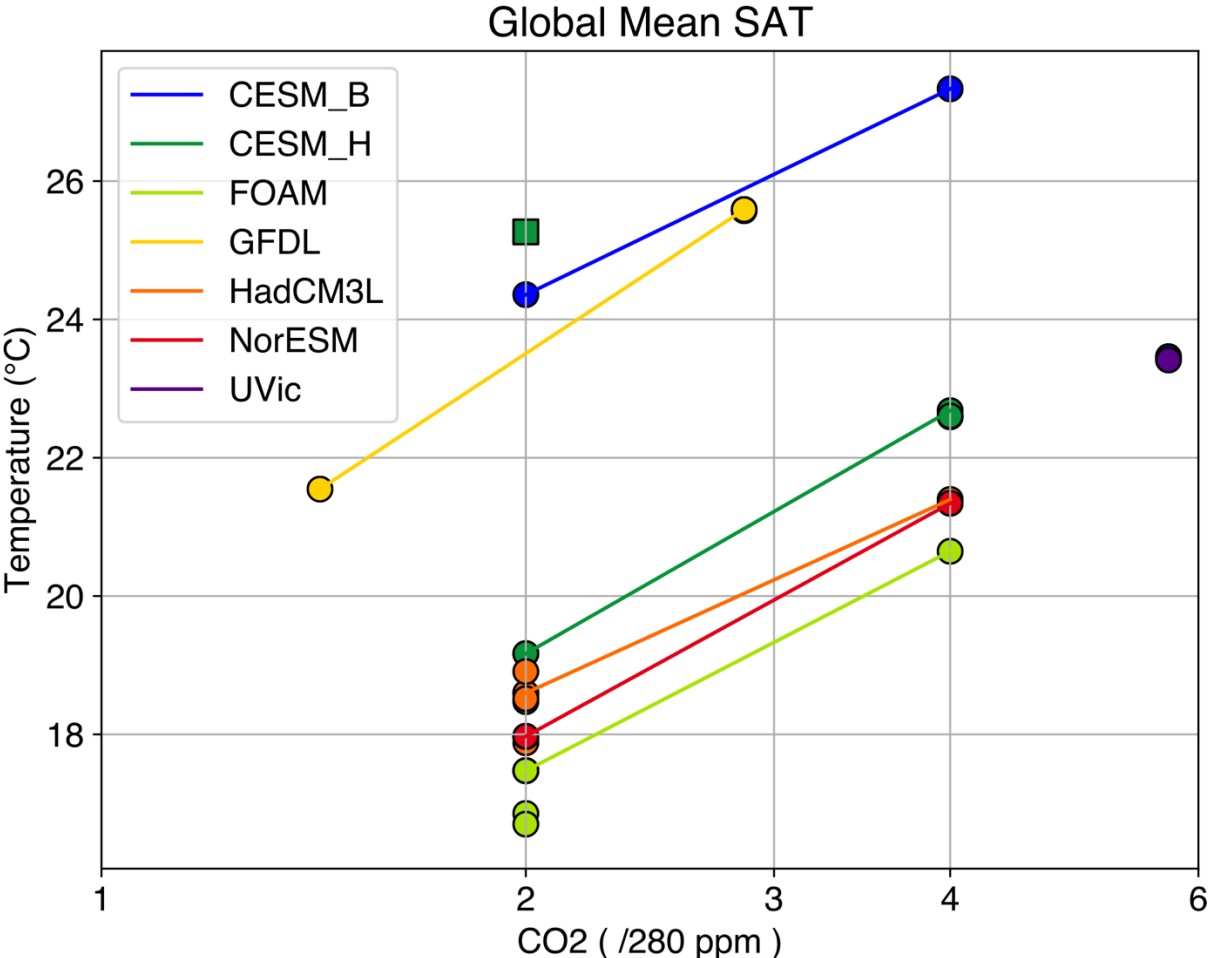

**Figure 6:** Global mean surface air temperature (SAT) for all models included in this intercomparison as a function of $CO_2$ concentration. The lines join simulations from a single model at different $CO_2$ concentrations. References for each model are CESM_H: (Goldner et al., 2014), UVic: (Sijp et al., 2016), FOAM : (Ladant et al., 2014a, 2014b), GFDL: (Hutchinson et al., 2018, 2019), HadCM3BL: (Kennedy et al., 2015), NorESM-L: (Zhang et al., 2014), CESM_B: (Baatsen et al., 2020). The dark green square is an additional simulation of CESM_H with 2° atmosphere resolution (Table 2).

The SAT responses of each of the individual models to the three forcings, $\Delta T_{CO2}$, $\Delta T_{ICE}$, $\Delta T_{GEO}$, are shown in Supplementary Figures S1, S2 and S3 respectively. We also include the annual mean sea ice distribution in each of CO2, ICE and GEO experiments in Supplementary Figures S4, S5 and S6 respectively. It is important to highlight that the changes shown have not necessarily been chosen to best represent the EOT transition. In particular, the $CO_2$ forcing shown is a halving of $CO_2$ in all models, and although some proxy $CO_2$ estimates are not inconsistent with this change (Pagani et al., 2011; Pearson et al., 2009), the data come with large uncertainties, albeit more so for absolute concentrations than for relative changes. In Section 7.3 we will explore this further, but here we recognise that the model responses are highly idealised and we treat them as sensitivity studies.

### 7.1.1 SAT response to $CO_2$ decrease, $\Delta T_{CO2}$

Here we consider the response to halving $CO_2$ in the absence of an ice sheet. There appear to be two different modes of SAT response to a halving of $CO_2$ (Supplementary Figure S1). In the first mode, CESM_H, CESM_B, FOAM, and GFDL respond with a cooling over all the globe, with greatest cooling in the higher latitudes. In the second mode, HadCM3BL and NorESM-L respond with cooling in most regions (with greatest cooling in the north Pacific), but with warming in the Pacific sector of the Southern Ocean. In HadCM3BL, this is associated with a switch in regions of deep water formation from dominant sinking in the southern Atlantic and north Atlantic at high $CO_2$, to dominant sinking in the southern Pacific and north Atlantic at low $CO_2$. The onset of sinking in the southern Pacific at low $CO_2$ leads to increased heat transport from the equatorial Pacific southwards, to such an extent that it leads to net warming in the Pacific sector of the Southern Ocean, despite the decrease in $CO_2$. Similar but weaker changes in ocean circulations happen in the NorESM-L. However, this warming response is highly sensitive to the boundary conditions, with other qualitatively similar simulations behaving very differently in the region, with some showing only cooling (Kennedy-Asser et al., 2019). In CESM, a switch in the mode of ocean circulation does not occur, with deep water formation in the Pacific sector of the Southern Ocean at both high and low $CO_2$ (albeit increased in intensity at low $CO_2$). Similarly, for GFDL there is no switch, with deep water formation in the South and North Pacific at high and low $CO_2$, and for FOAM there is no switch, with deep water formation predominantly in the North Pacific at high and low $CO_2$. The patterns of change in HadCM3BL and NorESM-L are remarkably similar except in the Arctic, where NorESM-L shows much more cooling than HadCM3BL. In this region FOAM also has very little cooling. This is because both HadCM3BL and FOAM have Arctic sea ice in both high and low $CO_2$ simulations, which maintains the SST close to 0°C.

The ensemble mean SAT change due to a halving of $CO_2$ is shown in Figure 7a. This shows that the greatest cooling is in the North Pacific and in the Atlantic and Indian sectors of the Southern Ocean. Most of the regional cooling is "robust" in that all models show a change of the same sign and are all within ±2 °C of the ensemble mean change. Exceptions are in the south Pacific (because some models show warming rather than cooling) and in the North Pacific and Arctic (because there is large variability in the amount of cooling predicted). Overall, the zonal mean cooling is approximately symmetric about the equator, with equatorial cooling of -2.6 °C and mid-high latitude cooling of -5.0 °C. While this symmetry is at odds with an inferred northward migration of the intertropical convergence zone from dust geochemistry (Hyeong et al., 2016), we stress that this result reflects the fact that the far-field cooling induced by imposing an Antarctic ice sheet (see Figure 7b) is much smaller than the global cooling induced by $CO_2$ forcing in these models.

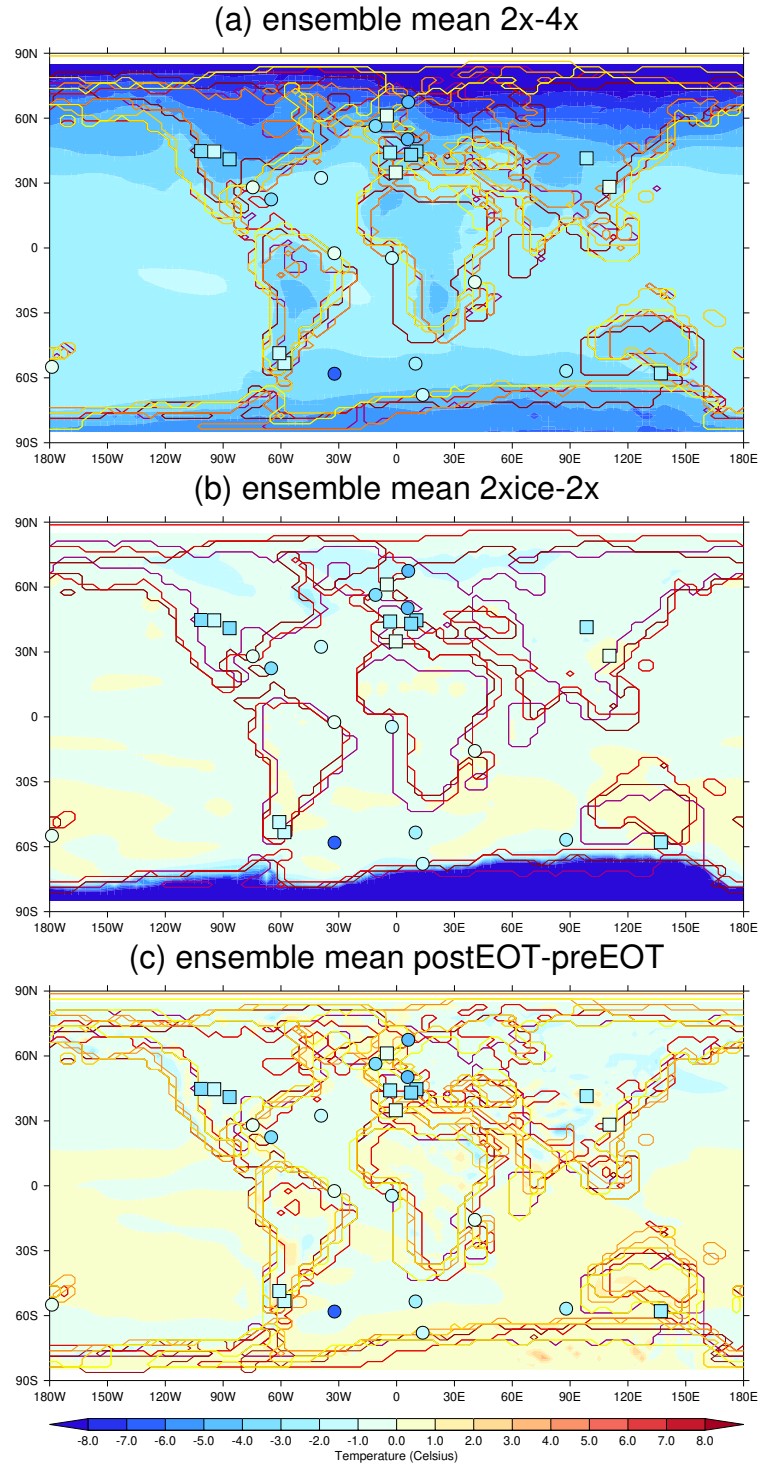

**Figure 7:** Ensemble mean modelled SAT response to (a) CO2 halving (ΔTCO2), (b) onset of ice on Antarctica (ΔTICE), and (c) paleogeographic change (ΔTGEO) across the EOT. The continental outlines for all models in each ensemble are shown. The marine proxy data are shown as filled circles, while the terrestrial proxy data are shown as filled squares.

### 7.1.2 SAT response to Antarctic ice, $\Delta T_{ICE}$

The three models that have carried out simulations with and without an Antarctic ice sheet show differing responses to the forcing (Supplementary Figure S2). CESM shows a cooling around the margins of Antarctica, and in the Pacific and Atlantic sectors of the Southern Ocean; FOAM shows cooling around the margins of Antarctica but warming throughout much of the Southern Ocean, and HadCM3BL shows cooling in the Southern Ocean except in the southern Pacific. The mechanisms behind the changes are described in the respective papers. In brief, Kennedy et al. (2015) attribute the warming in the Pacific sector

of the Southern Ocean in HadCM3BL to an increased N-S pressure gradient close to the polar front leading to stronger westerlies, intensification of the Ross Sea gyre, and a resulting increase in oceanic poleward heat transport. Goldner et al. (2014) focus on changes to deep ocean temperatures, highlighting the importance of increased easterly winds around the margins of Antarctica and resulting Ekman transport for bringing cold water to depths. They do not discuss mechanisms for the warming around Australia. Ladant et al. (2014a) do not discuss the mechanism of SST change following glaciation, but

Ladant et al. (2014b) do, for a similar pair of simulations. In their model, the presence of the Antarctic ice sheet enhances the strength of the Antarctic Circumpolar Current, and as a result, the Ross Gyre and Weddell Gyre initiate. They also highlight the importance of sea ice changes in amplifying the changes in SSTs. More recent work (Kennedy-Asser et al., 2019) has highlighted that the particularly strong Southern Ocean warming response in the HadCM3BL simulations could be an artefact of insufficient spin-up, with very long simulations showing a more muted temperature response. As a result, these HadCM3BL

results should be treated with caution.

In terms of the ensemble mean (Figure 7b), there are only a few regions where there is a robust SST signal. Robust cooling in response to the addition of the Antarctic ice sheet are around the margins of the East Antarctic ice sheet, in the Drake Passage, south of southern Africa, and the tropical and North Atlantic. There is a seemingly robust warming east of Australia, but this

is a small region and as such it is unclear if it occurs by chance.

### 7.1.3 SAT response to paleogeographic change, $\Delta T_{GEO}$

For examining the response to paleogeographic change (Supplementary Figure S3), for each model we first identify the pair of simulations for each model that represents the largest change in paleogeography across the EOT. For CESM this is an idealised gateway change from a closed Tasman and Drake Passages to open Tasman and Drake Passages. This forcing results

in a cooling in the Pacific sector of the Southern Ocean, and a slight warming in the rest of the Southern Ocean, but these changes are all small compared with those caused by $CO_2$ or ice sheet changes. For UVic, the forcing is an idealised gateway change from a closed to an open Drake Passage. This has a large impact on SSTs in the Southern Ocean, with cooling south of southern Africa, and a N-S dipole of warming and cooling in the Pacific sector of the Southern Ocean, associated with the transition from a gyre circulation to an Antarctic Circumpolar Current. For FOAM, the forcing consists of a localised change

in West Antarctica in which continent becomes ocean. However, despite the relatively small forcing the response is quite substantial, leading to global cooling of about 1 °C (Figure 6), especially in the south west Pacific. For HadCM3BL, the forcing is global in nature, but consists of relatively small changes to continental position and topography and bathymetry associated with plate tectonic movements from the Priabonian (34-38 Ma) to the Chattian (23-28 Ma). As a global mean the response is very small, but regionally it is quite large, for example in the North Pacific where there is a N-S dipole response, cooling in

the North Atlantic, and warming in the Southern Ocean. These changes are associated with the strengthening of deep water formation in the Atlantic sector of the Southern Ocean. NorESM-L shows a relatively muted response, but a substantial cooling of about 2 °C in the southwest Pacific.

Given that all models have carried out different simulations to differing forcings, interpreting the differences in response is

challenging. However, the ensemble mean response, shown in Figure 7c, can be interpreted as the best estimate of the SAT-response to paleogeographic change across the EOT, given the uncertainty in the paleogeographic forcing itself, as well as in the different models. The only substantial region of robust change is in the tropical Atlantic, where all models indicate a cooling response of about 1 °C.

**7.2 Model-data comparison across the EOT**

It is important to assess the realism (or otherwise) of the model simulations by comparison with evidence from the geological record. Such model-data comparison can also improve our understanding of the likely mechanisms that drove change. Given the 'snapshot' nature of the model simulations, and uncertainties in dating and limitations due to sparse data coverage, it is necessary to use data that extend throughout the EOT, and that are clearly either 'pre-EOT' or 'post-EOT'. Here we use an updated compilation of SST (Figure 3) and terrestrial surface temperature proxies (Figure 4), which we present in

Supplementary Tables S1 and S2 respectively. In total there are 24 data points from marine sources and 20 data points from terrestrial sources. Before performing the intercomparison, we first combine and average data points that either come from different proxies from the same location, or from neighbouring data points when they are less than one grid cell apart. This process yields a final proxy dataset of 27 data points, shown in Supplementary Table S3. Each data point is then given an equal weight in determining a root mean square error skill score.

**7.2.1 Comparison of model simulations with proxy data**

The observed proxy temperature changes compared with the individual model responses to $CO_2$, Antarctic ice and paleogeography are shown in Supplementary Figures S1-S3. As discussed in Section 7.1.1, when $CO_2$ is halved, all models predict a cooling at all sites, in agreement with the data, except NorESM-L that warms at one of the Arctic sites. There are no data to evaluate the warming signal in the southern Pacific in HadCM3BL and NorESM-L. When an Antarctic ice sheet is

imposed, the agreement is not so good, with all models showing warming for at least 2 of the sites. When paleogeographic

changes are imposed, the model-data agreement is worse again for most models, with all models showing warming for at least 3 of the sites. The exception is FOAM, for which all sites cool, in agreement with the data. The ensemble means capture the broad changes reasonably well, with all sites cooling for the $CO_2$ case, and all but one site cooling for the ice and paleogeographic changes.


This model-data comparison is limited by the fact that the models have carried out idealised simulations, especially for $CO_2$ forcing for which the halving of $CO_2$ is somewhat arbitrary. Although some proxy $CO_2$ records do indicate a drop of this order of magnitude (Pagani et al., 2011; Pearson et al., 2009), the associated uncertainties are large. Similarly, the changes to the Antarctic ice sheet imposed in the model may be greater or less than in reality, or the imposed changes in paleogeography may

be too extreme. As such, we carry out the model-data comparison such that each model SAT response to each forcing is scaled by a constant in such a way that it best fits the data. To assess the goodness-of-fit, we calculate a skill score, $s$, for each pair of model simulations, simply as the RMS difference between the proxy temperature and modelled temperature, calculated from model gridpoint that is in closest proximity to the data. For the purposes of the skill score we treat neighbouring sites (e.g. tropical sites 925 and 929) as a single data point by averaging the proxy and the scaled modelled temperatures at the two sites.

The values of $s$ for each modelled best-fit change to the proxy SATs are shown in Table 3. When comparing models and proxies, it is informative to consider what may be called a "good agreement", and to provide a point of reference for assessing the skill scores. As such, in Table 3 we also show the skill score that would be obtained in the case of an idealised model simulating (i) no SAT change across the EOT, (ii) a global mean change that best fits the data, and (iii) a zonal-mean change of the form $\Delta SST = A + B\cos(\phi)$, (where $\phi$ is latitude), that best fits the data.


| Model | s for best-fit $\Delta T_{CO2}$ | s for best-fit $\Delta T_{ice}$ | s for best-fit $\Delta T_{geog}$ |
|---|---|---|---|
| CESM_B | 0.278 | | |
| CESM_H | 0.284 | 0.524 | 0.546 |
| FOAM | 0.278 | 0.492 | 0.402 |
| GFDL | 0.280 | | 0.546 |
| HadCM3BL | 0.311 | 0.458 | 0.546 |
| NorESM-L | 0.312 | | 0.546 |
| UVic | | | 0.537 |
| Ensemble mean | 0.274 | 0.478 | 0.546 |
| | s for idealised $\Delta T$ | | |
| No change | 0.546 | | |
| Constant change | 0.296 | | |
| cos($\phi$) change | 0.275 | | |

**Table 3:** Skill scores, s, for the best-fit modelled changes in response to $CO_2$, ice, and paleogeographic forcing, for each model and for the ensemble mean (a lower value of s represents a better fit to data). Also shown are the values of s for three idealised SAT changes. The models all achieve their best skill performance with $CO_2$ forcing (UVic does not include $CO_2$ forcing). Three models (CESM_B, FOAM and GFDL) achieve a better skill than an idealised constant temperature change, while CESM_B and the ensemble mean achieve better than the idealised cos(ø) case. However, the spread in skill across the different models is narrow. Changes highlighted in green are better than or equal to the idealised constant change case, while the ensemble mean is better than the cos(ø) case.

It is clear from Table 3 that the best modelled fit to the SAT proxy data arises from changes to $CO_2$. In particular, the ensemble mean response to a decrease in atmospheric $CO_2$ has the best (lowest) skill score, and performing slightly better than a cos(ø) fit. The only individual model that outperforms the cos(ø) fit is CESM_H, when including both $CO_2$ and ICE forcing changes (Table 4). We note however, that a cos(ø) fit to the data produces only a 7% improved skill score over the constant change fit. The CESM_B, CESM_H, FOAM and GFDL models perform somewhat better than the constant change fit to the data, while HadCM3BL and NorESM-L all perform slightly worse than this, but their skill scores are within a margin of ~5% of the constant change fit. The $CO_2$ change provides by far the best temperature fit over the ICE and GEOG changes. The UVic model does not apply a $CO_2$ change, and consequently achieves a poorer skill score. We note however, that the ICE changes improve the skill score in the CESM_H and HadCM3BL models, while the GEOG changes improve the skill score in the FOAM, NorESM-L and UVic models. Since those forcing factors are independent to $CO_2$, they also improve the overall skill score in combination, as we show below. These results broadly agree with a recent Southern Ocean-only model-data comparison, which showed that $CO_2$ forcing provided the best explanation of temperature changes across the EOT, with secondary improvements made from ice and paleogeography changes (Kennedy-Asser et al., 2020).

### 7.2.2 Mechanisms of change

The above analysis implies that the change in SST at the EOT can be best explained by a decrease in $CO_2$, as opposed to changes in ice or paleogeography. However, it is possible that changes in ice or paleogeography, combined with $CO_2$ change, may fit even better with the data. To test this possibility, we assume that the various responses add together linearly, and find the best scaled combination of each mechanism, i.e. we find α, β, γ such that the skill score, s, of $(\alpha\Delta T_{CO2} + \beta\Delta T_{ice} + \gamma\Delta T_{geo})$ is minimised. The result of this exercise for each model and for the ensemble mean is shown in Table 4. This shows that CESM_H and HadCM3BL achieve a better fit to the data when including the full response to ice sheet change and no paleogeographic change, while FOAM and NorESM-L achieve a better fit when including a paleogeographic change and no ice sheet change. The ensemble mean agrees best with the proxies when incorporating a $CO_2$ shift of 885 to 560 ppm (α=0.66), with the GEO forcing providing further improvement to the fit (γ=0.12) while the ICE forcing does not improve the ensemble mean skill score. This best-fit ensemble mean change is shown in Figure 8. Given the close agreement between the models in fitting a $CO_2$ change to the data, we can estimate from the full model spread that the $CO_2$ drop was by a factor of 1.58 ± 0.15. If for example we assume an Oligocene $CO_2$ value of 2x pre-industrial levels, the $CO_2$ drop would be from 885 ± 90 ppmv to 560 ppmv. However, we would caution that this estimate reflects the model spread in matching this particular set of data, and

that the true uncertainty is larger. Additionally, the models that included changes due to ice sheet forcing or paleogeographic forcing achieved some improvement in fitting the data, but this played a lesser role than $CO_2$ forcing, as measured by these skill metrics.

A 325 ppmv decrease is within the range of $CO_2$ proxy estimates, shown in Figure 5, with alkenone records in particular suggesting a drop of this magnitude across the EOT (Pagani et al., 2011; Zhang et al., 2013). Boron and stomatal records indicate that such a change is plausible but is likely of lesser magnitude. The recent multi-proxy compilation of Foster et al. (2017) uses a smoothed regression to derive 'best fit' $CO_2$ estimates of 893 ppmv and 806 ppm during the late Eocene (38-34 Ma) and early Oligocene (33.5-30 Ma) respectively, or a decrease of ~10%. Thus, our model-derived $CO_2$ decrease is likely

to be an overestimate of the change across the EOT. There are several reasons why this mismatch may occur in our model ensemble, which we discuss in Section 7.2.3.

| Model | s for best-fit $\alpha\Delta TCO2 + \beta\Delta Tice + \gamma\Delta Tgeo$ | $\alpha$ [CO2 change ppmv] | $\beta$ | $\gamma$ |
|---|---|---|---|---|
| CESM_B | 0.278 | 0.70 [910 to 560] | | |
| CESM_H | 0.270 | 0.56 [826 to 560] | 0.90 | 0 |
| FOAM | 0.277 | 0.60 [849 to 560] | 0 | 0.30 |
| GFDL | 0.280 | 0.52 [803 to 560] | | 0 |
| HadCM3BL | 0.307 | 0.56 [826 to 560] | 0.61 | 0 |
| NorESM-L | 0.311 | 0.66 [885 to 560] | 0 | 0.25 |
| UVic | 0.537 | | | 0.68 |
| Ensemble mean | 0.273 | 0.66 [885 to 560] | 0.0 | 0.12 |
| | s for idealised $\Delta T$ | | | |
| No change | 0.546 | | | |
| Constant change | 0.296 | | | |
| cos(ø) change | 0.275 | | | |

Table 4: Skill scores, s, for the best-fit modelled changes in response to a combination of $CO_2$, ice, and paleogeographic forcing, for each
1300 model and for the ensemble mean (a lower value of s represents a better fit to data). Also shown are the values of $\alpha$, $\beta$, and $\gamma$ that give the best fit, and the $CO_2$ change corresponding to $\alpha$, assuming a post-EOT value of 560 ppmv. Also shown are the values of s for three idealised SAT changes. Changes highlighted in green are better than or equal to the idealised constant change case, while CESM_H and the ensemble mean achieve better than the idealised cos(ø) case.

# Best Fit to EOT data

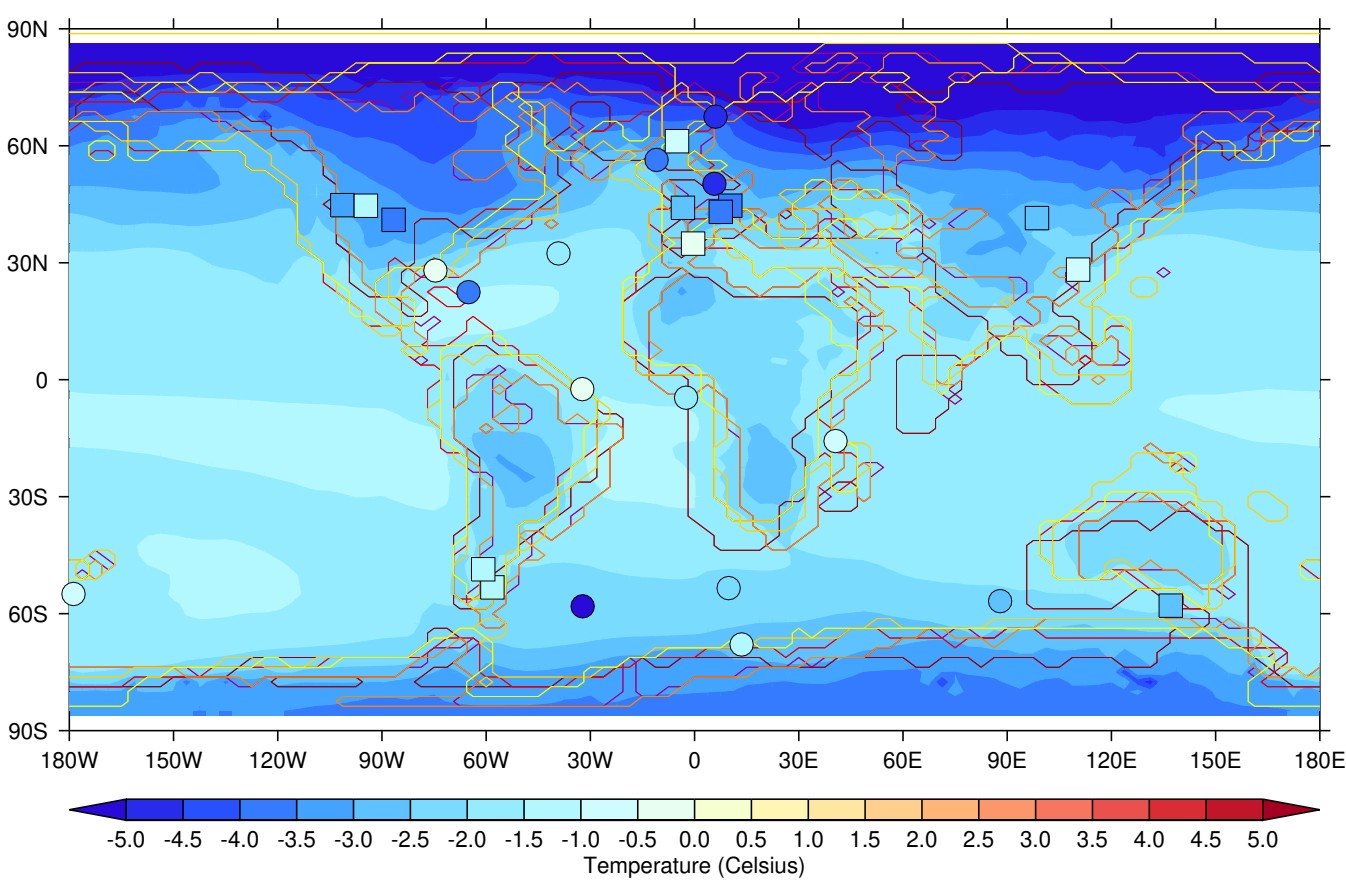

**Figure 8:** Ensemble mean modelled SAT response to a $CO_2$ decrease from 885 to 560 ppmv, representing the best fit to the proxy data. The marine proxy data are shown as filled circles, while the terrestrial proxy data are shown as filled squares. Coastlines from each model are plotted to illustrate the uncertainties associated with the paleogeographic reconstructions.

### 7.2.3 Uncertainties associated with the modelling

There are several uncertainties that should be considered when interpreting the results above. Some of these are discussed here. There is uncertainty in the models themselves. These models could be characterised as AR4-class or even TAR-class in that they were state-of-the-art at the time of the 4[th] or 3[rd] IPCC assessment report, as opposed to the most recent AR5, or the upcoming AR6. The use of less complex models can be an advantage for deep-time paleoclimate work, as these models allow greater length of simulation, which is especially important for the deep ocean where the initial condition may be far from the equilibrium state, which is unknown at the start of the simulation. However, there is a trade-off between simulation length and model complexity, and some of the model simulations presented here are relatively short (e.g. HadCM3BL; Table 2). A potential manifestation of this lack of complexity relates to the modelled change in land-sea contrast. The EOT temperature change from marine records is slightly larger (-2.5 °C) than that recorded from land temperature proxies (-2.3 °C). This in part

reflects the heterogeneous pattern of changes in plant species, from which the land temperature proxies are derived, but is in general a globally robust signal. On the other hand, the temperature changes recorded in the model simulations show the opposite; land temperature changes are more sensitive to greenhouse cooling than ocean temperature changes. This makes it challenging to achieve a close fit to all of the temperature records available.

There are several possible reasons why our model-derived $CO_2$ decrease is larger than the 'best fit' to the Foster et al (2017) $CO_2$ proxy compilation. First, the models may be under-sensitive to $CO_2$ forcing, as has been suggested in previous attempts to model the Eocene (Huber and Caballero, 2011; Lunt et al., 2012). The model ensemble climate sensitivity to doubling $CO_2$ is 3.3°C. A recent synthesis of Paleocene-Eocene proxy records suggests that climate sensitivity during the lateset Paleocene, PETM and EECO was 4.5°C, 3.6°C and 3.1°C respectively (Inglis et al., 2020). In addition the 66% confidence interval of climate sensitivity was ~2-7°C over the three intervals combined. Thus, our model ensemble compares well with the best estimates of the early Eocene climate sensitivity, but is towards the lower end of the 'likely' confidence interval, suggesting a considerably higher climate sensitivity is possible. Second, the models do not permit dynamic feedbacks between $CO_2$ forcing and the growth of an ice sheet, which in reality would cause ice-albedo, land-albedo, and cloud radiative forcing feedbacks. These missing feedbacks may amplify the global temperature change when crossing a critical threshold of glaciation. Since these models prescribe their ice sheets as on or off, such feedbacks are not possible here. However, it is still useful to examine which forcing mechanisms provide the best explanation of the proxy record. In this sense, it is notable that $CO_2$ forcing provides the best explanation of the far-field cooling away from Antarctica, while imposing an ice sheet has a lesser effect on the global mean (Goldner et al., 2014).

The boundary conditions applied to the models is a large source of uncertainty. The spread in paleogeographic forcing between models is testament to the fact that the paleogeography both pre-EOT and post-EOT is relatively uncertain (See Section 2.1 and 2.2). Recent climate model simulations of the early Eocene suggest that the paleogeographic forcing plays a similarly important role as $CO_2$ forcing in capturing Eocene warm climates (Lunt et al., 2020). These simulations suggest that 3-5°C of the warming from pre-industrial to Eocene climates comes from non-$CO_2$ forcing, i.e. forcing from changes to geography, land surface properties and the removal of ice sheets. Therefore, while increasing the climate sensitivity may be part of solving the Eocene model-data mismatch, improving the paleogeographic boundary conditions is also a high priority for obtaining a better model-data agreement. The proxy dataset we have used is also limited in areal coverage. As such, the models remain untested in several key regions, such as the North and tropical Pacific, or the Indian Ocean. New data in the regions predicted to warm in response to a $CO_2$ drop in HadCM3BL and NorESM-L would be particularly useful for discriminating between models. The model-data comparison does not consider any uncertainty in the proxy estimates themselves. In reality, each site is associated with a different and substantial uncertainty in its estimate of SST change across the EOT.

Here we have considered only the change in temperature across the EOT, rather than absolute temperatures. However, this can mask biases in the simulated base state of the pre-EOT and post-EOT simulations. Not all models do a good job of simulating the base state (see e.g. discussion of meridional temperature gradients in Section 6.2). Although the ensemble results achieve the best fit from $CO_2$ forcing (with a small contribution form ice sheet forcing), it is worth noting that the ultimate cause of the $CO_2$ drop itself remains unclear; it could itself be driven by changes to geological sources and sinks, changes in weathering rates, or feedbacks associated with ocean and land sinks due to circulation changes (see Section 6.3).

## 8 Conclusions

Earth's modern 'icehouse' climate is defined by the presence of significant ice masses on land at both poles but that ice is now retreating. Understanding the drivers and scale of polar ice growth during the initial inception of this icehouse at the EOT, which involved cooling and glaciation under a yet still warm Eocene-Oligocene climate, can provide crucial insights into ice sheet stability and behaviour under a warm climate, something that is more critical than ever. Here, we have reviewed current literature regarding the EOT, in terms of stratigraphic definitions, geological records of paleogeographic and Earth system change, and modelling insights into mechanisms of change. Marine records currently provide the most extensive global record of temperature change across the EOT, with a SST cooling found across most regions. The marine records suggest a global average temperature change of approximately -2.5 °C across the EOT, although individual records range from approximately 0 to -8 °C change. Terrestrial records of temperature change are more geographically limited, with a concentration of records in the midlatitude Northern Hemisphere, and very little coverage of the Southern Hemisphere. The terrestrial records average change is -2.3 °C, but like the marine records, the change recorded at individual locations ranges from approximately 0 to -8 °C across the EOT. Records of $CO_2$ across the EOT also indicate some differences between marine and terrestrial records. Marine records suggest a higher concentration of $CO_2$ overall and a clearer transition towards lower $CO_2$ across the EOT. Terrestrial records, by contrast, indicate a lower $CO_2$ concentration overall and a more gradual $CO_2$ decline with no obvious shift coinciding with the EOT. There is an ongoing need to reconcile the different $CO_2$ signals found in the marine and terrestrial realms.

A new model-data comparison presented in this paper reveals that a halving of atmospheric $CO_2$ across the EOT has a substantially greater effect on global mean and regional SAT than either the onset of Antarctic glaciation or changes in paleogeography. The response to $CO_2$ forcing is robust across models, with cooling increasing towards higher latitudes, helping to explain high latitude cooling in the marine records. While individual models achieved a better fit to the data by including paleogeographic forcing and ice sheet forcing, these changes are more variable across the models. As a result, the best fit in the ensemble mean is dominated by decreasing $CO_2$ by a factor of 1.58x, with ice sheet forcing and paleogeography forcing playing a secondary role. Assuming an Oligocene value of 560 ppmv, the corresponding pre-EOT value is 885 ppmv. However, we do not exclude the importance of contributions from other forcings. Indeed, two models in the ensemble achieve their best

fit to the temperature records with a combination of $CO_2$ and ice-sheet induced changes, while two models show an improved

fit when paleogeographic changes are combined with $CO_2$ forcing. Paleogeographic changes and ice sheet feedbacks are inherently regional and harder to aggregate across different model experiments. Nevertheless, it remains possible that gateway-induced ocean circulation change is somehow implicated in $CO_2$ decline. For a more complete understanding of these feedbacks, future climate modelling of the EOT must incorporate dynamic feedbacks between these different forcing factors.

**Appendix – List of Acronyms**

ACC: Antarctic Circumpolar Current

AMOC: Atlantic meridional overturning circulation

CCD: Calcite compensation depth

CMMT: cold month mean temperature

CLAMP: Climate Leaf Analysis Multivariate Program

EAIS: East Antarctic ice sheet

EOGM: Early Oligocene glacial maximum

EOB: Eocene-Oligocene boundary

EOT: Eocene-Oligocene transition

EOIS: Early Oligocene oxygen isotope step

GSSP: Global boundary stratotype section and point

MAT: mean annual temperature

MECO: Middle Eocene climatic optimum

NLR: Nearest Living Relative

ODP: Ocean drilling program

PrOM: Priabonian oxygen isotope maximum

SAT: Surface air temperature

SST: Sea surface temperature

**Data Availability**

The data compilation plotted in the manuscript are included in the supplementary material. The model data are available upon

request from the corresponding authors listed in Table 2.

**Author Contributions**

The manuscript was conceived and planned by DKH, AMdB, HKC, and MS, in collaboration with all authors. Section 1 was led by HKC, PNP and PAW. Section 2 was led by DKH, MB, HDS and AvdH. Section 3 was led by KKS, HKC and CHL. Section 4 was led by MS, MJP and LK, with contributions from KM. Section 5 was led by MS, CHL and LK. Section 6 was

led by DKH and AMdB. Section 7 was led by DJL and DKH. Compilations of SST records were made by KKS and MH, terrestrial records by MJP, US and MH, coordinated by DKH. $\delta^{18}O$ records were compiled by EP. Model simulation data were contributed by DKH, ATK-A, J-BL, MB, MH, WPS, ZZ. All authors contributed to editing and review of the manuscript.

**Competing interests**

The authors declare that they have no conflict of interest.

**Acknowledgments**

This work originated from a workshop on the Eocene-Oligocene Transition in Stockholm in February 2017, funded by the Bolin Centre for Climate Research, Research Area 6. This work was also supported by the Swedish Research Council project 2016-03912 and FORMAS project 2018-01621. KKS acknowledges financial support from by the Danish Council for Independent Research – Natural Sciences (DFF/FNU; grant no. 11-107497).

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
