# Peer review of "The Eocene-Oligocene transition: a review of marine and terrestrial proxy data, models and model-data comparisons"

_Climate of the Past, 2020_

## Referee Comment (RC1) · Anonymous Referee #1 · 13 Jun 2020

General Comments

This manuscript, entitled "The Eocene-Oligocene transition: a review of marine and terrestrial proxy data, models and model-data comparisons" by Hutchinson et al., is an outstanding review paper worthy of publication in Climate of the Past. This paper makes large strides and provides a comprehensive update to Eocene-Oligocene research as much has been done since the 2007 review of Coxall and Pearson. The authors take a methodical approach, first addressing terminology and a framework for the Eocene-Oligocene, then boundary conditions such as paleogeographic reconstructions, paleoceanography, and constraints on glaciation. Next, the authors move

to a comprehensive review of marine and terrestrial paleoclimate and pCO2 reconstructions, mechanistic modeling studies of paleogeographic, CO2 and temperature changes. Finally, and most importantly, the authors end the paper with a metaanalysis of the factors affecting paleoenvironmental changes during the E-O transition and conclude that CO2 decrease likely served as the primary driver of cooling and Antarctic glaciation. As this manuscript is very well written, I do not have large structural comments. Instead, I include my few specific comments along with line-by-line technical corrections below.

Specific Comments and Technical Corrections

Lines 111 and 125: This is just one example of "Fig." vs "Figure." Be consistent throughout.

Lines 111 and 123: Again, just an example...I think it may help to structure these paragraphs in a way that really highlights these key terms such as EOGM and EOIS. This reorganization could help readers not as familiar with key E-O events as this is a review intended for a broad audience.

Figure 1. Along with the comment above, perhaps it would help to differentiate old/existing terms with your new preferred framework. Possibly a different colored font for existing terms e.g. Step 1 vs your new proposed terms e.g. EOIS.

Line 199. Isotope

Lines 202 to 205. Possibly one more sentence to explain the mammalian phenomenon

Figure 3 caption. UK37 formatting.

Line 527. Possibly one additional sentence to discuss the d18Osw ice volume assumption and its robustness

Line 645: Revise "multiple evidence"

Line 657: UK37 formatting

Section 5.2. Possibly a short discussion of challenges with leaf proxies (e.g. preservation bias, sampling bias) would supplement the existing discussion of challenges with chronology in terrestrial records in general.

Line 920: The CCD was introduced on line 421 and should be abbreviated there.

Lines 927-930: This sentence is very jargony and should be revised (and probably split into 3 or so sentences). At the very least, "glacio-eustatic sea level-led shelf-to-basin fractionation" and "C12-enriched carbon capacitors" should be explained.

Line 929: "C12-enriched" formatting

Line 930: clathrates

Lines 931-935: These mechanisms require additional explanation and reduced jargon. Explain "labile and refractory components"

Line 955: This description of GENIE is very similar to line 932

Line 968: Define SAT here and not in line 971.

Figure 7. This figure is perhaps the most important in the paper and possibly the hardest to interpret. At the very least, when finally published, these three panels should be much larger. Many of the other figures can be smaller. I found 7a particularly hard to interpret, but the coastlines and proxy-model comparisons in all three panels are a bit challenging to see.

Line 1082: For UVic,

Line 1085: For FOAM,

Table 2 caption: This caption requires the statement as in Table 3 about the meaning of green highlighted cells

Line 1164: This is the first of several instances where "CO2 forcing alone" is cited as the best fit. Perhaps I'm not understanding the skill score analysis, but in Table 3 it

appears that the ensemble mean skill score of 0.326 is achieved with coefficients of 0.7, 0.06 and 0.26 for alpha, beta and gamma respectively. Wouldn't this mean that $CO_2$ is the primary driver but that ice and paleogeography make modest contributions as well? If this is the case, the authors should clarify modify their interpretation to include these additional factors. If this is not the case, a quick mention of why this result indicates $CO_2$ is the sole driver would be helpful.

Figure 8 caption: 910 ppm, not 900 right?

Line 1217: Here,

Line 1229: There is therefore a . . . . . .

---

## Referee Comment (RC2) · Anonymous Referee #2 · 2 Jul 2020

This paper provides a comprehensive review of the current state of understanding of climate change during the Eocene-Oligocene transition, and in the process, attempts to assess the cause of this change, CO2 versus paleogeography, the added role of ice. Published observations from marine and terrestrial archives are compiled and compared against climate simulations from a collection of modeling studies (i.e., a model inter-comparison). The observations include SAT (SST), SSS, continental ice extent, sea-ice and ocean circulation. The comparisons are for 2 broad intervals, ~late Eocene and early Oligocene. Presumably, the observations are binned over long (»1-4 my?) windows. For the modeling studies, the boundary conditions, including ice sheets, are generally similar though not identical given the lack of coordination, and each of the

models are run to equilibrium (∼thousands of years). The study looks at the simulated responses to changes in paleogeography (i.e., gateways), GHG levels, and Antarctica ice-sheets. For GHG levels, 900 ppm (just above the threshold for ice accumulation on Antarctica,) is used at the Eocene pCO2 and 560 ppm for the Oligocene. Because the published model experiments were not coordinated and thus run with a range of CO2, for comparison the output of some models are scaled to approximate the same ∆pCO2. The equilibrium climate states for each model and an ensemble are then assessed for a best fit with observations. Given differences in resolution and other parameters, the absolute T in the models vary widely, with a cool group and warm group, so the focus is primarily on ∆SAT. In general, most of the models are showing ∼2°C cooling in mean global SAT (figure 5). There are minor regional discrepancies which are attributed primarily to differences in ocean mixing/heat transport. In the end, the ensemble is deemed to show a good fit with observations supporting the conclusion that a reduction in atmospheric CO2 was the primary driver of the EOT.

I have mixed feelings about this paper. Clearly a considerable amount of time and effort went into compiling the observations, the synthesis of modeling work. This alone will be a valuable contribution to the EOT literature. However, I am not convinced that the modeling comparison is a useful exercise, at least not in the way it was intended or designed.

The main issue concerns the finding/conclusions about the forcing behind the transition, specifically a ∼halving of pCO2 (from 900 to 560 ppmv) with the EOT. To my knowledge, based on observations (see figure 5) or theory, there is no basis for a 40 to 50% reduction in atmospheric CO2 across this transition. The existing B isotope data, albeit sparse, even suggests a slight increase, and the alkenones suggest a decline but not nearly of that magnitude. More importantly, just from a purely theoretical perspective, there is no reason to expect such a large decline. Recall that when the first detailed, high resolution benthic O isotope records were produced, it became clear that the EOT, or at least the appearance of continental ice-sheets on Antarctica was
relatively abrupt, consistent with the threshold hypothesis; a relatively small drop in GHG would be sufficient to trigger the rapid accumulation of ice on a polar continent (i.e., climatic threshold/tipping point, e.g., Crowley and North, 1988). This concept was reinforced by the ice-sheet modeling of DeConto and Pollard 2004 which demonstrated how the local albedo feedback on summertime T could accelerate ice accumulation. Granted that by today's standards the ice-sheet model of the D/P study was relatively coarse and simplistic, but the general theory of a bifurcation point in the climate system still seems valid. We can debate the exact magnitude of the $CO_2$ drop required, but it was probably small ($\sim$ 100 ppm), especially with the proper orbital configuration. And even with the large uncertainties, the $CO_2$ proxies are consistent with this hypothesis. This is the most compelling and important aspect of the EOT, a relatively large change in climate in response to a relatively small change in forcing. The observations of a few degrees of cooling, switches in the mode of ocean circulation are consistent with this hypothesis. The Goldner et al (2014) paper nicely illustrates the regional/global effects on ocean T of just adding the ice-sheet (w fixed $pCO_2$). Also, why such a rapid and large reduction in $pCO_2$ at that time? Positive feedbacks involving biogeochemical cycles, ocean uptake, could potentially draw down $CO_2$ but the effect would likely be relatively small, <100 ppm, as suggested by a variety of modeling studies (& observations). More than likely, the decline in $pCO_2$ from the latest Eocene to earliest Oligocene was probably minor.

With all this in mind, the fact that the model ensemble is in agreement with observations is problematic. In other words, to match climate observations, a much larger change in forcing (1.6x) is required than justified by observations or theory. This is the same recurring issue with simulating high $CO_2$ climates of the past, that a much larger forcing is required than justified by observations? The bottom line is that the models are under sensitive to GHG forcing. Arguably, this should be one of the main conclusions of this paper. As the paper is currently written, I almost get the opposite sense.

Has this paper achieved the stated goal of identifying what drove the EOT, at least the

abrupt appearance of ice-sheets ~34 Mya? Am sure we can all agree that over the long-term, a reduction in GHG was the primary driver of Eocene cooling and key to triggering Antarctic glaciation. What we won't agree on are the specifics of timing and magnitude, the appropriate alignment of GHG forcing with the climate response.

Recommendation: A key question for revision - what is the real purpose of this paper? If it's simply to provide a comprehensive review of the existing literature on observations and modeling of the EOT, recommendations for future research, minor/moderate revision (see comments below) would suffice. For the reasons stated above, the data-model comparison (section 7) could be dropped. If retained, it would be essential to include a discussion of the caveats; the aforementioned mismatch between observations/models, implications for model sensitivity and feedbacks, and recommendations for setting up future experiments.

Additional Comments/Recommendations:

Window of Observations (Apples vs. Oranges); Considerable effort is spent in defining the duration of the EOT, "Hence the stratigraphic interval of the EOT according to our preferred definition is now given an estimated duration of 790 kyr (fig 1)". The problem is that the collected observations (Figures 3-5) span a much wider range of time, millions of years of the late Eocene/early Oligocene. I assume this is by necessity, especially with the inclusion of terrestrial climate proxies. However, to make the model-data comparison more meaningful, it would be best to only include climate observations that straddle the O isotope excursion at 34 Ma, lets say within windows of 500 kyr immediately above and below. This might exclude a lot of data but the comparison pre- and post EOT conditions would be more meaningful.

1065 -1067 A figure showing the change in sea-ice distribution would be useful.

7.1.3 SAT response to paleogeographic change, ΔTGEO I think the assessment of simulations with differing tectonic configurations, ΔTGEO, is useful only for assessing how the sensitivity of a model to a given change in GHG levels varies under different

configurations, e.g., an open or closed Drake passage. We already know from previous studies that the geographic changes alone produce relatively minor changes in global climate, and sometimes in the wrong direction. Also, the uncertainties about the timing of the gateways are large enough that this is not really worth focusing on. The section should be condensed or simply moved to SOM.

1121-22 yes, the CO2 is somewhat arbitrary.

1124 - As we all know, the error in CO2 reconstructions is quite large. Nevertheless, it is unlikely that CO2 was halved pre-EOT to EOT. More likely, the change in pCO2 was much smaller, at least initially with inception of glaciation (at 34 Ma) which involved a threshold CO2 enhanced by regional feedbacks via ice-sheet growth. It is possible that biogeochemical feedbacks enhance the drawdown of CO2, but probably not more than 100 ppm or so.

1165 – Based on the goodness of fit, you derive an estimate of $\Delta$CO2? I understand the strategy here, but it seems backwards when the primary motivation behind reconstructing paleoclimates is to assess climate models. As stated above, I think the observations suggest that the models (as they were prior to 2017) are under sensitive to GH forcing.

Figure 5 (pCO2 reconstruction) – This figure made me cringe. The terrestrial proxy pCO2, given the coarse stratigraphic control, low temporal resolution, could be misleading. And let's be honest, given the concerns about the reduced sensitivity of the stomata proxy to higher CO2, who would really expect that proxy to accurately capture the $\Delta$pCO2 across the EOT? Lets not even get into the issues with soil carbonates. As most of the climate data are marine based, it would make sense to only include the pCO2 estimates from marine proxies plotted along with the benthic d18O of figure 2 over a narrower window of time. This would eliminate any uncertainties about the relative timing of changes in climate versus forcing.

---

## Author Comment (AC1) · 29 Aug 2020

We thank the reviewer for their positive feedback and constructive comments on the manuscript. Here we outline our proposed response to each comment in blue text.

Kind Regards,
David Hutchinson

**General Comments**

This manuscript, entitled "The Eocene-Oligocene transition: a review of marine and terrestrial proxy data, models and model-data comparisons" by Hutchinson et al., is an outstanding review paper worthy of publication in Climate of the Past. This paper makes large strides and provides a comprehensive update to Eocene-Oligocene research as much has been done since the 2007 review of Coxall and Pearson. The authors take a methodical approach, first addressing terminology and a framework for the Eocene-Oligocene, then boundary conditions such as paleogeographic reconstructions, paleoceanography, and constraints on glaciation. Next, the authors move to a comprehensive review of marine and terrestrial paleoclimate and pCO2 reconstructions, mechanistic modeling studies of paleogeographic, CO2 and temperature changes. Finally, and most importantly, the authors end the paper with a metaanalysis of the factors affecting paleoenvironmental changes during the E-O transition and conclude that CO2 decrease likely served as the primary driver of cooling and Antarctic glaciation. As this manuscript is very well written, I do not have large structural comments.

Thank you for the positive overall assessment.

Instead, I include my few specific comments along with line-by-line technical corrections below.

**Specific Comments and Technical Corrections**

Lines 111 and 125: This is just one example of "Fig." vs "Figure." Be consistent throughout.

All instances have been changed to "Figure".

Lines 111 and 123: Again, just an example: I think it may help to structure these paragraphs in a way that really highlights these key terms such as EOGM and EOIS. This reorganization could help readers not as familiar with key E-O events as this is a review intended for a broad audience.

We have re-organised these paragraphs to set out the terms more clearly.

Figure 1. Along with the comment above, perhaps it would help to differentiate old/existing terms with your new preferred framework. Possibly a different colored font for existing terms e.g. Step 1 vs your new proposed terms e.g. EOIS.

We have now highlighted the EOIS term in blue text, since it is a new term that we define in this manuscript. All other terms in the Figure are defined previously and left in black text. We have also added a new Table 1 listing and defining terms and differentiating old/existing terms in an easily accessible form.

Line 199. Isotope

This has been fixed.

Lines 202 to 205. Possibly one more sentence to explain the mammalian phenomenon

We have added an extra sentence and minor edits to explain the mammalian evolutionary turnover.

Figure 3 caption. UK37 formatting.

This formatting has been corrected.

Line 527. Possibly one additional sentence to discuss the d18Osw ice volume assumption and its robustness

This sentence has been re-ordered to come after the discussion of inter-basin similarity (previously Lines 531-533), and added a further comment on the assumption's limitations.

Line 645: Revise "multiple evidence"

This is now "multiple lines of evidence".

Line 657: UK37 formatting

This has been fixed.

Section 5.2. Possibly a short discussion of challenges with leaf proxies (e.g. preservation bias, sampling bias) would supplement the existing discussion of challenges with chronology in terrestrial records in general.

We have added a short paragraph on the challenges of leaf proxy estimates of CO2.

Line 920: The CCD was introduced on line 421 and should be abbreviated there.

We have used to the abbreviation here.

Lines 927-930: This sentence is very jargony and should be revised (and probably split into 3 or so sentences). At the very least, "glacio-eustatic sea level-led shelf-to-basin fractionation" and "C12-enriched carbon capacitors" should be explained.

This part has been revised with improved explanations each process and simpler terms:

Carbon cycle box models suggest that the best fit to observations is achieved by a shift from shelf-to-basin carbonate fractionation (Armstrong McKay et al., 2016; Merico et al., 2008). In this interpretation of events, the fall in sea-level due to Antarctic glaciation (i) reduces the global flux of carbonate into shallow water (reef, bank and shelf) sediments; and (ii) exposes fresh, readily dissolved shelf carbonate sediments around the world to rapid subaerial weathering (Merico et al., 2008). The first of these two mechanisms drives the sustained CCD deepening from Eocene to Oligocene. The second mechanism drives a one-off dump of carbonate into the ocean that explains the initial transient overshoot behaviour (Zachos and Kump, 2005), and the transient increase in benthic $\delta^{13}$C occurs because the shelf carbonate reservoir is enriched in $^{13}$C relative to pelagic carbonate reservoir (Swart and Eberli, 2005; Swart, 2008; Merico et al., 2008; Armstrong McKay et al., 2016). If the isotopic fractionation between these two carbonate sediment reservoirs is modest, however, shelf-basin fractionation can only fully explain the transient increase in oceanic $\delta^{13}$C if the one-off dump of weathered shelf carbonate is questionably large (Merico et al., 2008). In their follow up study, Armstrong McKay et al. (2016) considered this problem in detail and concluded that, unless shelf carbonates were substantially enriched in $^{13}$C relative to pelagic carbonates (by $\sim$3‰), an additional process must also have contributed, with sequestration of $^{12}$C-enriched carbon into carbon capacitors, and possibly increased ocean ventilation, offering the best fit to the paleorecords when combined with shelf basin fractionation.

Line 929: "C12-enriched" formatting

This term has been reformatted correctly.

Line 930: clathrates

This has been fixed.

Lines 931-935: These mechanisms require additional explanation and reduced jargon. Explain "labile and refractory components"

We now provide definitions of labile and refractory carbon and their relevance to the mechanisms discussed, as follows:

"They suggest several mechanisms are needed to explain the CCD change in addition to the shelf-basin fractionation hypothesis above: (i) perturbations to continental weathering and solute input to the deep ocean, or (ii) changes in the partition of organic carbon flux between labile (organic carbon that is readily available for oxidation and driving carbonate dissolution) and refractory (carbon that is more resistant to degradation and largely preserved and buried) components."

Line 955: This description of GENIE is very similar to line 932

We have removed the duplicate explanation of cGENIE.

Line 968: Define SAT here and not in line 971.

This has been done.

Figure 7. This figure is perhaps the most important in the paper and possibly the hardest to interpret. At the very least, when finally published, these three panels should be much larger. Many of the other figures can be smaller. I found 7a particularly hard to interpret, but the coastlines and proxy-model comparisons in all three panels are a bit challenging to see.

We agree – in journal form this figure would have been hard to read. For the sake of clarity, we have removed the stippling, and changed the colouring of the coastlines to make them more visible. We have also removed the numbers written adjacent to the proxy locations, since they were difficult to read, and the proxy values are already presented in table form.

Line 1082: For UVic,

The comma has been added.

Line 1085: For FOAM,

The comma has been added.

Table 2 caption: This caption requires the statement as in Table 3 about the meaning of green highlighted cells

We have added the explanation of green highlighted cells.

Line 1164: This is the first of several instances where "CO2 forcing alone" is cited as the best fit. Perhaps I'm not understanding the skill score analysis, but in Table 3 it appears that the ensemble mean skill score of 0.326 is achieved with coefficients of 0.7, 0.06 and 0.26 for alpha, beta and gamma respectively. Wouldn't this mean that

CO2 is the primary driver but that ice and paleogeography make modest contributions as well? If this is the case, the authors should clarify modify their interpretation to include these additional factors. If this is not the case, a quick mention of why this result indicates CO2 is the sole driver would be helpful.

We agree with the interpretation in this comment and have removed the references to "CO2 forcing alone". CO2 is the primary (but not the sole) driver according to our analysis and we have clarified this.

Figure 8 caption: 910 ppm, not 900 right?

Yes, this has been corrected.

Line 1217: Here,

The comma has been added.

Line 1229: There is therefore a ... :

The sentence has been completed.

**References**

Armstrong McKay, D. I., Tyrrell, T., and Wilson, P. A.: Global carbon cycle perturbation across the Eocene-Oligocene climate transition, Paleoceanography, 31, 311–329, https://doi.org/10.1002/2015PA002818, 2016.

Merico, A., Tyrrell, T., and Wilson, P. A.: Eocene/Oligocene ocean de-acidification linked to Antarctic glaciation by sea-level fall, Nature, 452, 979–982, https://doi.org/10.1038/nature06853, 2008.

Swart, P. K.: Global synchronous changes in the carbon isotopic composition of carbonate sediments unrelated to changes in the global carbon cycle, Proceedings of the National Academy of Sciences, 105, 13 741 – 13 745, https://doi.org/10.1073/pnas.0802841105, 2008.

Swart, P. K. and Eberli, G.: The nature of the $\delta$13C of periplatform sediments: Implications for stratigraphy and the global carbon cycle, Sedimentary Geology, 175, 115–129, https://doi.org/10.1016/j.sedgeo.2004.12.029, 2005.

Zachos, J. C. and Kump, L. R.: Carbon cycle feedbacks and the initiation of Antarctic glaciation in the earliest Oligocene, Global and Planetary Change, 47, 51–66, https://doi.org/10.1016/j.gloplacha.2005.01.001, 2005.

---

## Author Comment (AC2) · 29 Aug 2020

We thank the reviewer for their thoughtful and constructive comments on the manuscript. Here we outline our proposed response to each comment in blue text.

Kind Regards,
David Hutchinson

[Figure]

**General Comments**

This paper provides a comprehensive review of the current state of understanding of climate change during the Eocene-Oligocene transition, and in the process, attempts to assess the cause of this change, CO2 versus paleogeography, the added role of ice. Published observations from marine and terrestrial archives are compiled and compared against climate simulations from a collection of modeling studies (i.e., a model inter-comparison). The observations include SAT (SST), SSS, continental ice extent, sea-ice and ocean circulation. The comparisons are for 2 broad intervals, late Eocene and early Oligocene. Presumably, the observations are binned over long (1-4 my?) windows. For the modeling studies, the boundary conditions, including ice sheets, are generally similar though not identical given the lack of coordination, and each of the models are run to equilibrium (thousands of years). The study looks at the simulated responses to changes in paleogeography (i.e., gateways), GHG levels, and Antarctica ice-sheets. For GHG levels, 900 ppm (just above the threshold for ice accumulation on Antarctica,) is used at the Eocene pCO2 and 560 ppm for the Oligocene. Because the published model experiments were not coordinated and thus run with a range of CO2, for comparison the output of some models are scaled to approximate the same $\Delta$pCO2. The equilibrium climate states for each model and an ensemble are then assessed for a best fit with observations. Given differences in resolution and other parameters, the absolute T in the models vary widely, with a cool group and warm group, so the focus is primarily on $\Delta$ SAT. In general, most of the models are showing $\sim 2°$C cooling in mean global SAT (figure 5). There are minor regional discrepancies which are attributed primarily to differences in ocean mixing/heat transport. In the end, the ensemble is deemed to show a good fit with observations supporting the conclusion that a reduction in atmospheric CO2 was the primary driver of the EOT.

Thank you for the overall assessment and constructive comments on improving

the manuscript.

I have mixed feelings about this paper. Clearly a considerable amount of time and effort went into compiling the observations, the synthesis of modeling work. This alone will be a valuable contribution to the EOT literature. However, I am not convinced that the modeling comparison is a useful exercise, at least not in the way it was intended or designed.

We have edited the model-data comparison to better acknowledge the limitations and uncertainties that were previously missing from this section. See further discussion below.

The main issue concerns the finding/conclusions about the forcing behind the transition, specifically a ∼halving of pCO2 (from 900 to 560 ppmv) with the EOT. To my knowledge, based on observations (see figure 5) or theory, there is no basis for a 40 to 50% reduction in atmospheric CO2 across this transition. The existing B isotope data, albeit sparse, even suggests a slight increase, and the alkenones suggest a decline but not nearly of that magnitude. More importantly, just from a purely theoretical perspective, there is no reason to expect such a large decline. Recall that when the first detailed, high resolution benthic O isotope records were produced, it became clear that the EOT, or at least the appearance of continental ice-sheets on Antarctica was relatively abrupt, consistent with the threshold hypothesis; a relatively small drop in GHG would be sufficient to trigger the rapid accumulation of ice on a polar continent (i.e., climatic threshold/tipping point, e.g., Crowley and North, 1988). This concept was reinforced by the ice-sheet modeling of DeConto and Pollard 2004 which demonstrated how the local albedo feedback on summertime T could accelerate ice accumulation. Granted that by today's standards the ice-sheet model of the D/P study was relatively coarse and simplistic, but the general theory of a bifurcation point in the climate system still seems valid. We can debate the exact magnitude of the

[Figure]

CO2 drop required, but it was probably small (∼100 ppm), especially with the proper orbital configuration. And even with the large uncertainties, the CO2 proxies are consistent with this hypothesis. This is the most compelling and important aspect of the EOT, a relatively large change in climate in response to a relatively small change in forcing. The observations of a few degrees of cooling, switches in the mode of ocean circulation are consistent with this hypothesis. The Goldner et al (2014) paper nicely illustrates the regional/global effects on ocean T of just adding the ice-sheet (w fixed pCO2). Also, why such a rapid and large reduction in pCO2 at that time? Positive feedbacks involving biogeochemical cycles, ocean uptake, could potentially draw down CO2 but the effect would likely be relatively small, <100 ppm, as suggested by a variety of modeling studies (& observations). More than likely, the decline in pCO2 from the latest Eocene to earliest Oligocene was probably minor.

The reviewer raises some important points here regarding the conclusions drawn from the modelling work. We have considered these carefully and agree that the interpretation of the model-data comparison needs to be changed to acknowledge the limitations and uncertainties of the modelling approach. It was not our intention to suggest that the models are entirely correct and that it was a 350 ppm drop in CO2 that caused the climate transition, but can see that the reader could think that from the way it was written. In order to address the reviewer's comments, we now include a discussion of the caveats; including the mismatch between observations and models, implications for model sensitivity and feedbacks, and recommendations for setting up future experiments. We have also updated the conclusion and the abstract to reflect these changes.

Regarding the CO2 change, the multi-proxy compilation of Foster et al. (2017) suggests 'best-fit' CO2 estimates of 893 ppm and 806 ppm for our late Eocene and early Oligocene windows respectively, or a drop of 10%. We have now acknowledged in the manuscript that our scaled estimate of CO2 change is larger than this best-fit estimate, however we note that there are some CO2 proxies that are consistent

with such a change. For example, the alkenone records of Pagani et al. (2011) are consistent with a decrease of 300 ppm within a 1 Myr window of the EOT. The boron and stomatal records are more equivocal, showing that a drop of 200-300 ppm is plausible, though this can be considered an 'end-member' rather than a 'most probable' change.

The point about crossing a threshold of glaciation is an important limitation on the models used. These models must prescribe an ice sheet to be on or off, with no scope for dynamic feedbacks when crossing a glaciation threshold. We have added further comments here about the limitation in the ice sheet components of the models. However, it is still useful to compare in our ensemble: which forcing mechanisms provide the best explanation of the proxy temperature change, and how should each be relatively weighted. One clear result from our ensemble study is that the far-field temperature change (i.e. much of the global delta T at the EOT), is much better explained by $CO_2$ forcing than by imposing an ice sheet or paleogeographic changes. This highlights the fact that a global cooling mechanism is needed, whether from $CO_2$ or other feedback mechanisms, and such a change is not easily triggered by the gateway or ice sheet forcing experiments.

The results of Goldner et al. (2014), included in our ensemble, also bear this out: they show a large change in Antarctic and Southern Ocean surface temperatures, but little change in the global mean. However, those results do show a cooling of the deep ocean (since the deep ocean waters are sourced from Southern Ocean sinking), which could help to explain some of the benthic temperature proxy trends. Since the models are in general not fully equilibrated, we cannot undertake a model ensemble assessment of benthic temperature changes.

With all this in mind, the fact that the model ensemble is in agreement with observations is problematic. In other words, to match climate observations, a much larger

change in forcing (1.6x) is required than justified by observations or theory. This is the same recurring issue with simulating high CO2 climates of the past, that a much larger forcing is required than justified by observations? The bottom line is that the models are under sensitive to GHG forcing. Arguably, this should be one of the main conclusions of this paper. As the paper is currently written, I almost get the opposite sense. Has this paper achieved the stated goal of identifying what drove the EOT, at least the abrupt appearance of ice-sheets ∼34 Mya? Am sure we can all agree that over the long-term, a reduction in GHG was the primary driver of Eocene cooling and key to triggering Antarctic glaciation. What we won't agree on are the specifics of timing and magnitude, the appropriate alignment of GHG forcing with the climate response.

It is true that some of the simulations appear to be under-sensitive to CO2 forcing when reconstructing Eocene climates. The climate sensitivity to doubling CO2 implied by this ensemble is 3.3°C, which may be lower than necessary to fit to the Eocene proxy record, but not radically so. A recent synthesis of climate sensitivity estimates based on the DeepMIP proxy ensemble suggests that climate sensitivity during the latest Paleocene, PETM and EECO was 4.5°C, 3.6°C and 3.1°C respectively (Inglis et al., 2020), with 66% confidence intervals ranging from 2-7°C during the three intervals combined. Thus, while there is scope for a higher climate sensitivity in the models, the current best estimates are not much higher than the ensemble mean climate sensitivity.

A large component of the Eocene warmth (compared to present day) can be attributed to non-GHG changes, including paleogeographic changes, vegetation, soil and albedo effects. These effects are illustrated more fully in the DeepMIP model ensemble of the early Eocene (Lunt et al., 2020). This shows that stronger GHG forcing is just one element that is needed to properly solve the model-data mismatch in the Eocene (or Oligocene). We have added further discussion to Section 7.2.3 to address these issues.

Recommendation: A key question for revision - what is the real purpose of this paper? If it's simply to provide a comprehensive review of the existing literature on observations and modeling of the EOT, recommendations for future research, minor/moderate revision (see comments below) would suffice. For the reasons stated above, the data model comparison (section 7) could be dropped. If retained, it would be essential to include a discussion of the caveats; the aforementioned mismatch between observations/models, implications for model sensitivity and feedbacks, and recommendations for setting up future experiments.

We have chosen to retain section 7, and as noted above, we have revised the interpretations to clarify the issues raised here. In line with the reviewer's comments, we have acknowledged the need for missing feedbacks to enable the global cooling signal seen in the proxies.

**Additional Comments/Recommendations:**

Window of Observations (Apples vs. Oranges); Considerable effort is spent in defining the duration of the EOT, "Hence the stratigraphic interval of the EOT according to our preferred definition is now given an estimated duration of 790 kyr (fig 1)". The problem is that the collected observations (Figures 3-5) span a much wider range of time, millions of years of the late Eocene/early Oligocene. I assume this is by necessity, especially with the inclusion of terrestrial climate proxies. However, to make the model-data comparison more meaningful, it would be best to only include climate observations that straddle the O isotope excursion at 34 Ma, lets say within windows of 500 kyr immediately above and below. This might exclude a lot of data but the comparison pre- and post EOT conditions would be more meaningful.

[Figure]

Using the larger window was necessary to capture an acceptable spatial extent of proxy data. This is especially the case with the terrestrial data, which generally has lower temporal resolution.

1065 -1067 A figure showing the change in sea-ice distribution would be useful.

We have added a supplementary figure showing the sea ice distribution for each model. This cannot be integrated with our scaled temperature responses, because unlike temperature, sea ice exhibits threshold behaviour that cannot be interpolated smoothly between different levels of CO2.

7.1.3 SAT response to paleogeographic change, ΔTGEO I think the assessment of simulations with differing tectonic configurations, ΔTGEO, is useful only for assessing how the sensitivity of a model to a given change in GHG levels varies under different configurations, e.g., an open or closed Drake passage. We already know from previous studies that the geographic changes alone produce relatively minor changes in global climate, and sometimes in the wrong direction. Also, the uncertainties about the timing of the gateways are large enough that this is not really worth focusing on. The section should be condensed or simply moved to SOM.

While we agree that the TGEO changes produce overall minor changes in global climate, this section is important to include here because the gateway cooling hypothesis is a long-standing theory of what caused the EOT cooling, and some aspects of the gateway hypothesis are still actively debated in modelling literature of the EOT (Toumoulin et al., 2020, e.g.).

1121-22 yes, the CO2 is somewhat arbitrary.

Agreed – but it is important to flag this here.

1124 - As we all know, the error in CO2 reconstructions is quite large. Nevertheless, it is unlikely that CO2 was halved pre-EOT to EOT. More likely, the change in pCO2 was much smaller, at least initially with inception of glaciation (at 34 Ma) which involved a threshold CO2 enhanced by regional feedbacks via ice-sheet growth. It is possible that biogeochemical feedbacks enhance the drawdown of CO2, but probably not more than 100 ppm or so.

We agree that halving CO2 is not a realistic scenario, but it is a clear and useful experimental protocol for establishing the climate sensitivity to GHG changes. We scale the CO2 forcing to best fit the data, and in doing so our derived estimate is much less than a halving. We have clarified that certain feedbacks are missing from these models, and thus the threshold behaviour (which requires dynamic ice sheets) cannot be simulated in them.

1165 – Based on the goodness of fit, you derive an estimate of ΔCO2? I understand the strategy here, but it seems backwards when the primary motivation behind reconstructing paleoclimates is to assess climate models. As stated above, I think the observations suggest that the models (as they were prior to 2017) are under sensitive to GH forcing.

As noted above, we have added further discussion on the models' climate sensitivity to CO2, including recent advances from the DeepMIP intercomparison. We note that the warmer models from that ensemble also have considerable non-CO2 warming effects, i.e. climate sensitivity is not the only factor that can potentially resolve the model-data mismatch.

Figure 5 (pCO2 reconstruction) – This figure made me cringe. The terrestrial proxy

pCO2, given the coarse stratigraphic control, low temporal resolution, could be mis-leading. And let's be honest, given the concerns about the reduced sensitivity of the stomata proxy to higher CO2, who would really expect that proxy to accurately capture the ΔpCO2 across the EOT? Let's not even get into the issues with soil carbonates. As most of the climate data are marine based, it would make sense to only include the pCO2 estimates from marine proxies plotted along with the benthic d18O of figure 2 over a narrower window of time. This would eliminate any uncertainties about the relative timing of changes in climate versus forcing.

We have better acknowledged the challenges of reconstructing CO2 from terrestrial proxies, as noted in our response to Reviewer 1. However, these are an important branch of the EOT literature and we think it is best to include those data, with some added discussion around its limitations. We have also adjusted the age limits of Figure 5 to be shorter (38 to 30 Ma), to be more coherent with our temperature proxy compilations.

**References**

Foster, G. L., Royer, D. L., and Lunt, D. J.: Future climate forcing potentially without precedent in the last 420 million years, Nature Communications, 8, 14 845, https://doi.org/10.1038/ncomms14845, 2017.

Goldner, A., Herold, N., and Huber, M.: Antarctic glaciation caused ocean circulation changes at the Eocene-Oligocene transition, Nature, 511, 574–577, https://doi.org/10.1038/nature13597, 2014.

Inglis, G. N., Bragg, F., Burls, N., Evans, D., Foster, G. L., Huber, M., Lunt, D. J., Siler, N., Steinig, S., Wilkinson, R., Anagnostou, E., Cramwinckel, M., Hollis, C. J., Pancost, R. D., and Tierney, J. E.: Global mean surface temperature and climate sensitivity of the EECO, PETM and latest Paleocene, Climate of the Past Discussions, 2020, 1–43, https://doi.org/10.5194/cp-2019-167, 2020.

Lunt, D. J., Bragg, F., Chan, W.-L., Hutchinson, D. K., Ladant, J.-B., Niezgodzki, I., Steinig, S., Zhang, Z., Zhu, J., Abe-Ouchi, A., de Boer, A. M., Coxall, H. K., Donnadieu, Y., Knorr, G., Langebroek, P. M., Lohmann, G., Poulsen, C. J., Sepulchre, P., Tierney, J., Valdes, P. J., Dunkley Jones, T., Hollis, C. J., Huber, M., and Otto-Bliesner, B. L.: DeepMIP: Model intercomparison of early Eocene climatic optimum (EECO) large-scale climate features and comparison with proxy data, Climate of the Past Discussions, 2020, 1–27, https://doi.org/10.5194/cp-2019-149, 2020.

Pagani, M., Huber, M., Liu, Z., Bohaty, S. M., Henderiks, J., Sijp, W., Krishnan, S., and DeConto, R. M.: The Role of Carbon Dioxide During the Onset of Antarctic Glaciation, Science, 334, 1261–1264, https://doi.org/10.1126/science.1203909, 2011.

Toumoulin, A., Donnadieu, Y., Ladant, J.-B., Batenburg, S. J., Poblete, F., and Dupont-Nivet, G.: Quantifying the Effect of the Drake Passage Opening on the Eocene Ocean, Paleoceanography and Paleoclimatology, 35, e2020PA003 889, https://doi.org/10.1029/2020PA003889, 2020.

---

## Author Response (AR2)

Dear Dr Feng,

Thank you for your helpful comments and fast turnaround of the manuscript. Below we outline our responses to the reviewer in blue text.

Kind Regards,
David Hutchinson

**Reviewer 3**

I was asked to comment on the revised version of the manuscript by Hutchinson and colleagues. I have read the revision along with authors' response to the comments provided by the two previous reviewers. In general, I feel that authors have done an excellent job addressing reviewers' comments and improving the presentation of the manuscript.

In this study, Hutchinson et al. conducted a thorough and up-to-date review on the EOT study. The review includes marine and terrestrial proxy data, model simulations, and model-data comparisons, as well as paleogeography and forcing mechanisms. Even though I have been working on this topic for a while, I could still get some new information from this review. Re-defining and clarifying the terminology used in the EOT study in Section 1 is also very helpful to eliminate the ambiguity of the terminology used in current studies. Thus my overall assessment on this review is very positive and I'd like to see this review officially published after technical corrections/minor revisions. Below I provide some minor comments/suggestions for authors' consideration in their revision, which are mostly for clarification and correcting typos.

We thank the reviewer for their positive assessment and constructive comments on the manuscript.

Comments and Suggestions:

1. Low-latitude TEX86 records: It is pretty clear that some of the low latitude TEX86 records showing large amplitude of EOT cooling is problematic. Liu et al. (2009) made it clear that the signal, especially from Site 803 and 998, may be related to non-thermal factors, and suggested that low-latitude cooling should be on the order of 2-3C. I hope these problematic TEX86 data were not used in authors' data-model comparisons (Fig. 8). Otherwise, it would affect authors' assessment on the model performance. The data points based on these TEX86 records could also be removed from Fig. 3 in order not to mislead readers.

In the previous submission, Site 803 was included in the data compilation and the model-data comparison (Site 998 was not). However, since the reviewer has highlighted that the $TEX_{86}^H$ data from Site 803 are regarded as having a spuriously large temperature change (Liu et al, 2009), we have now removed this site from the model-data comparison. We have also removed Site 803 from Figure 3, and updated our skill metrics and model-data comparison figures accordingly. This changes our global mean metrics slightly, but our main conclusions are unaffected.

2. Authors correctly discussed influencing factors for individual temperature indicators, but I feel the completeness of the individual records also matters in order to derive reliable temperature changes across the EOT. For instance, the available 336 UK'37 SST record only

has a couple of data points from a snapshot time interval. Thus it may not be representative of the mean SSTs for the broadly defined Late Eocene and Early Oligocene periods. As currently data resolution in many SST records is not very high and some key intervals are missing due to sediment hiatus, this could largely contribute to the uncertainty in the estimates of SST changes for the two intervals. Authors could add one or two sentences somewhere to address this issue.

Our priority in creating this data compilation was to ensure as wide a geographic coverage as possible, and therefore to include all sites which recorded a signal on both sides of the transition, even if the sampling is minimal in some cases. To this end, we have used broader averaging windows than might be applied to the highest resolution SST records, but which records a wide global signal. Our choice of averaging helps to capture a response from both marine and terrestrial records, and allows a consistent methodology independent of how coarse or fine the resolution of individual records might be. We have added a discussion at the end of Section 3.1 which clarifies the choices we have made in the averaging windows, and how this affects the SST signal.

3. The time interval to represent post-EOT: Authors used a broad time interval between 33.9 Ma and 30 Ma to represent post-EOT. However, based on the relatively well resolved 1404 SST record, SST values between 29 Ma and 31 Ma could reach the late Eocene level, thus the mean SST over this broad time interval could potentially underestimate post-EOT temperature changes. I would suggest using a time interval broadly consistent with the defined interval for the EOGM, 33.9-33.16 Ma for post-EOT. On the other hand, I recognize that this approach could reduce the number of records to be qualified for this requirement. If the number is significantly reduced, I would suggest discarding my suggestion here, but perhaps it is better to mention in the text that the shorter time interval is more appropriate but due to practical reasons not adopted. Lastly, I note that 1404 SST record was not used in Fig. 3, which could fit the data gap around 40N.

As discussed above, we have mentioned in Section 3.1 the rationale for using a broad averaging window. In particular, we have mentioned that:
"A consequence of this choice is that the averaging may dampen the peak-to-peak signal of EOT SST change in high-resolution records."

U1404 was indeed included in Figure 3, but perhaps it was not obvious, since it was plotted at its paleolatitude of 32°N, rather than its modern day latitude of 40°N. We have clarified in both the figure and the manuscript that these data points are plotted at their paleolatitudes (at 34 Ma), derived using the paleomagnetic reference frame of Torsvik et al. (2012) and cross-checked using the online calculator of van Hinsbergen et al. (2015).

4. CO2 changes across the EOT: Authors mentioned the intriguing rebound of CO2 in Pearson et al. (2009) record. Indeed, such a feature is quite common in alkenone-based CO2 records reconstructed from Southern Ocean. Pagani et al. (2011) suggested that oceanic changes in the Southern Ocean due to opening gateways could bias the long-term CO2 trend, and the reconstruction from tropical sites 925/929 is more representative of atmospheric CO2 changes across the EOT. Also recently Zhang et al. (2020, GCA, 281, 118-134, Fig. 10) have adjusted the assumed b values to allow the reconstructed CO2 values falling into a reasonable range. I'm not sure whether it is worth for authors to discuss this in the text, but it does provide a slightly better justification for the CO2 changes used in model experiments (~300

ppm from data and halving in model), as tropical sites provide larger CO2 changes across the EOT than southern Ocean sites.

We considered this suggestion carefully, and in Section 7.2.2 we note the agreement with the Pagani et al (2011) record:
"A 325 ppmv decrease is within the range of CO2 proxy estimates, shown in Figure 5, with alkenone records in particular suggesting a drop of this magnitude across the EOT (Pagani et al., 2011; Zhang et al., 2013)."
However, given that there are multiple lines of evidence of $CO_2$ change, we feel it is best not to over-emphasise this result. As noted in our response to Reviewer 2, we acknowledge that the 'best fit' $CO_2$ change across the EOT from the multi-proxy compilation of Foster et al (2017) is a decrease of approximately 90 ppmv.

Other minor ones:

1. Table 1: EOGM should be 33.65 Ma to ~33.16 Ma in order to be consistent with the ~490 kyr duration. A typo here? Fig. 1 also suggests 33.16 Ma.

Thanks, this has been corrected to 33.16 Ma.

2. Line 325, Meridional Overturning Circulation: Authors could briefly mention/summarize the various timing of the onset of AMOC suggested in previous studies.

We have edited this paragraph to highlight several perspectives on the onset of the AMOC. These perspectives broadly suggest two alternatives: (i) that the AMOC either started up or expanded at the EOT, (ii) that the AMOC was present in some form from the middle Eocene. Both perspectives are now more clearly explained here.

3. Line 496: The UK'37 index saturates at ~29C for the Muller et al. calibration (1C makes slight difference).

This has been corrected to 29°C.

4. Line 508-509: see my previous comment # 1.

We have added a comment that such a large tropical change in SST is regarded as a spurious signal.

5. Line 519: I believe they suggest an apparent decrease in CO2 over that period, not increase.

Thanks, this correction has been made.

6. Line 576-578. I feel that the 2C surface cooling is OK for tropics, but for high latitude southern hemisphere, the cooling should be >5C. I recognize that some of the records could not resolve the detailed structure of cooling, but if only 2C cooling is assumed at the first step in the Southern Ocean, then additional cooling has to take place at the second step? Perhaps rephrase this sentence?

We have rephrased this to clarify that low latitude surface waters cooled by ~2 °C

"… Step 1 of the EOT was associated with a cooling of both deep waters and low latitude surface waters on the order of 2 °C, while the increase in global ice volume was relatively minor…"

7. Line 721: I think this is related to the completeness of the SST record. The same might be also true for Site 277 and 1090 where data points representing the EOGM interval are generally missing.

We have added a sentence:
"Some of the discrepancies between the temperature signals may be due to differences in sampling rates during key events of the EOT."

8. Line 924-926: So the shallow Barents Sea gateway re-opened some time later? Is there any geological evidence for that?

We have added the following:

[revised manuscript text omitted]